



# Multiphase processes in the EC-Earth Earth System model and their relevance to the atmospheric oxalate, sulfate, and iron cycles

Stelios Myriokefalitakis[1], Elisa Bergas-Massó[2,3], María Gonçalves-Ageitos[2,3], Carlos Pérez García-Pando[2,4], Twan van Noije[5], Philippe Le Sager[5], Akinori Ito[6], Eleni Athanasopoulou[1], Athanasios Nenes[7,8], Maria Kanakidou[9,10,7], Maarten C. Krol[11,12], and Evangelos Gerasopoulos[1]

[1] Institute for Environmental Research and Sustainable Development (IERSD), National Observatory of Athens, Penteli, Greece
[2] Barcelona Supercomputing Center (BSC), Barcelona, Spain
[3] Universitat Politècnica de Catalunya (UPC), Barcelona, Spain
[4] ICREA, Catalan Institution for Research and Advanced Studies, Barcelona, Spain
[5] Royal Netherlands Meteorological Institute (KNMI), De Bilt, the Netherlands
[6] Yokohama Institute for Earth Sciences, JAMSTEC, Yokohama, Japan
[7] Institute for Chemical Engineering Sciences, Foundation for Research and Technology, Patras, Greece
[8] School of Architecture, Civil and Environmental Engineering, École Polytechnique Fédérale de Lausanne, Lausanne, Switzerland
[9] Environmental Chemical Processes Laboratory (ECPL), Department of Chemistry, University of Crete, Heraklion, Greece
[10] Institute of Environmental Physics, University of Bremen, Bremen, Germany
[11] Institute for Marine and Atmospheric Research (IMAU), Utrecht University, Utrecht, the Netherlands
[12] Wageningen University, Wageningen, the Netherlands

*Correspondence to*: Stelios Myriokefalitakis (steliosm@noa.gr)

**Abstract.** Understanding how multiphase processes affect the iron-containing aerosol cycle is key to predict ocean biogeochemistry changes and hence the feedback effects on climate. For this work, the EC-Earth Earth system model in its climate-chemistry configuration is used to simulate the global atmospheric oxalate (OXL), sulfate ($SO_4^{2-}$), and iron (Fe) cycles, after incorporating a comprehensive representation of the multiphase chemistry in cloud droplets and aerosol water. The model considers a detailed gas-phase chemistry scheme, all major aerosol components, and the partitioning of gases in aerosol and atmospheric water phases. The dissolution of Fe-containing aerosols accounts kinetically for the solution's acidity, oxalic acid, and irradiation. Aerosol acidity is explicitly calculated in the model, both for accumulation and coarse modes, accounting for thermodynamic processes involving inorganic and crustal species from sea-salt and dust.

Simulations for present-day conditions (2000-2014) have been carried out with both EC-Earth and the atmospheric composition component of the model in standalone mode driven by meteorological fields from ECMWF's ERA-Interim reanalysis. The calculated global budgets are presented and the links between the 1) aqueous-phase processes, 2) aerosol dissolution, and 3) atmospheric composition, are demonstrated and quantified. The model results are supported by comparison to available observations. We obtain an average global OXL net chemical production of $12.61 \pm 0.06$ Tg yr$^{-1}$ in EC-Earth, with glyoxal being by far the most important precursor of oxalic acid. In comparison to the ERA-Interim simulation,





differences in atmospheric dynamics as well as the simulated weaker oxidizing capacity in EC-Earth result overall in a ~30 % lower OXL source. On the other hand, the more explicit representation of the aqueous-phase chemistry in EC-Earth compared to the previous versions of the model leads to an overall ~20% higher sulfate production, but still well correlated with atmospheric observations.

The total Fe dissolution rate in EC-Earth is calculated at $0.806 \pm 0.014$ Tg Fe yr$^{-1}$ and is added to the primary dissolved Fe
(DFe) sources from dust and combustion aerosols in the model ($0.072 \pm 0.001$ Tg Fe yr$^{-1}$). The simulated DFe concentrations show a satisfactory comparison with available observations, indicating an atmospheric burden of ~0.007 Tg Fe, and overall resulting in an atmospheric deposition flux into the global ocean of $0.376 \pm 0.005$ Tg Fe yr$^{-1}$, well within the range reported in the literature. All in all, this work is a first step towards the development of EC-Earth into an Earth System Model with fully interactive bioavailable atmospheric Fe inputs to the marine biogeochemistry component of the model.





## 1 Introduction

Clouds, fog, and deliquescent aerosols host chemical reactions involving inorganic and organic polar atmospheric compounds (Calvert et al., 1985; Chameides and Davis, 1983; Collett et al., 1999; Donaldson and Valsaraj, 2010; Jacob, 1986; Lelieveld and Crutzen, 1991). These reactions result in the production of species that can neither be formed via gas-phase processes

directly, nor explained solely by primary sources. These compounds participate in chemical transformations across the gas, aqueous, and solid phases. Such multiphase processes have a significant impact on the atmospheric cycles of important inorganic species like sulfur (e.g., Hoyle et al., 2016; Seinfeld and Pandis, 2006; Tsai et al., 2010), and act as a complementary pathway for the formation of organic particulate matter (e.g., Lin et al., 2014; Liu et al., 2012; Myriokefalitakis et al., 2011). The produced inorganic and organic aerosols serve as cloud condensation nuclei and thus affect the Earth's energy balance

(Stocker et al., 2013).

Multiphase processes may also impact the global carbon balance indirectly by altering the atmospheric cycles of species that act as nutrients for the marine biota (Hamilton et al., 2022; Kanakidou et al., 2018; Mahowald et al., 2017; Myriokefalitakis et al., 2020a). Nutrient availability in marine ecosystems is key for the primary production that modulates both the surface oceanic concentrations and the uptake of $CO_2$ (e.g., Le Quéré et al., 2007, 2013; Smith, 2019). A large portion of the global ocean is

found, however, to be iron-limited (Krishnamurthy et al., 2009, 2010), therefore, the importance of iron (Fe) to oceanic productivity is well established (Hamilton et al., 2020; Kanakidou et al., 2020; Meskhidze et al., 2019; Tagliabue et al., 2016). Besides rivers and sea ice, along with sediment dissolution and hydrothermal vents, which are the main sources of bioavailable Fe in the ocean, the atmospheric deposition of nutrients is the most effective external pathway that provides Fe in the open ocean. Fe is a critical micronutrient for marine biota that is mainly utilized in its dissolved form (e.g., aqueous, colloidal, or

nanoparticulate). Thus, the atmospheric processing of Fe-containing minerals, i.e., the conversion from insoluble to soluble that is readily available Fe for marine organisms, is a central step in the atmospheric and marine Fe cycles and directly connected to atmospheric multiphase processes.

Fe is mainly present in the atmosphere in crystalline lattices of aluminosilicates or as iron-oxides in dust aerosols (~95%; Mahowald et al., 2009), and tends to be rather insoluble when emitted (up to ~1 % solubility; Journet et al., 2008). In fact,

observed high Fe solubility downwind of dust source regions can be only explained via the atmospheric processing of dust aerosols (Baker and Jickells, 2017; Oakes et al., 2012). Enhanced Fe solubility is observed for biomass burning aerosols (e.g., ranging 2-46 %; Bowie et al., 2009; Guieu et al., 2005b; Mahowald et al., 2018; Oakes et al., 2012; Paris et al., 2010), depending strongly on the source region and/or the type of burned wood. Significantly higher Fe solubility (up to 80-90 %) is found, however, for Fe-containing aerosols from oil combustion (Ito, 2013, 2015; Ito et al., 2021; Myriokefalitakis et al.,

2011), with Fe in oil fly ash being mainly in the form of ferric sulfate and nanosized iron oxide aggregates (Chen et al., 2012; Schroth et al., 2009). The uncertainty in Fe-containing combustion aerosol solubility is nevertheless also reflected in modeling studies, with some models assuming relatively high solubility at emission (e.g., Hamilton et al., 2019; Myriokefalitakis et al.,





2011) depending on the aerosol size, and others assuming an almost completely insoluble emitted Fe whose solubility is then enhanced during transport via atmospheric processing (Ito, 2015; Ito et al., 2021). Recent multimodel studies estimate an overall global dissolved Fe (DFe) production rate due to atmospheric processing of dust and combustion aerosols of 0.56 ± 0.29 Tg Fe yr$^{-1}$ (Ito et al., 2019; Myriokefalitakis et al., 2018), indicating that a large uncertainty still remains in the impact of atmospheric processing on the mineral Fe solubilization processes.

During atmospheric transport, inorganic strong acids along with organic ligands may coat mineral aerosols and eventually convert part of the contained insoluble Fe forms (e.g., hematite) to bioavailable forms of Fe for marine biota in the euphotic zone (e.g., free ferrous forms, inorganic soluble Fe, and organic Fe-complexes). Mineral dissolution rates depend on the solution's acidity levels, the mineral surface concentration of organic ligands, sunlight, and ambient temperature (e.g., Hamer et al., 2003; Lanzl et al., 2012; Lasaga et al., 1994; Zhu et al., 1993). Although sulfate ($SO_4^{2-}$) is the dominant aerosol species that controls the aerosol liquid water content and acidity, oxalate ($(COO^-)_2$; hereafter OXL) acts as an organic ligand for the Fe-containing aerosol dissolution processes (e.g., Paris et al., 2011; Paris and Desboeufs, 2013) that can effectively break the Fe-O bonds at the mineral's surface via the formation of ligand-containing surface structures (Yoon et al., 2004). Despite the dominant role of acidity in the mineral Fe dissolution processes, modeling estimates (Ito, 2015; Johnson and Meskhidze, 2013; Myriokefalitakis et al., 2015) show the importance of OXL to atmospheric DFe concentrations (e.g., including the formation of Fe(II/III)-oxalato complexes). The dissolution of Fe by OXL may further contribute to the organic-bounded pool of nutrients deposited into the ocean, and thus affects the marine primary production especially in oligotrophic subtropical gyres (e.g., up to 20%; Myriokefalitakis et al., 2020a).

Notwithstanding their different roles and efficiencies in Fe solubilization processes, atmospheric observations demonstrate a strong correlation between $SO_4^{2-}$ and OXL concentrations (Yu et al., 2005), especially above clouds (Sorooshian et al., 2006), indicating common chemical production pathways despite the differences in their precursors and primary sources. $SO_4^{2-}$ and OXL are the most common species formed via aqueous-phase reactions of inorganic and organic origin, respectively, with modeling studies supporting that more than 60% of the sulfates (e.g., Liao et al., 2003) and about 90% of oxalates (Lin et al., 2012; Liu et al., 2012; Myriokefalitakis et al., 2011) are produced in clouds. OXL is the dominant dicarboxylic acid (DCA) in the troposphere (e.g., Kawamura and Ikushima, 1993; Kawamura and Sakaguchi, 1999; Norton et al., 1983) and is formed primarily through cloud processing of glyoxal and other water-soluble products of alkenes and aromatics of anthropogenic, biogenic, and marine origin (Carlton et al., 2007; Warneck, 2003). OXL is mostly present in the troposphere in the particulate form (Yang and Yu, 2008) with aerosol concentrations roughly four times larger than in the gas phase (Martinelango et al., 2007; Yao et al., 2002). OXL can be present in urban environments (Yang et al., 2009) as well as in remote regions (Sempére and Kawamura, 1994), and is produced during the photochemical aging of organic aerosols (Eliason et al., 2003). The observed correlation of OXL with ammonium ($NH_4^+$) (Martinelango et al., 2007) indicates that OXL is mostly present as a salt (i.e., ammonium oxalate; $(NH_4)_2C_2O_4$) in the atmosphere (Paciga et al., 2014). Ortiz-Montalvo et al. (2014) found that in the



presence of $NH_4^+$ under cloud-relevant conditions, the OXL produced by the aqueous-phase glyoxal oxidation is efficiently converted to ammonium oxalate, with its vapor pressure being several orders of magnitude lower than that of oxalic acid. However, in the presence of metals, such as calcium ($Ca^{2+}$) and magnesium ($Mg^{2+}$) from dust and sea salt aerosols, most of the oxalic acid is found to be present in the form of metal complexes (Furukawa and Takahashi, 2011). Nevertheless, due to their different solubility the stability of oxalate complexes can be rather diverse; while calcium- and magnesium-oxalates

precipitate from the solution, other salts, such as sodium or ammonium oxalates, remain in a deliquescent form (Furukawa and Takahashi, 2011).

Laboratory and modeling studies support that OXL is directly produced in atmospheric water via glyoxylic acid (GLX; HC(O)COOH) oxidation by hydroxyl (OH) and nitrate ($NO_3$) radicals. The estimated net global OXL production rate in atmospheric water ranges between 13 and 30 Tg $yr^{-1}$ (Lin et al., 2014; Liu et al., 2012; Myriokefalitakis et al., 2011). However,

modeling studies where the OXL production is only based on the GLX aqueous phase oxidation tend to underestimate its observed atmospheric concentrations (e.g., Lin et al., 2014; Myriokefalitakis et al., 2011). Based on laboratory experiments, Carlton et al. (2007) proposed that predictions of oxalic acid concentrations could be significantly improved when larger multifunctional compounds are allowed to be produced under elevated glyoxal concentrations in typical cloud conditions. These larger multifunctional products can act as precursors for the glyoxylic and oxalic acids via their rapid oxidation by OH

radicals (Carlton et al., 2007). When such reactions are included, models tend to predict a higher oxalate atmospheric load and thus match better the observations (e.g., Myriokefalitakis et al., 2011). Note that although small carbonyl compounds, such as glyoxal and methylglyoxal, can undergo oligomerization under concentrated acidic conditions (Ervens and Volkamer, 2010; Lim et al., 2010, 2013), the mechanism behind the production of larger multifunctional products in dilute solutions may be rather complex, e.g., for products with alcohol functional groups, covalently bonded oligomers, larger carboxylic acids, and

other humic-like substances (HULIS) components (Altieri et al., 2006; Blando and Turpin, 2000; Cappiello et al., 2003; Carlton et al., 2007).

The involvement of Fe chemistry in the aqueous phase decreases overall (by ~57.6 %) the global OXL net production rates, despite the increase in dissolved OH radical sources and thus the oxidation of OXL precursors (Lin et al., 2014). Besides the dissolved $H_2O_2$ photolysis that enhances drastically the OH production in the solution during the daytime, the presence of

transition metal ions (TMIs) may play a central role in aqueous-phase oxidizing capacity, especially under dark conditions (Tilgner et al., 2013; Tilgner and Herrmann, 2018). Among other metals, Fe is the most efficient for the aqueous-phase oxidizing capacity, since on one hand, it contributes to the OH reactivity via the Fenton reaction and the direct Fe photolysis, and on the other hand its dissolved concentrations are high due to the mineral dust contribution. The metal-oxalato complexes formed in the presence of Fe in the solution (Zuo and Deng, 1997), however, can also undergo Fenton reaction and further

increase the dissolved OH source, in particular for air masses of continental origin (Bianco et al., 2020). The photolysis of Fe-oxalato complex $[Fe(C_2O_4)_2]^-$ eventually transforms $C_2O_4^{2-}$ into $CO_2$ in the aqueous phase (Ervens et al., 2003). Overall, it is



clear that the impact of the Fe redox chemistry on the OXL production, and vice versa, is a rather complex issue, that it is expected to also affect the ligand-promoted dissolution process of the Fe-containing minerals under ambient atmospheric conditions.

For this work, we incorporate a comprehensive aqueous-phase chemistry scheme into a state-of-the-art global climate-chemistry model to simulate the atmospheric multiphase processes with respect to iron-containing aerosol dissolution. Section 2 provides an overview of the model, focusing mostly on the new implementations. In particular, we describe the multiphase chemistry scheme used to simulate the atmospheric OXL, $SO_4^{2-}$, and Fe cycles, along with the respective developments for the primary soil and combustion sources applied in the model. In Sect. 3, we present the model-derived OXL, $SO_4^{2-}$, and Fe-
containing aerosol atmospheric concentrations and their evaluation with available observations, and in Sect. 4 we discuss the impact of the simulated aqueous-phase processes on the DFe deposition fluxes to the global ocean. Finally, in Sect. 5, we summarize the global implications of explicitly resolving multiphase chemistry in a climate-chemistry model for the atmospheric Fe cycle, along with the plans for future model development.

## 2 Model description

### 2.1 The EC-Earth3 Earth System Model

Our tropospheric multiphase chemistry developments have been implemented in the global Earth System Model (ESM) EC-Earth3 (Döscher et al., 2021). EC-Earth3 took part in the Coupled Model Intercomparison Project phase 6 (CMIP6; Eyring et al., 2016). The atmospheric General Circulation Model (GCM) of EC-Earth3 is based on cycle 36r4 of the Integrated Forecast System (IFS) from the European Centre for Medium-Range Weather Forecasts (ECMWF), which includes the land surface
model H-TESSEL (Balsamo et al., 2009). The ocean model is the Nucleus for European Modeling of the Ocean (NEMO) release 3.6 (Rousset et al., 2015), with sea ice processes represented by the Louvain-la-Neuve sea ice model (LIM) (Rousset et al., 2015; Vancoppenolle et al., 2009). The ESM presents two configurations: the carbon cycle one that represents the marine biogeochemistry processes through PISCES (Aumont et al., 2015), the dynamic terrestrial vegetation through LPJ-Guess (Smith et al., 2001, 2014), and the atmospheric cycle of $CO_2$ through the Tracer Model version 5 release 3.0 (TM5-MP 3.0) as
well as the EC-Earth3-AerChem configuration (van Noije et al., 2021) that represents the atmospheric chemistry and transport of aerosols and reactive species, also through the TM5-MP 3.0. Most of the information exchange and interpolation between modules is handled through the Ocean Atmosphere Sea Ice Soil version 3 (OASIS3) coupler (Craig et al., 2017). For this work, specifically, we rely on the EC-Earth3-AerChem branch (van Noije et al., 2021).

EC-Earth3-AerChem includes TM5-MP to simulate tropospheric aerosols and the reactive greenhouse gases methane ($CH_4$)
and ozone ($O_3$) and allows the coupling of those species to relevant processes in the atmospheric module IFS (e.g., radiation and clouds). The model can be executed in an atmospheric mode only, i.e., using prescribed sea surface temperature and sea



ice concentration, or coupled to the NEMO-LIM ocean and sea-ice model. In addition, TM5-MP can run as a standalone (offline) atmospheric Chemistry and Transport Model (CTM) driven by meteorological and surface fields (Krol et al., 2005). The present work is structured around a recently released version of TM5-MP that incorporates a rather detailed gas-phase

tropospheric chemistry scheme, the MOGUNTIA (Myriokefalitakis et al., 2020b). MOGUNTIA simulates explicitly the organic polar species that partition in the atmospheric aqueous phase and allows for a sophisticated parameterization of the multiphase processes needed for this study.

All major aerosol components such as sulfate, black carbon, organic aerosols, sea salt, and mineral dust aerosols are included in TM5-MP and are distributed (depending on the aerosol type) in seven lognormal modes, i.e., four soluble modes (i.e.,

nucleation, Aitken, accumulation, and coarse) and three insoluble modes (i.e., Aitken, accumulation and coarse). The aerosol microphysics in the model is calculated by the modal aerosol scheme M7 (Aan de Brugh et al., 2011; Vignati et al., 2004), which represents both the evolution of the total particle number and mass of the different species in each mode. Ammonium, nitrate, and aerosol water are determined based on gas/particle partitioning. Primary emissions of anthropogenic, biogenic, and biomass burning processes are defined through a variety of datasets; the most updated being those produced for the CMIP6

project. Natural emissions of mineral dust, sea salt, marine dimethyl sulfide (DMS), and nitrogen oxides from lighting are calculated online, while other natural emissions are prescribed. Details on the various parameterizations used for the definition of the gas and aerosol emissions in the model can be found in van Noije et al. (2021).

## 2.2 The EC-Earth3-Iron model

EC-Earth3-Iron is the new version of the model developed and used for this work that builds on EC-Earth3-AerChem. The

new features required to determine the global aqueous-phase OXL formation, the atmospheric acidity, and the Fe cycle in the atmosphere can be summarized as:

1. Treatment of mineral dust emission that considers soil mineralogical composition variations to account for the emission of Fe-containing minerals (and calcite), along with a detailed speciation of anthropogenic combustion and biomass burning emissions to explicitly account for Fe both in soluble and insoluble forms,

2. Acidity calculations for water contained in fine and coarse aerosols, as well as, for cloud droplets,

3. A comprehensive aqueous phase chemistry scheme in cloud droplets and aerosol water, and

4. An explicit description of the Fe-containing aerosol dissolution processes of mineral dust, anthropogenic combustion and biomass burning aerosols.





### 2.2.1 Speciated emissions

EC-Earth3-Iron includes a characterization of the dust mineralogical composition at emission and explicitly traces the Fe and calcium-containing species. The relative amounts of eight different minerals, namely: illite, kaolinite, montmorillonite, calcite, feldspars, quartz, gypsum, and hematite, are derived from the soil mineralogy atlas of Claquin et al. (1999), including the updates proposed in Nickovic et al. (2012). The atlas provides the soil mineralogical composition in arid and semi-arid regions of the world, distinguishing between two soil size classes (i.e., the clay-size fraction, up to 2 μm, and the silt-size fraction from

2 to 50 μm diameter). The mineral fractions emitted in the accumulation and coarse insoluble modes of TM5-MP are estimated from the soil mineralogy atlas based on the Brittle Fragmentation Theory (BFT) from Kok (2011). BFT posits that the emitted particle size distribution (PSD) is independent of wind and soil conditions and additionally allows estimating the size-resolved mineral fractions (Pérez García-Pando et al., 2016; Perlwitz et al., 2015a, 2015b). The resulting mineral mass fractions are then applied to the dust emission fluxes, as calculated online in the model, yielding the corresponding accumulation and coarse

mode emission of each mineral. We note that although we derive the mineral dust fractions in each mode using BFT, we maintain the dependence of the emitted dust PSD (i.e., the ratio between the accumulation and coarse mode dust mass at emissions) upon wind and soil conditions of the original dust emission scheme (Tegen et al., 2002).

In EC-Earth3-Iron, the different Fe-containing minerals are not prognostic variables (tracers). Instead, we trace the mineral dust Fe according to three dissolution classes, namely fast, intermediate, and slow Fe pools (Ito and Shi, 2016). Moreover, an

initial solubility of 0.1% to all Fe-containing mineral soil emissions (Ito and Shi, 2016) is considered. The emitted amounts of calcium (i.e., in calcite) and Fe (i.e., in illite, kaolinite, montmorillonite, feldspars, and hematite) are derived either from the average elemental compositions of minerals or based on experimental analyses (Journet et al., 2008; Nickovic et al., 2013). The average fractions between the years 2000 to 2014 applied to mineral dust sources are listed in Table S1.

Fe is also emitted in the model from anthropogenic combustion and biomass burning sources following Ito et al. (2018) and

Hajima et al. (2019). The Fe-containing combustion emissions are estimated here from the total particulate carbon emissions (i.e., the sum of organic carbon and black carbon), based on the Fe content in the sub-micron and super-micron combustion aerosols. The scaling factors for each aerosol size (and solubility) are applied to the sectors energy, industrial, iron and steel industries, residential and commercial shipping, and waste treatment, as well as to the biomass burning emissions. The historical anthropogenic emissions are taken from the Community Emissions Data System (Hoesly et al., 2018) and the

historical fire emissions from the BB4CMIP6 data set (van Marle et al., 2017). Fe-containing aerosol combustion emissions are considered to be insoluble (Ito, 2015), except for ship oil combustion that is assumed to be mostly soluble (~79% on average for the years 2000-2014). The year-to-year variation in anthropogenic combustion Fe-emission fractions follows Ito et al. (2018), except for biomass burning where no variation is considered. The average Fe fractions between the years 2000 to 2014 applied to the total particulate carbon emissions are also listed in Table S1.





EC-Earth3-Iron also includes OXL primary emissions from natural and anthropogenic wood-burning processes, that mainly account for its rapid formation in the sub-grid plumes not represented in the model. Indeed, OXL is well correlated with elemental carbon and levoglucosan (Cao et al., 2017; Cong et al., 2015), which are observed at significant levels during biomass burning episodes in the Amazon (Kundu et al., 2010), suggesting that oxalic acid could be either directly emitted or formed rapidly via combustion processes. During biomass burning episodes enhanced emissions of ionic species have been

generally measured, indicating an average OXL mass concentration measured in plumes of ~0.04-0.07 %w/w (Yamasoe et al., 2000). Furthermore, domestic wood combustion is a potential OXL source (Schmidl et al., 2008) since measurements indicate an OXL contribution to the total particulate concentrations of ~0.09-0.28 %w/w. Gasoline engines may also contribute to total dicarboxylic acid mass emitted to the atmosphere (Kawamura and Kaplan, 1987), although their direct contribution to ambient OXL concentrations is generally found to be low (Huang and Yu, 2007) and, therefore, neglected here. All in all, primary OXL

sources are quite uncertain and, given the current estimates, may only have a limited impact on the calculation of its atmospheric concentrations (e.g., Myriokefalitakis et al., 2011).

### 2.2.2 Thermodynamic equilibrium and atmospheric acidity calculations

The gas/particle equilibrium calculations of $NH_3/NH_4^+$ and $HNO_3/NO_3^-$ have been substantially revised in EC-Earth3-Iron. In EC-Earth3-AerChem, EQSAM (Metzger et al., 2002) is used to determine the partitioning of $NH_3/NH_4^+$ and $HNO_3/NO_3^-$. In

EC-Earth3-Iron, the ISORROPIA II thermodynamic equilibrium model (Fountoukis and Nenes, 2007), replaces EQSAM to determine the equilibrium between the inorganic gas and the aerosol phases. ISORROPIA-II calculates the gas/liquid/solid equilibrium partitioning of the $K^+$-$Ca^{2+}$-$Mg^{2+}$-$NH_4^+$-$Na^+$-$SO_4^{2-}$-$NO_3^-$-$Cl^-$-$H_2O$ aerosol system, and is used in the forward mode, assuming that all aerosols are in a metastable (liquid) state. The inclusion of sea salt and dust aerosols in the aerosol thermodynamic calculations has been shown, nevertheless, to substantially affect the ion balance and thus the partitioning of

$HNO_3/NO_3^-$ and $NH_3/NH_4^+$ species, especially in areas with abundant mineral dust and/or sea spray aerosols (Athanasopoulou et al., 2008, 2016; Karydis et al., 2016). In EC-Earth3-Iron nitrate aerosols are calculated for both the accumulation and coarse modes, in contrast to the bulk aerosol approximation used in the EC-Earth3-AerChem. For this, kinetic limitations by mass transfer and transport between the gas and the particulate phases in accumulation and coarse modes (Pringle et al., 2010) are considered, with ISORROPIA-II then re-distributing the respective masses between the gas and the aerosol phases. We note

that $Ca^{2+}$ from calcite is simulated prognostically in the model based on mineralogy maps (Sect. 2.2.1), in contrast to other crustal elements in soils that are calculated by assuming constant mass ratios to dust concentrations of 1.2%, 1.5%, and 0.9% for $Na^+$, $K^+$, and $Mg^{2+}$, respectively (Karydis et al., 2016; Sposito, 1989). For sea spray aerosols, mean mass fractions of 55% $Cl^-$, 30.6% $Na^+$, 7.7% $SO_4^{2-}$, 3.7% $Mg^{2+}$, 1.2% $Ca^{2+}$, and 1.1% $K^+$ (Seinfeld and Pandis, 2006) are also applied.

The acidity levels of deliquescent aerosols are calculated in the model based on thermodynamic processes for accumulation

and coarse particles. Aerosol acidity impacts the scavenging efficiency and the dry deposition of inorganic reactive nitrogen species due to changes in the partitioning of total nitrate and ammonium between the gas and aerosol phases as well as between


the various aerosol sizes (Pye et al., 2020). Acidity levels also play a fundamental role in the aqueous-phase chemistry by controlling the dissociation reactions and thus, the reactivity of the chemical mechanism. Indeed, aqueous-phase species, such as organic and inorganics acids, are oxidized with higher rates when they are dissociated. Nevertheless, in the case of the

forward and reverse reactions, they typically occur fast and thus the concentrations of the reactants and the products are generally assumed to be in equilibrium in the global model due to its relatively long timestep and large model grid. Note, however, that recent modeling studies showed that the metastable assumption applied here could lead to an increase of aerosols' acidity (i.e., regionally up to 2 pH-units in the presence of crustal elements over dust sources, and roughly 0.5 pH units globally; Karydis et al., 2020) compared to the stable aerosol state assumption (i.e., the aerosols both in solid and liquid phases).

Under ambient atmospheric conditions, the water vapor uptake on aerosols depends on both the inorganic and organic components, along with the meteorological conditions (e.g., the temperature and the relative humidity conditions). ISORROPIA II does not, however, include water associated with organic aerosols, possibly leading to an underestimation of the aerosol hygroscopicity especially within the boundary layer where the contribution of water-soluble organics to total aerosol mass can be substantial. For this, we account here for a contribution of aerosol water from organic particles in the

acidity calculations, using a hygroscopicity parameter $\kappa_{org}=0.15$ (Bougiatioti et al., 2016). In more detail, the particulate water due to the organics ($W_{org}$) that is added to the aerosol water associated with the inorganic aerosol as calculated from ISORROPIA-II ($W_{inorg}$), is determined in the model as:

$$W_{org} = m_s \cdot \frac{\rho_w}{\rho_s} \cdot \frac{\kappa_{org}}{(\frac{1}{RH}-1)} \tag{1}$$

where $m_s$ is the soluble organic mass concentration ($\mu g\ m^{-3}$) as simulated by the TM5-MP chemistry scheme, $\rho_w$ is the water

density (1 kg m$^{-3}$), $\rho_s$ is the organic aerosol density (1.4 kg m$^{-3}$), and RH (0-1) is the relative humidity.

Cloud acidity is also an important factor for simulating the multiphase processes in the atmosphere. The in-cloud proton concentration is initially determined by the electroneutrality of strong acids and bases (i.e., $H_2SO_4$, $SO_4^{2-}$, methanesulfonate ($MS^-$), $HNO_3$, $NO_3^-$, and $NH_4^+$), and then the subsequent dissociations of $CO_2$, $SO_2$, and $NH_3$ (Jeuken et al., 2001) are solved iteratively in the model. For the cloud acidity calculations, the liquid water content, and the respective cloud cover fraction

(i.e., 0-1) are obtained from meteorology. Note, however, that the effect of mineral dust, and especially calcium, on cloud proton concentrations is neglected. This assumption may result in some overestimation of cloud acidity, although the overall impact should be small particularly in dusty areas with a low presence of clouds. Another limitation in the determination of cloud acidity is the omission of light gaseous organic acids (such as formic and acetic acids), possibly leading to some underestimation in cloud acidity where their concentration is important.





### 2.2.3 The aqueous phase chemistry scheme


The aqueous-phase chemistry scheme used in this work is based to a large extent on the Chemical Aqueous Phase Radical Mechanism (CAPRAM) (e.g., Deguillaume et al., 2004; Ervens et al., 2003; Herrmann et al., 2000, 2015). However, CAPRAM includes more than 70 aqueous-phase species, 34 equilibria for compounds that are present both in the gas and the aqueous phases, along with numerous photolytic and aqueous-phase reactions, also covering a large series of acid-base and metal-

complex equilibria. Note that various updates may further extend the mechanism by including among others, the oxidation of aromatic hydrocarbons (Hoffmann et al., 2018), the multiphase oxidation of DMS (Hoffmann et al., 2016), and the tropospheric multiphase halogen chemistry (Bräuer et al., 2013). For this, some reactions are considered here in a more simplified way based on various assumptions published in literature. Indeed, the level of chemical complexity of such a detailed mechanism is beyond the computational resources available for three-dimensional global climate-chemistry simulations, and thus

simplifications, that preserve however the essential features of the aqueous mechanism, are needed.

Aqueous-phase chemical transformations are considered at the interface and in the bulk, initiated mainly by free radicals and oxidants produced both via photochemical reactions and in dark conditions (Bianco et al., 2020). The sources of OH radicals in the aqueous phase, however, strongly differ from those in the gas phase, primarily because of the presence of ionic species and TMIs in the solution. OH radicals are the main oxidant in the aqueous phase, either produced directly in the aqueous

medium or diffused from the gas phase (i.e., via a gas-to-liquid transfer). However, aqueous phase oxidation can also be induced by non-radical species, such as ozone ($O_3$) and hydrogen peroxide ($H_2O_2$). A characteristic example is the formation of $SO_4^{2-}$ in cloud droplets, via the oxidation of dissolved sulfur dioxide ($SO_2$) by $O_3$ and $H_2O_2$, with $H_2O_2$ being nevertheless the most effective oxidant (Seinfeld and Pandis, 2006), especially when the solution becomes acidic. Upon the absorption of $SO_2$ in cloud droplets, the establishment of the equilibrium between the dissolved sulfur species in oxidation state 4, i.e.,

$SO_2.H_2O$, $HSO_3^-$ ($pKa_1 = 1.9$), and $SO_3^{2-}$ ($pKa_2 = 7.2$) (hereafter also as S(IV)) is calculated in the model. Then, depending on the availability of oxidants and the solution's acidity, the different S(IV) species can participate in the formation of S(VI) (i.e., dissolved sulfur in oxidation state 6).

In EC-Earth3-Iron, the aqueous-phase sulfur scheme is applied both in cloud droplets and aerosol water, replacing the S(VI) production through the dissolved S(IV) oxidation in cloud droplets previously included in the EC-Earth-AerChem (van Noije

et al., 2014, 2021). In more detail, besides the two classic reactions of bisulfite and sulfite with hydrogen peroxide and ozone included in EC-Earth3-AerChem, additional reactions of S(IV) oxidation via methyl hydroperoxide ($CH_3O_2H$), peroxyacetic acid, and with the hydroperoxyl radical ($HO_2$)/superoxide radical anion ($O_2^-$) are considered. Nevertheless, in acidic solutions, the oxidation by peroxides, and especially $H_2O_2$, is significantly more important than other oxidants (Herrmann, 2003; Jacob, 1986). $H_2O_2$ is produced in the gas phase and can be rapidly dissolved in the liquid phase due to its high solubility. The

dissolved $H_2O_2$ (as well as the organic peroxides, such as $CH_3OOH$) can react rapidly with the $HSO_3^-$. However, the pH-independent reaction of $HSO_3^-$ with $CH_3OOH$ (or other organic peroxides) is expected to be less important than $H_2O_2$ under





typical cloud conditions due to the much lower solubility of $CH_3OOH$. Note that the dissociation of $H_2O_2$ is here neglected since it is not expected to significantly influence the total $H_2O_2$ concentrations under typical tropospheric conditions (Herrmann, 2003; Jacob, 1986). In contrast, at a higher pH, the S(IV) oxidation by ozone tends to dominate the S(IV) oxidation
(Seinfeld and Pandis, 2006). $O_3$ oxidizes rapidly all three S(IV) forms in the aqueous phase, becoming significant at pH higher than 4 (Seinfeld and Pandis, 2006), even in the absence of light. S(IV) oxidation by $O_3$ is also predicted to dominate S(VI) formation during winter in arctic regions due to the lack of photochemical production of OH and $H_2O_2$ at high latitudes, as well as to the high anthropogenic $SO_2$ emissions in the Northern Hemisphere (Alexander et al., 2009). Laboratory studies indicate that S(IV) compounds may be also oxidized in the aqueous phase via other pathways. For example, the aqueous S(VI)
production can be enhanced by TMIs (Harris et al., 2013), such as the Mn(II) catalyzed oxidation of S(IV) by dissolved $O_2$. In a global modeling study, Alexander et al. (2009) attributed 9-17% of the total S(VI) production to the latter mechanism. However, such reactions would require several oxysulfur radicals as intermediates (e.g., Deguillaume et al., 2004; Herrmann et al., 2005), like a free radical chain mechanism initiated by reactions of $HSO_3^-$, $SO_3^{2-}$ with radicals and radical anions, or TMIs catalyzed oxidation of several S(IV)-compounds, which is not considered in our model. Thus, in the case of the sulfate
radical anion ($SO_4^-$) production via the Fe(III)-sulfato-complex $[Fe(SO_4)]^+$ photolysis (Table S2), the sulfate radical anion is simply added to the S(VI) pool.

Gas-phase organics can be also oxidized in the interstitial cloud space, form water-soluble compounds like aldehydes, and rapidly partition into the droplets. In the presence of oxidants such as OH and $NO_3$ radicals in the solution, the dissolved organics undergo chemical conversions and form low volatile organics that remain, at least partly, in the particulate phase
upon droplet evaporation (Blando and Turpin, 2000). The dissolved OH radicals react with organic compounds in the aqueous phase by hydrogen abstraction or electron transfer, forming alkyl radicals (R) which, in the presence of dissolved oxygen, further form peroxyl radicals ($RO_2$). The OH oxidation of organic compounds in the aqueous phase can lead either to fragmentation or, to the formation of oxidized organic species, resulting overall in $CO_2$. However, the recombination of organic radicals can also be a favorable pathway when the water evaporates, and the aqueous solution becomes more concentrated.
Box-model simulations have shown that the cloud processing of polar products from isoprene oxidation can be an important contributor to secondary organic aerosol (SOA) production (Lim et al., 2005). Indeed, laboratory measurements show that the aqueous-phase photooxidation of C2 and C3 carbonyl compounds (Perri et al., 2009, 2010), such as glyoxal (Carlton et al., 2007, 2009), methylglyoxal (Altieri et al., 2008), glycolaldehyde, pyruvic acid (Carlton et al., 2006), and acetic acid (Tan et al., 2012) leads to the production of low volatility DCAs, which are commonly found in atmospheric aerosols and clouds
(Sorooshian et al., 2006).

In EC-Earth3-Iron, gas-phase species can be reversibly transferred to the aqueous phase and oxidized by radicals and radical anions. The partitioning of 15 organic species that exist in both phases are considered in the aqueous-phase mechanism, namely methyl-peroxy radical ($CH_3O_2$), methyl hydroperoxide ($CH_3O_2H$), formaldehyde (HCHO), methanol ($CH_3OH$), formic acid





(HCOOH), acetaldehyde ($CH_3CHO$), glycolaldehyde (GLYAL; $HOCH_2CHO$), glyoxal (GLY; $CH(O)CH(O)$), ethanol
($CH_3CH_2OH$), acetic acid ($CH_3COOH$), methylglyoxal (MGLY; $CH_3C(O)CHO$), hydroxyacetone (HYAC; $CH_3C(O)CH_2OH$),
pyruvic acid (PRV; $CH_3C(O)COOH$), GLX, and oxalic acid ($H_2C_2O_4$). The aqueous phase oxidation is taking place by the OH
and $NO_3$ radicals, as well as the $CO_3^-$ radical anion. OH is either produced by photolytic reactions of dissolved compounds or
via a direct transfer from the gas phase into the solution, as well as by Fenton reaction (Deguillaume et al., 2010). $NO_3$ radicals
are transferred from the gas phase, while the $CO_3^-$ radical anion is produced mainly via the oxidation of hydrated $CO_2$. In
general, the aqueous phase oxidation largely proceeds via OH radicals, followed by $NO_3$ radicals under dark conditions, while
the $CO_3^-$ radical has an overall small impact on the oxidizing capacity of the solution.

Upon their transfer to the solution, aldehydes are considered to be in equilibrium with the corresponding diols. The hydrated
aldehydes are oxidized via H-atom abstraction with radicals (OH, $NO_3$) or radical anions ($CO_3^-$), followed by the elimination
of $HO_2$ in reaction with $O_2$, leading overall to the formation of organic acids. Alcohols, such as $CH_3OH$ and $C_2H_5OH$, are also
oxidized via an H-atom abstraction; the resulting α-hydroxy-alkyl radicals, however, are not explicitly resolved, but the direct
formation of aldehydes (e.g., formaldehyde and acetaldehyde) is considered via the respective peroxyl radical reactions with
molecular oxygen to yield $HO_2$. Moreover, the glycolic acid ($HOCH_2COOH$) production via glycolaldehyde oxidation is not
also explicitly described in the aqueous-phase scheme, and only the direct production of GLX is considered (Lin et al., 2012;
Myriokefalitakis et al., 2011). This assumption is expected to have a negligible impact on the overall chemical mechanism
since the glycolic acid is rapidly oxidized into glyoxylic acid with its net in-cloud production being rather small (Liu et al.,
2012).

After cloud evaporation, OXL and $SO_4^{2-}$ are considered to reside entirely in the particulate phase of the model. This
approximation may, nevertheless, result in an overestimate of OXL ($pKa_1 = 1.23$; $(COO^-)_2$, $pKa_2 = 4.19$) concentrations, since
low levels of gas-phase oxalic acid have been also observed in the atmosphere under favorable conditions (e.g., Baboukas et
al., 2000; Martinelango et al., 2007). Note that other products, such as pyruvate, glyoxylate as well as the oligomers from GLY
and MGLY, are also considered to reside in the particulate phase upon cloud evaporation (Lim et al., 2005; Lin et al., 2012;
Liu et al., 2012) and are thus added directedly to the SOA pool of the model. However, in contrast to OXL and the low-
volatility oligomers, the pyruvic and glyoxylic acids are allowed to be partially transferred back to the gas phase of the model
when the cloud droplets evaporate.

For the present work, the aqueous reaction rate coefficients are taken (where available) from the available literature of the
CAPRAM schemes and supplemented with reaction rates from laboratory and modeling studies (i.e., Carlton et al., 2007;
Deguillaume et al., 2009; Lim et al., 2005; Sedlak and Hoigné, 1993). For the sulfur chemistry, the aqueous reaction rates are
taken from Seinfeld and Pandis (2006). In the case of missing experimental data for temperature dependencies, the rate
constants for T = 298 K are only applied in chemistry calculations. $O_3$, $H_2O_2$, $NO_3$, $HONO/NO_2^-$, $HNO_3/NO_3^-$, and $CH_3O_2H$,
along with $Fe^{3+}$, $[Fe(SO_4)]^+$ and $[Fe(OXL)_2]^-$ are photolyzed in the aqueous phase. Aqueous photolysis frequencies are taken





from the gas-phase chemistry (where available), and increased in the case of cloud droplets due to refraction effects by a factor of 1.5 (Barth et al., 2003). For Fe-species (e.g., $Fe^{3+}$, $[Fe(SO_4)]^+$, $[Fe(OXL)_2]^-$), their maximum (i.e., noontime at 51° N) photolysis frequencies, as proposed by Ervens et al. (2003), are scaled based on the gas-phase $H_2O_2$ photolysis rates. A list of all aqueous and photochemical reactions included in the chemical scheme of this study is presented in Table S2, with the

respective equilibrium reactions shown in Table S3.

### 2.2.4 The iron solubilization scheme

A three-stage kinetic approach (Shi et al., 2011) is applied to describe the solubilization of the Fe-containing dust mineral pools (Ito and Shi, 2016); i.e., representing: 1) a rapid dissolution of ferrihydrite on the surface of minerals (i.e., fast pool), 2) an intermediate stage dissolution of nano-sized Fe oxides from the surface of minerals (i.e., intermediate pool), and 3) the Fe

release from heterogeneous inclusion of nano-Fe grains in the internal mixture of various Fe-containing minerals, such as aluminosilicates, hematite, and goethite (i.e., slow pool). A separate Fe pool for combustion aerosols (Ito, 2015) is also considered in the model.

The dissolved Fe in the model is produced via dissolution processes in aerosol water and cloud droplets depending on the acidity levels of the solution (i.e., proton-promoted dissolution scheme), the OXL concentration (i.e., ligand-promoted

dissolution scheme), and irradiation (photo-reductive dissolution scheme), following Ito (2015) and Ito and Shi (2016). The Fe release from different types of minerals depends, thus, on the solution acidity (pH) and the temperature (T), as well as on the degree of solution saturation. In more detail, the dissolution rates for each of the three dissolution processes considered can be empirically described (e.g., Ito, 2015; Ito and Shi, 2016; Lasaga et al., 1994) as:

$$RFe_i = K_i(pH,T) \cdot \alpha(H^+)^{m_i} \cdot f_i \cdot g_i \tag{2}$$

where $K_i$ (moles Fe $g_i^{-1}$ $s^{-1}$) is the Fe release rate due to the dissolution process $i$, $\alpha(H^+)$ is the $H^+$ activity of the solution, $m_i$ is the empirical reaction order for protons derived from experimental data, and the functions $f_i$ and $g_i$ represent the suppression of the different dissolution rates due to the solution saturation state (Eq. 4 and Eq. 5). The net Fe dissolution rate results from the sum of the three rates. The activation energy that accounts for the temperature dependence is derived as a function of acidity based on soil measurements (Bibi et al., 2014; Ito and Shi, 2016), i.e.:

$$E_{pH} = -1.56 \; 10^3 \cdot pH + 1.08 \; 10^4 \tag{3}$$

The functions $f_i$ and $g_i$ represent the suppression of the different dissolution rates due to the solution saturation state, i.e.:

$$f_i = 1 - (a_{Fe^{3+}} \cdot a_{H^+}^{-n_i})/K_{eq_i} \tag{4}$$

$$g_i = 0.17 \cdot ln(\frac{a_{OXL}}{a_{Fe^{3+}}}) + 0.63 \tag{5}$$



where, $\alpha_{H+}$, $\alpha_{Fe+3}$, and $\alpha_{OXL}$ stand for the solution's activities of 1) protons, 2) ferric cations, and 3) OXL, respectively, as
calculated each time step in the model, and $K_{eqi}$ (mol$^2$ kg$^{-2}$) is the equilibrium constant. All parameters used for the calculation
of dissolution rates for this work are presented in Table S4.

**2.3 The chemistry solver**

All concentrations of gas, aqueous, and aerosol species evolve dynamically in the model. The ordinary differential equations
that govern the production and destruction terms due to chemical reaction and interphase mass transfer in the model are:

$$\frac{dG}{dt} = R_G - LWC k_{mt} G + \frac{K_{mt}}{HRT} A \tag{6}$$

$$\frac{dA}{dt} = R_A + LWC k_{mt} G - \frac{K_{mt}}{HRT} A \tag{7}$$

where,

G = Gas-phase concentrations (molecules cm$^{-3}$ of air)

A = Aqueous-phase concentrations (molecules cm$^{-3}$ of air)

$R_G$ = Gas-phase reaction terms (molecules cm$^{-3}$ of air s$^{-1}$)

$R_A$ = Aqueous-phase reaction terms (molecules cm$^{-3}$ of air s$^{-1}$)

LWC = Liquid water content (cm$^3$ of water cm$^{-3}$ of air)

$k_{mt}$ = Mass transfer coefficient (s$^{-1}$)

H = Henry's Law coefficient (moles L$^{-1}$ atm$^{-1}$)

R = Ideal gas constant (L atm mol$^{-1}$ K$^{-1}$)

T = Temperature (K)

The mass transfer between the gas- and aqueous phases (Lelieveld and Crutzen, 1991; Schwartz, 1986) is applied only for
those species that exist in both phases and is represented in the mechanism by two separate reactions, i.e., one reaction for
transfer from the gas to the aqueous phase and one for the transfer from the aqueous to the gas phase. All Henry's law solubility
constants (H) used in this work are taken from Sander (2015) and are presented in Table S5.

The mass transfer coefficient ($k_{mt}$) for a species is calculated as:

$$k_{mt} = \left( \frac{r^2}{3D_g} + \frac{4r}{3v\alpha} \right)^{-1} \tag{8}$$



where $r$ is the effective droplet or aqueous aerosol radius (m), $D_g$ is the gas-phase diffusion coefficient (m$^2$ s$^{-1}$), $v$ the mean molecular speed (m s$^{-1}$), and $\alpha$ the mass accommodation coefficient (dimensionless). $D_g$ and $\alpha$ used for this study are also

presented in Table S5. The mean molecular speed of a gaseous species is calculated as:

$$v = \sqrt{\left(\frac{8R_g T}{\pi M_W}\right)} \qquad (9)$$

where, $M_W$ is the respective molecular weight (kg mol$^{-1}$) and $R_g$ is the ideal gas constant (J mol$^{-1}$ $K^{-1}$) (Herrmann et al., 2000). The cloud droplet effective radius may vary between ~3.6 - 16.5 μm for remote clouds, 1 - 15 μm for continental clouds, and ~1 - 25 μm for polluted clouds (Herrmann, 2003). For this work, the effective radius of cloud droplets (ranging between 4-30

μm in the model) is calculated online based on the cloud liquid water content and the cloud droplet number concentration (van Noije et al., 2021). The effective radii (i.e., the ratio of the third to the second wet aerosol moments) for the accumulation and coarse deliquescence particles, are based on the respective M7 calculations. According to Eq. (8), the gas transfer to small droplets is faster, owing to the larger surface-to-volume ratio of smaller droplets. However, sensitivity model simulations using different droplet radii showed that varying droplet sizes result only in small changes in the chemical production of aqueous-

phase species (Lelieveld and Crutzen, 1991; Liu et al., 2012; Myriokefalitakis et al., 2011).

KPP version 2.2.3 (Damian et al., 2002; Sandu and Sander, 2006) was used to generate the Fortran 90 code for the numerical integration of the aqueous-phase chemical mechanism. For this, a separate model driver was developed to arrange the respective couplings to the TM5-MP I/O requirements (e.g., species that partition in the aqueous-phase, the reaction and dissolution rates, and the photolysis coefficients). The Rosenbrock solver is used in this work as the numerical integrator, since

it is found to be rather robust and capable of integrating very stiff sets of equations (Sander et al., 2019). However, as for the case of the gas-phase mechanism's coupling (Myriokefalitakis et al., 2020b), minor changes were needed to be applied in the original KPP code. For instance, the aqueous and photolysis reactions are not calculated inside KPP, but directly provided through calculations in the aqueous chemistry driver. In contrast, for the Fe dissolution scheme, the suppressions of the mineral dissolution rates due to the solution saturation are calculated online by KPP (see Eq. 4 and Eq. 5).

**2.4 Simulations**

We performed a range of present-day simulations, including experiments using EC-Earth3-Iron atmosphere-only runs (hereafter referred to as EC-Earth), and TM5-MP standalone driven by ERA-Interim (Dee et al., 2011) reanalysis fields (hereafter referred to as ERA-Interim), covering the period 2000-2014. For the EC-Earth simulation, TM5-MP is coupled to the IFS atmospheric dynamics. We used prescribed sea surface temperature and sea-ice concentration fields from a set of input

files through the AMIP interface (Taylor et al., 2000). Thus, for the atmosphere and chemistry modules, our setup follows the EC-Earth3-AerChem standard configuration in CMIP6 experiments. The IFS horizontal resolution is T255 (i.e., a spacing of roughly 80 km), 91 layers are used in the vertical direction up to 0.01 hPa, and a time step of 45 min is applied. Respectively,



TM5-MP (both for the online and offline configurations) has a horizontal resolution of 3° in longitude by 2° in latitude and 34 layers in the vertical direction up to 0.1 hPa (~60 km).

The ERA-Interim setup allows constraining the model with the assimilated observed atmospheric circulation data and is therefore used for budget analysis and comparison with other estimates from the literature. ERA-Interim is further used to explore uncertainties regarding the aqueous-phase chemistry scheme. Specifically, an additional simulation is performed to identify the potential importance of glyoxal-derived oligomers and high molecular weight species in the aqueous phase (Carlton et al., 2007) on the OXL production rates and the respective ambient concentrations. In this sensitivity simulation,

(hereafter referred to as ERA-Interim(sens)), the OXL formation via high molecular weight species formation from glyoxal oxidation is neglected. Comparisons between the corresponding 15-year climatologies from the EC-Earth and ERA-Interim simulations are used to identify uncertainties in the aqueous-phase production terms of OXL, the iron-dissolution rates, and finally the atmospheric concentrations and deposition rates of Fe-containing aerosols due to the applied meteorology (i.e., online vs. offline). Note that the same emission datasets are used both in the ERA-Interim driven and the EC-Earth3-Iron

experiments, thus only natural primary sources depending on meteorology may differ (see Sect. 2.1). A summary of the simulations is listed in Table 1.

### 2.5 Observations

A general evaluation of the modeled Aerosol Optical Depth (AOD) at 550 nm allows for characterizing EC-Earth3-Iron's ability to reproduce the aerosol fields. The Aerosol Robotic Network (AERONET) version 3 (Giles et al., 2019) level 2.0 direct

sun retrievals at a monthly basis are used to calculate annual mean AOD values for the 2000-2014 period. The model's coarse horizontal resolution hinders, however, the representation of high-altitude locations, thus, following Huneeus et al. (2011), we exclude sites above 1000 m asl., leaving 738 locations with information available during the simulated period. In addition, we perform a specific evaluation of mineral dust, which constitutes a key modulator of the outcome of our new developments as a source of Fe and Ca. To that end, we apply two additional filters to the AERONET data mentioned above, also following

Huneeus et al. (2011), to identify dust-dominated sites. First, we exclude those sites where the monthly mean Angstrom exponent is above 0.4 more than 2 months in the selected period. To further discriminate dust from sea salt, a minimum threshold of 0.2 for AOD at 550 nm is considered (i.e., if more than half of the retrieved AOD is above that threshold, the site is considered as dust-dominated). This filtering allows identifying a subset of stations potentially dominated by dust aerosols, however, it cannot ensure that there is no influence of other aerosol types in the monthly retrievals. Therefore, the evaluation

of AOD at 550 nm at those sites is taken as a proxy for the dust optical depth, acknowledging that other aerosols may also be present.

Pure dust measurements of surface concentration and deposition complement our evaluation of the model. The modeled annual mean surface dust concentration for 2000-2014 is compared to climatological observations from the Rosenstiel School of



Marine and Atmospheric Science (RSMAS) of the University of Miami (Arimoto et al., 1995; Prospero, 1996, 1999; Prospero
et al., 1989) and the African Aerosol Multidisciplinary Analysis (AMMA) international program (Marticorena et al., 2010)
observations. The 23 available sites cover locations close to sources (e.g., the AMMA stations over the Sahelian dust transect),
in transport regions (e.g., stations from RSMAS on the Atlantic), and remote regions (e.g., RSMAS sites close to Antarctica).
The modeled dust deposition fluxes are compared to the compilation of observations for the modern climate in Albani et al.
(2014) including measurements at 110 locations, the mass fraction for particles with diameter lower than 10 μm is used to
keep the observed mass fluxes within the range of the modeled sizes.

Model simulations are also evaluated against *in-situ* (surface and cruise) observations. The simulated OXL and $SO_4^{2-}$
concentrations are compared against measurements for representative sites, such as the Eastern Mediterranean (Finokalia,
Greece; Koulouri et al., 2008), central Europe (Puy de Dome, France; Legrand et al., 2007), and the North Atlantic Ocean
(Azores Portugal; Legrand et al., 2007). Simulated monthly mean surface concentrations of OXL are also compared against a
range of observations (n=143) from remote sites around the world as compiled in Myriokefalitakis et al. (2011). Moreover,
$SO_4^{2-}$ monthly mean surface concentrations over Europe and the USA are also compared against observations (n=3828)
obtained from the European Monitoring and Evaluation Programme (EMEP; http://www.emep.int) and the Interagency
Monitoring of Protected Visual Environments (IMPROVE; http://vista.cira. colostate.edu/improve/), respectively, as compiled
in Daskalakis et al. (2016). The simulated Fe-containing aerosol concentrations are evaluated against cruise measurements
covering a period from late 1999 up to early 2015, as compiled by Myriokefalitakis et al. (2018) and Ito et al. (2019a), and
include daily observations for fine, coarse, and the total suspended particles.

Statistical parameters are here used to demonstrate the model performance. These are the correlation coefficient (R) that
reflects the strength of the linear relationship between model results and observations (i.e., the ability of the model to simulate
the observed variability), the normalized mean bias (nMB), and the normalized root mean square error (nRMSE) as a measure
of the mean deviation of the model from the observations due to random and systematic errors. The equations used for the
statistical analysis of model results are provided in the Supplement (Eq. S1–S3) and the locations (and regions) of the various
observations used for evaluating the model for this work are presented in Fig. 1. Overall, a summary of statistics used for the
evaluation of model simulations with the observations is presented in the Appendix (see Tables A1 and A2).

## 3 Results

**3.1 Budget calculations**

The chemical production and destruction terms of OXL and its precursors, along with the Fe-containing aerosols' dissolution
rates from combustion (FeC) and mineral dust (FeD), their emissions, and their removal terms from the atmosphere are
presented for EC-Earth and ERA-Interim model configurations in this section. Additionally, we discuss differences compared



to sensitivity simulations. Due to the common formation pathways of $SO_4^{2-}$ and OXL in the atmosphere, the $SO_4^{2-}$ budget
calculations are also presented and discussed. All calculations are presented as a mean (± standard error) for the years 2000-2014.

### 3.1.1 Oxalate

The annual net chemistry production of OXL (Table 2a) in EC-Earth is 12.61 ± 0.06 Tg yr$^{-1}$, which is lower than in ERA-Interim (18.12 ± 0.07 Tg yr$^{-1}$). The difference is explained by a higher oxidizing capacity in ERA-Interim than in EC-Earth.
ERA-Interim calculates higher OH concentrations in the tropical and subtropical troposphere (Fig. S1b). In contrast, zonal mean OH levels in EC-Earth are slightly higher in the extratropics, causing a more efficient oxidation of the OXL precursors, such as GLY (Fig. S1d), GLYAL (Fig. S1f), MGLY(Fig. S1h), and $CH_3COOH$ (Fig. S1j) at higher latitudes, especially in the SH. Note that, van Noije et al. (2014) also showed that in large parts of the troposphere, the simulated oxidizing capacity in the previous version of EC-Earth (EC-Earth v2.4) was lower compared to a respective ERA-Interim configuration, due to the
simulated lower temperatures (cold biases) and specific humidities. However, since SSTs and sea-ice concentrations are prescribed in our EC-Earth atmosphere-only simulations, the long-term means of tropospheric temperatures and water vapor are not expected to differ significantly to ERA-Interim close to the surface levels, as also indicated by the low differences in the OH levels of the two simulations at low altitudes (Fig. S1b).

The production term of OH in EC-Earth is ~5 % lower than in ERA-Interim, due to a lower amount of water vapor available
to react with $O(^1D)$. In addition, a ~6 % lower OH production through the $H_2O_2$ photolysis is simulated in EC-Earth. Note, that $H_2O_2$ is an important driver of the aqueous-phase oxidizing capacity in the model, with about 78-79 % of the OH radicals in the liquid phase being produced by photolysis of the dissolved $H_2O_2$. In more detail, the lower atmospheric abundance of the gas-phase $H_2O_2$ in EC-Earth (~ 11%) leads to smaller $H_2O_2$ uptake in the aqueous phase (~13%), and thus to a slower oxidation of OXL precursors due to the respective lower dissolved OH radical production (~19%). Overall, the total OH production is
~7 % lower in EC-Earth, which corresponds to a ~18% lower aqueous-phase OH production, resulting in a ~30 % lower OXL net chemistry production compared to ERA-Interim.

The total OXL production is 15.5 Tg yr$^{-1}$ in Lin et al. (2014) and 14.5 Tg yr$^{-1}$ in Liu et al. (2012), both lower compared to our ERA-Interim estimates (~20.8 Tg yr$^{-1}$; Table 2a), but close to EC-Earth (~15.0 Tg yr$^{-1}$; Table 2a). The main reason for the lower chemistry production of other published estimates compared to our results is the contribution of the aqueous-phase
glyoxal oxidation scheme proposed by Carlton et al. (2007) that is applied in our simulations. The oxidation of the glyoxal-derived high molecular weight products formed mainly in the cloud droplets is calculated to contribute significantly to the global OXL production in our model (Table 2a). This result is in line with Carlton et al. (2007), who indicated that the GLX-pathway may not be the primary pathway for oxalic acid formation, but instead the rapid oxidation of GLY-multifunctional products via the OH radicals (i.e., 3.1 10$^{10}$ L mol$^{-1}$ s$^{-1}$; Table S2). For ERA-Interim(sens) however, where no such reactions





are considered, the total OXL chemical production is calculated on average 11.5 Tg yr⁻¹ (Table 2); i.e., closer to the estimates of Lin et al. (2014) and Liu et al. (2012). On the other hand, our ERA-Interim net chemistry production calculations are close to the estimates of Myriokefalitakis et al. (2011) (i.e., ~21.2 Tg yr⁻¹) when no potential effects of the ionic strength (e.g., Herrmann, 2003) on OXL precursors are considered, although no Fe chemistry was considered that latter study. Indeed, the enhanced aqueous-phase oxidation capacity due to the Fenton reaction increases both the production and the destruction terms

of OXL in our model, leading to ~7% lower net OXL production and, respectively, a lower (~8%) atmospheric abundance. Nonetheless, our calculations indicate that Fe chemistry impacts on OXL net production drastically, increasing by at least ~50% the destruction of the dissolved oxalic acid. The potential primary sources (0.373 ± 0.005 Tg yr⁻¹) accounted for in the model (Table 2a) do not, however, significantly contribute to the simulated OXL atmospheric levels, and only a small fraction of OXL is calculated to be formed in aerosol water (~6 %) for all simulations in this work.

Focusing further on the atmospheric sinks of OXL, roughly ~13 % in ERA-Interim and ~16 % in EC-Earth of the produced oxalic acid is oxidized into $CO_2$ in the aqueous phase, mainly via the photolysis of the $[Fe(OXL)_2]^-$ complex (~55 %) and via OH radicals (~45 %). The fraction of the total produced OXL that is destroyed in the aqueous-phase is higher than in Liu et al. (2012) by ~7 %, where no Fe chemistry was considered, but lower compared to Lin et al. (2014) and Myriokefalitakis et al. (2011), where roughly 30 % of the produced OXL is oxidized into $CO_2$ in the aqueous phase. Finally, a total average

deposition rate of ~18.5 Tg yr⁻¹ is calculated in ERA-Interim, primarily due to wet scavenging (~99 %), resulting in a global atmospheric lifetime of ~5.7 days, close to Liu et al. (2012) and Lin et al. (2014), but higher compared to Myriokefalitakis et al. (2011) (~3 days) probably because the more intense OXL production at higher altitudes in our model.

The major pathways of global OXL production, both in ERA-Interim and EC-Earth, are the oxidation of glyoxal (~74 %), followed by glycolaldehyde (~11 %), methylglyoxal (~8 %), and acetic acid (~7 %). Glyoxylic acid is, nevertheless, an

important intermediate species because it is directly converted to OXL in the aqueous phase upon oxidation. Other important findings concerning the chemical budgets are summarized below:

1.    *Glyoxal*: About 70 Tg yr⁻¹ GLY is produced in the gas-phase in ERA-Interim, similar to Lin et al. (2014). EC-Earth calculates that is ~ 3 % lower. The global gas-phase production of the present work is higher than other global model estimates, e.g., ~56 Tg yr⁻¹ (Myriokefalitakis et al., 2008), ~40 Tg yr⁻¹ (Fu et al., 2009, 2008), and ~21 Tg yr⁻¹ (Liu et

al., 2012). This difference can be explained by the more comprehensive isoprene chemistry of the gas-phase scheme used here (Myriokefalitakis et al., 2020b). Indeed, isoprene secondary oxidation products (e.g., epoxides) are significant precursors of GLY in the atmosphere (Knote et al., 2014) and the contribution of isoprene epoxides (IEPOX) from the gas-phase isoprene oxidation is here considered as a pathway of GLY formation. Note that the oxidation of other biogenic hydrocarbons, like terpenes and other reactive organics, may also result in GLY formation,

since their chemistry is lumped on the first generation peroxy radicals of isoprene in the model (Myriokefalitakis et al., 2020b). Besides the biogenic hydrocarbon oxidation, the model considers GLY formation due to the oxidation of



other organic species (e.g., Warneck, 2003), such as acetylene (~4.8 Tg yr$^{-1}$) and aromatics (~18.8 Tg yr$^{-1}$). In the gas phase, other hydrocarbons, like ethene, further contribute to the atmospheric production of GLY via their oxidation products, mainly glycolaldehyde (~5.4 Tg yr$^{-1}$). However, as in many modeling studies, additional primary and/or secondary glyoxal sources might be still missing in our model. Indeed, the elevated glyoxal concentrations over oceans that have been observed from space (e.g., Wittrock et al., 2006) would require at least ~20 Tg yr$^{-1}$ of extra marine sources to reconcile model simulations with satellite retrievals (Myriokefalitakis et al., 2008). Great uncertainties, however, still exist on these oceanic sources (Alvarado et al., 2020; Sinreich et al., 2010) and therefore the only glyoxal primary sources accounted for in the model are from biofuel combustion and biomass burning processes (e.g., Christian et al., 2003; Fu et al., 2008; Hays et al., 2002), resulting overall in ~7 Tg yr$^{-1}$ on average in the model (Myriokefalitakis et al., 2020b). Glyoxal is rapidly destroyed in the atmosphere via photolysis (~70 %), followed by its oxidation in the gas phase (~15 %) and the aqueous phase (~15 %). Roughly 5.4 Tg yr$^{-1}$ of glyoxal are produced in the aqueous phase via the dissolved GLYAL oxidation in ERA-Interim, close to the Liu et al. (2012) calculations, but somehow higher compared to EC-Earth. Overall, the net cloud uptake of glyoxal in ERA-Interim is ~6.3 Tg yr$^{-1}$, which is higher than the estimates from Liu et al. (2012) (~1.6 Tg yr$^{-1}$). As expected, this increase is due to the applied glyoxal oxidation scheme in the aqueous phase of our base simulations. Finally, ~4.2 Tg yr$^{-1}$ of glyoxal are removed from the atmosphere via wet scavenging (~73 %) and dry deposition (~27 %).

2. *Glycolaldehyde*: GLYAL is also a significant species for OXL atmospheric abundance since its oxidation directly produces GLY both in the gas and the aqueous phase. In ERA-Interim, the gas-phase production is ~92.5 Tg yr$^{-1}$ on global scale, with the primary sources accounting for ~5.4 Tg yr$^{-1}$ (Myriokefalitakis et al., 2020b) on average. In EC-Earth, the gas-phase production is ~1 % lower. GLYAL is destroyed via gas-phase photolysis (~55 %) and by OH radicals in the gas phase (~35 %) and the aqueous phase (~10 %). Ethene oxidation products contribute ~39 % to GLYAL production, but isoprene chemistry dominates its chemical production in the model. The only source of GLYAL in the aqueous phase is nevertheless the transfer from the gas phase. The dissolved GLYAL is oxidized to produce GLY (~60 %) and GLX (~40 %), overall resulting in a net aqueous uptake of ~8 Tg yr$^{-1}$ in ERA-Interim, close to the estimates of Liu et al. (2012), but almost 40 % higher than in Lin et al. (2014). This higher uptake of GLYAL in the aqueous phase is due to the respective higher (~14 %) gas-phase production in our model. Note that in ERA-Interim, the net aqueous uptake of GLYAL is calculated ~24 % lower compared to ERA-Interim.

3. *Methylglyoxal:* The global annual mean gas-phase production of MGLY in ERA-Interim is ~237 Tg yr$^{-1}$ on average, with the primary sources accounting for ~4.6 Tg yr$^{-1}$. The gas-phase production is higher than the 160 - 169 Tg yr$^{-1}$ reported by other modeling studies (Fu et al., 2008; Lin et al., 2014; Liu et al., 2012) owing to the contribution of oxidation products considered in the gas-phase isoprene chemistry scheme (Myriokefalitakis et al., 2020b). Roughly 56 % of MGLY is produced via the gas-phase oxidation of HYAC with OH radicals, which is lower than the estimated



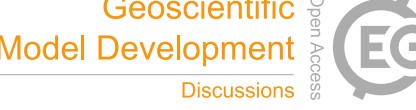

~75 % in Fu et al. (2008). The remaining MGLY production is due to isoprene oxidation products, i.e., ~10 % from
IEPOX oxidation, and ~7 % from methyl vinyl ketone (MVK) and methacrolein (MACR) oxidation. In the aqueous
phase, MGLY is produced via the dissolved HYAC oxidation (~13.0 Tg yr$^{-1}$) and then further oxidized by OH radicals
(~11.6 Tg yr$^{-1}$) into pyruvic acid (PRV), methylglyoxal oligomers (MGLYOLI), and to a lesser extent into GLX. Note
that the calculated contribution of dissolved HYAC to the aqueous-phase production of MGLY is higher compared
to the nearly negligible rates in Liu et al. (2012) because of the higher gas-phase production of HYAC in our model.
MGLY is chemically destroyed in the model mainly by gas-phase photolysis (~60 %), the OH radicals in the gas
phase (~35 %), and via oxidation in the aqueous phase (~5 %).

4. *Pyruvic and acetic acids:* The chemical production of PRV is ~14.7 Tg yr$^{-1}$ in ERA-Interim and ~16.7 Tg yr$^{-1}$ in EC-Earth. PRV is mainly produced by terpene oxidation via O$_3$ (~51 %) in the gas phase followed by methyl vinyl ketone (MVK) oxidation (~5 %). In the aqueous phase, PRV is solely produced from MGLY oxidation (~6.5 Tg yr$^{-1}$) and subsequently oxidized to CH$_3$COOH. PRV is mainly removed via photolysis in the gas phase and via oxidation by OH radicals in the aqueous phase (~30 %). However, more than half of the produced PRV in the aqueous phase directly contributes to the SOA mass of the model upon cloud evaporation. The gas-phase production of acetic acid is ~44.3 Tg yr$^{-1}$, with the primary sources accounting for approximately 24 Tg yr$^{-1}$. In the aqueous phase, roughly 3 Tg yr$^{-1}$ of CH$_3$COOH is produced via PRV oxidation. Note that the net uptake of CH$_3$COOH (~0.7 Tg yr$^{-1}$) is calculated in the model similar to the Lin et al. (2014) estimates, but smaller than the 6.7 Tg yr$^{-1}$ calculated by Liu et al. (2012).

5. *Glyoxylic acid:* The GLX production rate is 7.1 Tg yr$^{-1}$ in EC-Earth and it is ~30 % lower in ERA-Interim. About 55 % of the produced GLX is directly oxidized to oxalic acid in the aqueous phase and ~25 % is added directly to the SOA pool. Upon cloud evaporation, part of the produced GLX is also transferred in the gas phase, where it is either oxidized by OH radicals (~60 %), photolyzed (~33 %) or deposited (~7 %). Due to the destruction of GLX in the gas phase, its total production is lower (~60 %) compared to the production estimates in Lin et al. (2014) and Liu et al. (2012). For the EC-Earth and ERA-Interim, most of the produced GLX in the aqueous phase is derived from the oxidation of GLYAL (~48 %), followed by the oxidation of CH$_3$COOH (30 %), GLY and its oligomeric products (~ 15 %), and MGLY. The relative contributions in our calculations differ from the estimates in Lin et al. (2014), where GLX is primarily produced by GLY oxidation (~77 %) followed by GLYAL (~14 %), MGLY (~1 %), and acetic acid (~8 %). These differences are also caused by the direct contribution of the GLY oxidation products to the OXL formation. On the other hand, in the ERA-Interim(sens) simulation the calculated fractions agree well with other published estimates, where GLY overall dominates (~60 %) the GLX production in the aqueous phase.

Figure 2a presents the annual mean (average 2000-2014) net chemistry production rates of OXL in EC-Earth, and the respective
absolute differences compared to ERA-interim (Fig. 2c). The maximum OXL production rates are calculated around the tropics



and in the southern hemisphere, where both biogenic emissions (mainly isoprene) and the liquid cloud water are substantially enhanced (Fig. S2a). The Amazon region appears as the largest source of OXL, along with central Africa and Southeast Asia. At higher latitudes (> 45° N) the lower cloud liquid water content and vegetation cover lead to a lower OXL production over Asia and North America. However, over highly populated regions in the northern hemisphere, such as in Europe, the US, and

China, enhanced OXL production rates are calculated due to its anthropogenic precursors. Furthermore, a significant source of OXL is calculated downwind land areas, such as the South Pacific, and the tropical Atlantic Ocean, due to the long-range transport of OXL precursors.

The illustrated differences in OXL production between ERA-interim and EC-Earth (Fig. S2c) are caused due to the adopted atmospheric dynamics (i.e., online calculated versus offline), as both simulations use identical prescribed anthropogenic and

biogenic emissions. For instance, EC-Earth calculates higher cloud water concentrations at ~800-600 hPa around the tropics (30° S - 30° N) compared to ERA-Interim. In contrast, lower concentrations are derived aloft, with ERA-Interim presenting enhanced cloud water concentrations at ~400 hPa (Fig. S2b). Moreover, due to the lower OH concentrations in the tropical and subtropical troposphere (Fig. S1b), EC-Earth gives lower OXL production rates, especially over intense biogenic emission areas. Overall, the difference in the oxidizing capacity of the atmosphere between the two configurations significantly impacts

the aqueous phase OXL production efficiency in the model.

### 3.1.2 Sulfate

Sulfate ($SO_4^{2-}$) is the main inorganic aerosol species produced in the aqueous phase, and similar to OXL its production in the model mainly occurs in cloud droplets. In addition, these two species largely reside in the aerosol accumulation mode of the model (roughly 99 % for $SO_4^{-2}$ and ~97 % for OXL). $SO_4^{2-}$ is a key species for determining atmospheric acidity, and therefore

we also present here the sulfate budget in conjunction with that of OXL. Sulfate is produced both in cloud droplets and in aerosol water, with the production in aerosol water having however a negligible contribution on a global scale. In contrast to OXL, for which no gas-phase production is considered, the gas-phase oxidation of $SO_2$ via OH radicals contributes to the total $SO_4^{2-}$ concentrations with about 12.0 Tg S yr$^{-1}$ (Table 2b). Our global estimate of the gaseous sulfuric acid ($H_2SO_4$) production is higher than in EC-Earth v2.4 (~7.8 Tg S yr$^{-1}$ averaged for the years 2000–2009; van Noije et al., 2014), but slightly lower

than in the EC-Earth3-AerChem AMIP simulations (van Noije et al., 2021) used for the CMIP6 experiments (available in https://esg-dn1.nsc.liu.se/search/cmip6-liu/; last access 11/06/2021) where ~12.9 Tg S yr$^{-1}$ of $H_2SO_4$ are produced (averaged for the years 2000-2014). These differences can be directly attributed to the OH radical production rates in the gas phase between the new and the previous versions of the atmospheric model, as have been discussed in Myriokefalitakis et al. ( 2020b). Despite the generally lower gas-phase OH radical levels (Fig. S1b), the slightly higher (~8 %) global $H_2SO_4$ gas-phase

production rate in EC-Earth than in ERA-Interim (Table 2b), can be attributed to the higher (~6%) DMS emissions in EC-Earth (Fig. S4b), that contribute to the atmospheric $SO_2$ levels over the ocean.





The aqueous-phase $SO_4^{2-}$ chemistry production from the oxidation of dissolved $SO_2$ is 39.8 Tg S yr$^{-1}$ in EC-Earth (Table 2b), which is higher than in EC-Earth v2.4 (29.3 Tg S yr$^{-1}$; van Noije et al., 2014) and EC-Earth3-AerChem (32.5 Tg S yr$^{-1}$). The higher $SO_4^{2-}$ chemical production is mainly due to the higher $SO_2$ aqueous-phase oxidation rates by $H_2O_2$. In more detail, our

calculations show that ~84 % of the global $SO_4^{2-}$ production in EC-Earth is due to the dissolved $SO_2$ oxidation via $H_2O_2$; ~33.3 Tg S yr$^{-1}$ is produced due to $H_2O_2$, higher compared to 23.9 Tg S yr$^{-1}$ in van Noije et al. (2014). The dissolved $SO_2$ oxidation via $O_3$ (6.4 Tg S yr$^{-1}$) is also higher than in EC-Earth v2.4 (5.4 Tg S yr$^{-1}$). The contribution of $CH_3O_2H$ to the $SO_4^{2-}$ aqueous-phase production is however small (~0.05 Tg S yr$^{-1}$) in the model, with the $HO_2$ contribution being practically negligible on the global scale (~0.02 Tg S yr$^{-1}$) for all simulations performed in this study. A total annual mean deposition rate of ~53 Tg S

yr$^{-1}$ is simulated in EC-Earth, with wet scavenging dominating the total deposition rate (~93%). Note also that 2.5 % of the sulfur in the $SO_2$ emissions (~1.6 Tg S yr$^{-1}$) is assumed to be in the form of $SO_4^{2-}$ for all simulations, which accounts for its formation in the sub-grid plumes (Aan de Brugh et al., 2011; Huijnen et al., 2010). Overall, a global $SO_4^{2-}$ lifetime over deposition of ~4.8 days is calculated in EC-Earth, which is lower than in ERA-Interim (~6.6 days), but similar to the EC-Earth v2.4 estimate (~4.9 days).

Figure 2b also shows the annual mean $SO_4^{2-}$ net chemistry production rates in EC-Earth. High $SO_4^{2-}$ production rates are calculated downwind of major anthropogenic $SO_2$ emission hotspots, such as Central Europe, Eastern US, India, Russia, and Eastern Asia. Furthermore, relatively high production rates due to biomass burning and volcanic eruptions are calculated in South America, Southern Africa, and Indonesia. Significant $SO_4^{2-}$ production is calculated over almost all oceanic regions due to the $SO_2$ production via the gas-phase oxidation of marine DMS emissions (Fig. S4a). Compared to the ERA-Interim

simulation, however, the $SO_4^{2-}$ production rates in EC-Earth are on average slightly higher over land in the tropics and extratropics. This increase can be attributed to combined effects that result in differences in chemical production and deposition rates (Table 2b). Some differences over oceans are nevertheless expected due to the differences in DMS concentrations since DMS emissions are online calculated in the model based on sea surface temperature and wind velocity (Fig. S4b).

### 3.1.3 Iron

In EC-Earth, the total Fe (TFe) soil emissions result in 59.3 ± 1.2 Tg Fe yr$^{-1}$, while in ERA-Interim they are 49.0 ± 1.0 Tg Fe yr$^{-1}$. This difference results from the differences in wind speed between EC-Earth and ERA-Interim. EC-Earth produces higher dust emissions over large parts of the Middle East and Asia compared to ERA-Interim (Fig. S4f), which explains the differences in TFe emissions (TFeC emissions do not differ). However, most of the dissolved Fe from mineral dust in the model originates from atmospheric dissolution processes. In EC-Earth, FeD is primarily dissolved due to aerosol acidity at ~0.31 Tg Fe yr$^{-1}$,

followed by the ligand-promoted dissolution that additionally produces ~0.17 Tg Fe yr$^{-1}$, while the photoinduced processes have a small impact on the global dissolved Fe release from dust, with ~0.05 Tg Fe yr$^{-1}$ (Table 2d). Fe primarily resides (~98.4 %) in the slow pool of Fe-containing dust aerosols in the model, in particular in the coarse mode, with about 1.0% being emitted as nano-sized iron oxides (intermediate pool), and ~0.5% as ferrihydrite (fast pool). Thus, most of the dissolved Fe





release originates from the heterogeneous inclusion of nano-Fe grains in the internal mixture of various Fe-containing minerals
such as aluminosilicates, hematite, and goethite (~66 %), followed by nano-sized iron oxides (~24 %) and to a lesser extent by
ferrihydrite (~10 %). Note, however, that the Fe release from aluminosilicates, hematite, and goethite particles is a slower
process compared to the other soil classes considered in the model, as dictated by the three-stage approach applied for this
study (Table S4).

Fe emissions from combustion processes, such as biomass burning, coal, and ship oil combustion, are estimated at $2.52 \pm 0.10$
Tg Fe yr$^{-1}$ in both simulations, with 0.012 Tg Fe yr$^{-1}$ being emitted as dissolved from the primary oil combustion processes.
Roughly $0.274 \pm 0.01$ Tg Fe yr$^{-1}$ are released through Fe dissolution from combustion aerosols in EC-Earth, in good agreement
with ERA-Interim ($0.285 \pm 0.01$ Tg Fe yr$^{-1}$). The acid-promoted dissolution contributes ~17 % and photo-reductive processes
~16 % to the Fe release from combustion particles, thus most Fe release comes from the ligand-promoted dissolution. This
result is in line with laboratory studies (e.g., Chen and Grassian, 2013), where the contribution of oxalate-promoted dissolution
is several times larger than the proton-promoted pathway under highly acidic dark conditions. According to our calculations,
the relative contribution of atmospheric processing to the combustion aerosol Fe solubilization (~11 %) is significantly higher
compared to that of crystalline dust minerals (~1 %), in agreement with laboratory (e.g., Chen et al., 2012; Fu et al., 2012) and
modeling (e.g., Ito, 2015; Ito and Shi, 2016) studies. All in all, the total DFe atmospheric source in EC-Earth, accounting for
both primary emissions and atmospheric processing, is $0.88 \pm 0.04$ Tg-Fe yr$^{-1}$ for the present day, well in the range of estimates
presented in the model intercomparison study ($0.7 \pm 0.3$ Tg-Fe yr$^{-1}$) in Myriokefalitakis et al. (2018).

The annual mean dissolution rates of FeC and FeD in EC-Earth are presented in Fig. 3. For combustion aerosols, the maximum
dissolution rates occur downwind of biomass burning sources and highly populated regions, such as South America and Central
Africa, the Middle East, India, and China. High dissolution rates are more likely to coincide with high OXL concentrations
(Ito, 2015). Indeed, the model calculates important dissolution rates near regions where the OXL production rates are enhanced
(Fig. 2a), such as over the Amazon basin and Central Africa, as well as downwind of these regions as the combustion aerosols
are transported to the open ocean, in agreement with observations (e.g., Sholkovitz et al., 2012). For the mineral dust aerosols,
most of the FeD dissolution fluxes occur downwind of the major dust source regions (e.g., the Sahara and the Gobi Desert),
where the atmospheric transport of anthropogenic pollutants, such as SOx and NOx, enhances atmospheric acidity; e.g., the
Fe release from the dust minerals due to proton-promoted dissolution processes is enhanced over the Middle East. Significant
dissolution rates are also simulated over the Atlantic Ocean at the outflow of the Sahara, as well as at the outflow of Asian
desert regions to the Pacific Ocean. High rates due to the contribution of the organic ligand-promoted dissolution processes
are calculated downwind of central Africa and the equatorial Atlantic Ocean, where the oxidation of biogenic hydrocarbons in
the presence of cloudiness leads to enhanced OXL aqueous-phase formation rates. On the contrary, the efficiency of ligand-
promoted dissolution is substantially suppressed near dust source regions due to the low OXL availability (Fig. 2a).





Figure 3 also presents the absolute differences between the ERA-Interim and EC-Earth annual mean Fe dissolution rates. ERA-Interim has significantly lower dissolution rates in the tropics (e.g., Central Africa) and around the equator, both for FeC (Fig. 3c) and FeD (Fig. 3d). This decrease is attributed both to the differences in atmospheric dynamics between the two model configurations, as well as to the suppression of the organic-dissolution processes with lower OXL production. Indeed, Fig. 2c shows that in ERA-Interim, OXL production rates increase in the tropics, impacting Fe-dissolution rates. In contrast, FeC

dissolution rates increase in ERA-Interim over the Arabian Peninsula, India, and Eastern Asia, due to fluctuations in OXL production and aerosol acidity. EC-Earth also shows lower FeD dissolution rates over the Northern Pacific in the outflow of Asia. These differences are due to a higher to aerosol acidity (i.e., up to ~1 pH unit; Fig. S3d,f) in ERA-Interim due to changes in the buffering capacity of dust promoted by the higher calcite emissions in EC-Earth (Fig. S4f). This is the case especially for coarse dust aerosols where the majority of the Fe resides. EC-Earth also shows differences (positive or negative) with ERA-

Interim in the Fe dissolution rates over oceanic regions (Fig. 3e,d), likely due to differences in $SO_4^{2-}$ production over oceans from marine DMS emissions (Fig. S4b), and to the impact of sea salt emissions (Fig. S4d) upon the buffering capacity of the solution

### 3.2 Evaluation of new model features against observations

All developments described in this work have been implemented over the EC-Earth3-AerChem model version, which has

proved to simulate the atmospheric aerosol cycles comparably well to other global models and to behave outstandingly in the simulation of optical properties (Gliß et al., 2021). Thus, we do not expect substantial changes in EC-Earth3-Iron ability to represent the aerosol cycle, nor their optical properties, compared to EC-Earth3-AerChem. However, owing to the significant differences in the gas-phase and aqueous chemistry between versions, we provide an overall assessment of the Aerosol Optical Depth (AOD). In addition, as one of the novelties of this work is to consider explicitly how dust composition affects the

atmospheric iron burden and alters acidity (e.g., through calcite), and therefore a comparison of dust fields with in-situ observations is also provided. Finally, simulations of specific species key to our developments, such as oxalate, sulfate, and total and soluble iron are also evaluated.

### 3.2.1 AOD, dust concentration and deposition

The annual mean AOD at 550 nm modeled in EC-Earth for 2000-2014 compares favorably with AERONETv3 direct-sun level

2.0 data (Fig. 4a). Overall, the model presents an nMB of -9.1% and an nRMSE of 45.7%, considering information from 738 AERONET sites. The regional analysis suggests a slightly better behavior in northern hemisphere regions (e.g., North America, Europe, East Asia) dominated by anthropogenic aerosols (normalized errors and biases, below 45% and ±10%, respectively). The largest deviations from observed AOD occur in the Southern Hemisphere (e.g., South Africa, Australia, and Oceania) or remote regions. Over dust-dominated regions (e.g., North Africa, West Asia, and the Middle East) the model also behaves well

(with normalized errors and biases below 45% and ±10%, respectively). Selecting specifically dust-dominated sites for the





comparison (Fig. 4b) and following the criteria explained in Sect. 2.5, EC-Earth slightly overestimates the retrieved AOD at 550 nm over North Africa (nMB=21.5%) and shows underestimations over sources in West Asia and the Middle East, as well as in transport regions such as Central America. In general, EC-Earth's ability to reproduce the annual mean AOD at 550 nm holds for dusty sites, with a normalized mean bias of 2.7%, and a normalized root mean square error of 37.2% over 38 sites.

The comparison of model outputs with climatologies of dust surface concentration from the RSMA and the AMMA campaign (Fig. 4c) yields slightly poorer results, with an nMB of 19.3% and an nRMSE of 81.2%, as an average of the 23 sites available. EC-Earth best reproduces dust surface concentrations over source regions, such as North Africa (nMB=21.9%, nRMSE=37.7%), shows underestimations in transport areas (e.g., Central America: nMB=-37.5%, nRMSE=38.7%) and poorly represents the surface concentration in remote regions (e.g., South Pacific and Southern Ocean, with nMB up to -98.5% and

nRMSE up to 112.7%). This issue is also reflected in the evaluation of the dust deposition field (Fig. 4d), with positive and negative biases over source regions but a clear underestimation of the deposited mass on transport and remote regions (e.g., the Southern Ocean). These discrepancies point towards an overestimation of dust deposition. For instance, EC-Earth3-Iron may share the difficulties of many global dust models in representing the long-range transport of dust, in particular coarse dust downwind of dust sources (e.g., Adebiyi and Kok, 2020). As minerals in dust constitute the primary source of TFe to the

atmosphere, the overestimation close to the source regions and the underestimation in remote regions will also influence the representation of dust-related DFe in the model.

### 3.2.2 Oxalate

The averaged OXL surface concentrations in EC-Earth for the boreal winter (December, January, and February; DJF) and summer (June, July, and August; JJA) are presented in Fig. 5. OXL surface concentrations are distributed roughly between 60º

S and 60º N, mainly in regions where intensive VOC emissions from anthropogenic and biogenic sources coexist with cloud water. The highest OXL concentrations are calculated over tropical Africa, the Amazon Basin, eastern Asia, the eastern United States, and Europe, clearly showing the strong impact of OXL precursors (e.g., glyoxal) and the availability of cloud water, along with increased biogenic emissions at these latitudes. In the northern hemisphere, OXL concentrations are generally calculated higher in summer and lower in winter, indicating a strong impact of temperature and photochemistry on the

production rate of oxalic acid in the aqueous phase. During DJF, the model calculates lower OXL concentrations over mid- and high-latitude regions, such as East Asia, Central Europe, and Northern US. Over these highly populated regions, the aerosol water content is enhanced, following the increased $SO_4^{-2}$ production due to anthropogenic activities, and the aqueous-phase OXL production in deliquesce particles also contributes to OXL atmospheric concentrations. Furthermore, high OXL concentrations are calculated in the tropics for both seasons, due to the photochemical activity and the intense sources of

biogenic VOCs in these regions.





Figure 5 further presents the differences between EC-Earth and ERA-Interim. In general, OH levels in ERA-Interim are higher, which causes a more efficient oxidation of OXL precursors for both seasons. Moreover, ERA-Interim shows higher concentrations around the intertropical convergence zone (ITCZ) due to differences in meteorology between the two simulations, as discussed above. During boreal winter, some differences are observed over the subtropics of the Northern

Hemisphere. Although OXL concentrations are very low over these latitudes, the relatively strong increase in liquid water that serves as a medium for OXL production, both for clouds (Fig. S2b) in higher altitudes (~400 hPa), and at the surface for deliquescent particles (Fig. S2d,e) in the ERA-Interim simulation. In the vertical, OXL concentrations are distributed in the model from the surface to ~400 hPa with a maximum at around 900 hPa. The zonal mean differences, however, indicate strong increases in the southern hemisphere (30º S - 0º) during boreal winter (Fig. 5e). Compared to EC-Earth, ERA-Interim calculates

higher OXL concentrations in the upper troposphere for both seasons (Fig. 5e,f), mostly due to more efficient transport of OXL precursors by deep convection into the tropical and extratropical upper troposphere. In the lower and middle troposphere, higher concentrations are calculated in ERA-Interim depending on the location and the season. The concentrations in EC-Earth are also lower than in ERA-Interim in the NH extratropics during boreal summer due to a lower chemical production (Fig. 5f).

Figure 6 presents the comparison of the different model simulations performed for this work with OXL surface observations.

OXL concentrations show a strong seasonal dependence, with maxima during the warm season due to the intense photochemical activity combined with the higher precursor's abundance. Over the Mediterranean, and specifically the eastern part which is characterized by the long-range transport of air pollution and from surrounding urban centers (Kanakidou et al., 2011), the model underestimates the observed concentrations during winter at the Finokalia station in all simulations (Fig. 6a), either due to missing OXL primary and secondary sources, or to a too strong removal. During summer, ERA-Interim

satisfactorily simulates the observed OXL levels, also representing the observed trend, which indicates that the model reproduces the mixing and aging of the air masses in the region under favorable meteorological conditions and intense solar radiation. EC-Earth calculates lower OXL concentrations than ERA-Interim, due to the lower oxidizing capacity, underestimating thus the observed concentrations for all seasons. On the other hand, ERA-Interim(sens) tends to underestimate the observations for all seasons more than the other simulations, further indicating the important role of the secondary sources

to OXL atmospheric concentrations in the region. At the Puy de Dome site (Fig. 6c), which is located at 1450m asl, ERA-Interim underestimates the observed OXL concentrations, although simulates them more realistically compared to EC-Earth, especially during summer (Fig. 2c). The seasonal variation in the area can be explained by the stronger upward transport of air masses during summer (Legrand et al., 2007), increasing thus the OXL production in the region. However, the model fails to represent the observed OXL levels, possibly due to missing sources. The importance of other production pathways, not related

to the aqueous-phase GLX oxidation, is demonstrated in the comparison of the observed OXL levels with the ERA-Interim(sens) simulation. Again, ERA-Interim(sens) deviates more from measured values than the other simulations. Nevertheless, this indicates that other species may further contribute to OXL production, such as the decay of longer diacids (e.g., azelaic, and malonic acids) (Legrand et al., 2007) that are currently not included in the model. Another reason may be





the impact of the enhanced cloud LWC in the region, implying a more intense cloud processing compared to other surface
sites, and thus, faster oxidation of oxalic acid into $CO_2$ (Ervens et al., 2004) in the model. Finally, at the Azores (Fig. 6e), a
site that is characterized by a marine environment, the model tends to underestimate the observed OXL concentrations most
of the time, with ERA-Interim presenting again a better skill than other simulations. EC-Earth underestimates the observed
concentrations more than ERA-Interim, especially during summertime, and ERA-Interim(sens) simulates the lowest OXL
concentrations. The observed OXL levels in the region, however, can be explained either by to the transport of pollutants from
the continents or the photochemical production in the region. Thus, the illustrated differences against the observations between
EC-Earth and ERA-Interim can be attributed to differences in the oxidizing capacity and in simulated transport of EC-Earth,
such as the vertical mixing in the troposphere (e.g., van Noije et al., 2014), that has a further impact on OXL precursors like
glyoxal. Furthermore, since the long-range transport is found relatively constant in summer and winter in the region (Legrand
et al., 2007), other species of marine origin, such as the unsaturated fatty acids (e.g., linoleic and oleic acids) may also
contribute as precursors to the OXL production, especially during summer, but are not included in the model too. We
acknowledge that other formation pathways of OXL, primary or secondary, may exist in the atmosphere (e.g., Baboukas et al.,
2000). For example, higher DCAs (such as malonic, succinic, glutaric, and adipic acids) may act as precursors for smaller
dicarboxylic acids like OXL, both in the gas- (e.g., Kawamura and Ikushima, 1993) and the aqueous phase (e.g., Ervens et al.,
2004; Lim et al., 2005; Sorooshian et al., 2006) and could further contribute to atmospheric OXL concentrations.

In Fig. 6g, OXL observations reported in the literature are compared with monthly mean simulations. Due to the relatively low
resolution of the global model (i.e., 3º x 2º in longitude by latitude), the spatial variability of urban emissions cannot be not
expected to be resolved. Therefore, urban stations are omitted and the comparison is limited to locations representative of
background concentrations. All simulations tend to underestimate OXL observations, with lower biases in ERA-Interim (i.e.,
nMB = ~-46 %, nRSME = ~117 %). As expected, for almost all sites the ERA-Interim simulation calculates the highest OXL
concentrations and the ERA-Interim(sens) the lowest (i.e., nMB = ~-74 %, nRSME = ~125 %). The latter indicates that
additional production pathways need to be considered in modeling studies to capture the observed OXL concentrations. EC-
Earth underestimates the observed concentrations more than ERA-Interim (i.e., nMB = ~-64 %, nRSME = ~110 %) but less
than ERA-Interim(sens), highlighting the importance of the atmospheric oxidating capacity and atmospheric dynamics in the
OXL production. Overall, our analysis indicates that the model either misses OXL sources (primary and secondary) or
overestimates OXL sinks especially during winter. Thus, under relatively lower temperatures and irradiation, the model may
neither represent well the fast secondary OXL production in wood-burning plumes nor the secondary production through
species produced by the oxidation of emitted from vehicles and other anthropogenic activities, such as ethane and aromatic
hydrocarbons.



### 3.2.3 Sulfate

The averaged $SO_4^{2-}$ surface concentrations as calculated in EC-Earth for the boreal winter and summer are presented in Fig. 7. During both seasons, high $SO_4^{2-}$ concentrations are simulated near or downwind major anthropogenic emission hotspots, where the vast majority of the surface $SO_2$ emissions from anthropogenic origin occur (e.g., Tsai et al., 2010). Enhanced $SO_4^{2-}$ surface concentrations are also calculated downwind of biomass burning and volcanic eruptions, showing overall the impact of $SO_2$ primary sources and the abundance of cloud water over these latitudes. Over the remote oceans, however, DMS oxidation may

significantly contribute to the $SO_4^{2-}$ surface concentrations, as also the sulfur emissions over major shipping routes. The differences of between EC-Earth and EC-Earth-AerChem in the averaged $SO_4^{2-}$ surface and zonal mean concentrations are also presented, both for boreal winter (Fig. 7c,e) and summer (Fig. 7d,f). Considering that the EC-Earth version developed for this work is based on the EC-Earth-AerChem model version, the illustrated differences are solely due to the applied chemistry schemes in the model. During boreal winter, the largest differences appear over Eastern Asia, the Middle East, India, and

Central Europe. In EC-Earth-AerChem, the gas-phase $SO_2$ oxidation by OH radicals is roughly 8% larger than in EC-Earth (see Sect. 3.1.2), leading overall to a lower conversion of $SO_2$ to sulfate in the aqueous phase. Moreover, in our simulations $SO_4^{2-}$ is also produced in deliquesced particles, which partly contribute to $SO_4^{2-}$ atmospheric concentrations, especially over highly populated regions during the boreal winter when sulfur emissions due to anthropogenic activities are enhanced. On the contrary, during boreal summer some differences are illustrated (Fig. 7d) mostly over the Middle East where the EC-Earth-

AerChem simulation results in lower $SO_4^{2-}$ concentrations, and over Eastern Asia, where some higher concentrations in EC-Earth-AerChem occur. These negative/positive differences can be attributed to differences in oxidizing capacity between the two models, both in the gas and the aqueous phase. Finally, the zonal mean differences indicate higher concentrations in EC-Earth in the lower troposphere of the northern hemisphere during boreal winter (Fig.7e), as for the $SO_4^{2-}$ surface concentrations. Again, for boreal summer, no important differences are presented (Fig.7f).

Figure 6 further presents the model comparison with $SO_4^{2-}$ surface observations. Sulfate concentrations maximize during intense photochemical activity and high $SO_2$ atmospheric levels, generally suggesting a faster formation rate compared to oxalic acid (Ervens et al., 2004; Legrand et al., 2007). At the Finokalia station in the Mediterranean (Fig. 6b), the model overestimates the observed $SO_4^{2-}$ concentrations during boreal winter and summer in ERA-Interim, probably due to too high $SO_2$ background concentrations owing to the long-range transport from surrounding regions. During winter, EC-Earth better

reproduces the observations, probably due to the lower oxidizing capacity compared to ERA-Interim, but in late spring and early summer, the active photochemistry in the region leads to an overestimation of the observed concentrations. The EC-Earth-AerChem simulation leads to generally lower concentrations compared to our EC-Earth simulation, tending to somehow underestimate the observed concentrations, except for autumn. At Puy de Dome (Fig. 6d), ERA-Interim, which has a higher cloud liquid water content aloft and a more intense oxidizing capacity compared to EC-Earth, overestimates the observed $SO_4^{2-}$

concentrations in almost in all seasons. In contrast, EC-Earth better simulates the $SO_4^{2-}$ concentrations, although it seems to





underestimate the observations during summer. At that site, the EC-Earth-AerChem calculations agree well with EC-Earth, although again concentrations are slightly lower. At the Azores (Fig. 6f), ERA-Interim also simulates the observed concentrations well, following the observed annual cycle. In contrast, both EC-Earth and EC-Earth-AerChem underestimate the $SO_4^{2-}$ observations, especially during spring and summer. Note that the $SO_4^{2-}$ production in this marine site is attributed to

the $SO_2$ atmospheric levels both from air masses advected from industrialized regions and the local production due to the oxidation of marine DMS emissions. Thus, the differences between ERA-Interim and the other EC-Earth simulations presented in this study may indicate slower aging of the polluted air masses transported in the region. Finally, a comparison of the model's monthly mean predictions with a compilation of $SO_4^{2-}$ observations (n=3828) around the globe (Daskalakis et al., 2016) is presented in Fig. 6h. EC-Earth tends to overestimate the available $SO_4^{2-}$ observations, presenting positive biases (i.e.,

nMB = ~15.9 %, nRSME = ~5 %), that are slightly lower than in ERA-Interim (i.e., nMB = ~23 %, nRSME = ~57 %). In contrast, EC-Earth-AerChem tends to slightly underestimate the observed concentrations (i.e., nMB = ~-4.7 %, nRSME = ~57 %), showing a slightly lower correlation coefficient (R = 0.70) than EC-Earth (i.e., R = ~0.76) and ERA-Interim (i.e., R = ~0.75).

### 3.2.4 Dissolved iron

Figures 8 and 9 present the averaged dissolved FeC (DFeC) and FeD (DFeD) surface concentrations, respectively, for DJF and JJA. EC-Earth calculates an annual global DFeC atmospheric burden of ~0.002 Tg, while ERA-Interim calculates slightly higher concentrations (~0.003 Tg) due to the more intense ligand promoted dissolution rates (Table 2c). Elevated DFeC concentrations during boreal winter (Fig. 8a) are calculated over central Africa, eastern Asia, and India, where significant DFe concentrations (~0.01-0.1 µg Fe m$^{-3}$) are associated with biomass burning and anthropogenic combustion emissions. During

boreal summer (Fig. 8b), the maximum DFeC concentrations are calculated in the Northern latitudes, in particular over the Mediterranean Basin, the Middle East, the Western US, and China. The increase in the surface dissolved Fe concentrations over these regions, ranging from ~0.01-0.1 µg Fe m$^{-3}$, clearly highlights the anthropogenic contribution due to the enhanced solubilization of Fe when the combustion aerosols are mixed with acidic and organic pollutants during atmospheric transport (Fig. 3b). Due to the intense biomass burning in the Southern Hemisphere (i.e., South America, Central Africa, and Indonesia),

the enhanced OXL production rates over such regions (Fig. 2a) lead to higher dissolved Fe concentrations. Figures 8a and 8b demonstrate, overall, that the geographic pattern of the DFeC concentrations may change from boreal winter to summer, following the biomass burning activity and the atmospheric processing of Fe-containing combustion aerosols of anthropogenic origin.

Mineral dust emissions mostly occur in the mid-latitudes of the Northern Hemisphere (Figs. 9a, b), relatively close to where

the vast majority of the population exists and the anthropogenic emissions of acidic compounds dominate. For both seasons, high DFe concentrations from dust occur over the midlatitudes of the Northern Hemisphere, where the major dust sources are located. However, the equatorial maximum during boreal winter tends to shift to the North during boreal summer following





the migration of the ITCZ (Fig. 9b). DFe from mineral dust aerosols maximize over the major dust source regions, with surface concentrations of roughly 0.1-1 μg Fe m$^{-3}$ (Fig. 9a, b), overall dominating the Fe burden. The outflow from these regions

transports DFe over the global oceans, where secondary maxima of ~0.01-0.1 μg Fe m$^{-3}$ are calculated, mainly over the Northern Hemisphere in the tropical Atlantic Ocean. The dissolved Fe associated with Saharan dust is, nevertheless, attributed to the long-range transport and the atmospheric processing that converts the insoluble Fe minerals to soluble forms.

The differences between the EC-Earth and ERA-Interim are illustrated in the averaged DFeC (Fig. 8c,d) and DFeD (Fig. 9c,d) surface concentrations. The differences in DFeC between the EC-Earth and ERA-Interim are well correlated with those of

OXL concentrations (Fig. 5 c,d), indicating the strong impact of ligand-promoted dissolution on the DFeC atmospheric load (Table 2c). Note, however, that the most important relative differences between the two simulations are calculated over regions with low dissolved FeC concentrations, and consequently the total burden does not change significantly (Table 2c). The differences in the DFe concentrations associated with mineral dust aerosols follow the general anomaly pattern of the two model configurations due to differences in transport, leading overall to higher (~15 % globally) dust emissions in EC-Earth

(Fig. S4f).

Figure 10 presents a comparison of the different model simulations with cruise observations of dissolved Fe concentrations. The spatial distributions of the DFe observations for the accumulation, the coarse, and the total suspended aerosols are shown in Fig. 10a-c, respectively. The DFe concentration (mean ± standard deviation) in the accumulation mode amounts to 3.7 ± 6.4 ng m$^{-3}$ in the observations, while in EC-Earth and ERA-Interim is 5.8 ± 5.2 ng m$^{-3}$ and 5.7 ± 6.5 ng m$^{-3}$, respectively. The

observed concentration of DFe in the coarse mode is 5.0 ± 10.9 ng m$^{-3}$, while in EC-Earth and ERA-Interim the calculated values are around 6.2 ± 7.3 ng m$^{-3}$ and 5.7 ± 10.6 ng m$^{-3}$. Finally, the concentration of DFe in total suspended particles (tsp) is 7.8 ± 19.0 ng m$^{-3}$ in the observations, as well as 12.5 ± 19.1 ng m$^{-3}$ and 11.8 ± 20.4 ng m$^{-3}$ in EC-Earth and ERA-Interim, respectively. The correlation coefficients between the DFe cruise observations and the model results in EC-Earth and ERA-Interim are calculated as ~0.51 (nMB = ~58 %, nRMSE = ~170 %) and 0.59 (nMB = ~56 %, nRMSE = ~169 %) for the

accumulation aerosols, while for the coarse mode the values are calculated around 0.49 (nMB = ~23 %, nRMSE = ~194 %) and ~0.67 (nMB = ~13 %, nRMSE = ~174 %) for EC-Earth and ERA-Interim (Fig. S5).

The spatial distributions of the absolute differences between the simulated DFe concentrations in EC-Earth and the observations are also presented in Fig.10 d-e. The model shows a general overestimation of the observed DFe concentrations around the tropics (up to ~10 ng m$^{-3}$), but an underestimation at the mid-to-higher latitudes (up to ~5 ng m$^{-3}$). It is, however,

unclear why the model is unable to capture the observed concentrations over such regions. Reasons could be the misrepresentation of dust coarse emissions, missing anthropogenic primary Fe emissions, weak secondary sources of the dissolved Fe (especially in the southern latitudes), or even systematic errors in the transport of coarse particles. Over the Pacific, EC-Earth better predicts the average DFe concentrations. For completeness, Fig. S5 also presents a comparison of the simulated and observed TFe concentrations, showing that the model better captures the TFe concentrations in the accumulation



mode (Fig. S6g) than in coarse mode (Fig. S6h). Considering, however, that the TFe is mostly dominated by primary sources, the calculated differences to the observed concentrations (Fig. S6e-f) downwind continental sources should mainly depict errors in the emission parameterizations, or a misrepresentation in the mineralogical composition of the larger Fe-containing soil particles.

The overestimation around the tropics and the northern latitudes is further illustrated by a comparison of the model predictions
with observations as a function of latitude (binned at 2°) (Fig. 10g-i). Although the dissolved Fe in the accumulation mode (Fig.10g) is well simulated over the Southern Ocean, the model strongly overestimates the observed concentrations in the tropics especially around the Equator, as well as in the northern extratropics. In contrast, for the coarse and the total suspended aerosols, an underestimation of the DFe aerosol observations in the southern latitudes (i.e., around 30° - 60° S) is clearly visible for both simulations (Fig. 10h and Fig. 10i), along with an overestimation around the tropics, as for accumulation DFe aerosols.
All in all, the differences between the two model configurations are relatively small.

## 4 Discussion

The ocean is a critical component of the Earth's climate system and Fe plays a key role in the efficiency of the biological carbon pump. For this reason, accurate estimates of the bioavailable Fe inputs to the ocean are a prerequisite for climate simulations. Our work attempts at properly simulating the effects of atmospheric multiphase processes on the chemical sources
and sinks of the Fe-containing aerosols in an Earth System Model. Indeed, the atmospheric processing of Fe-containing aerosols is rather important for the geographical pattern of DFe deposition fluxes into the ocean, especially in remote regions away from land sources. In agreement with other studies (e.g., Ito et al., 2019; Mahowald et al., 2005; Myriokefalitakis et al., 2018; Scanza et al., 2018), we find that mineral dust is the principal source of atmospheric Fe in EC-Earth (95 ± 1 %), with most of the remaining sources attributable to biomass burning. Focusing on the bioavailable fraction for marine biota, OXL
aqueous-phase production is shown to be an important driver of aerosol DFe release, contributing ~44 % to the aerosol Fe dissolution, along with the atmospheric acidity that accounts for ~45 % of total DFe secondary sources in the model. We aim therefore here to realistically represent the atmospheric OXL concentrations to properly simulate the ligand-promoted mineral Fe dissolution.

Present-day simulations indicate that $61.82 \pm 1.29$ Tg yr$^{-1}$ of Fe in EC-Earth is deposited to the Earth's surface, which is
towards the low end (40 - 140 Tg yr$^{-1}$) of the model intercomparison study by Myriokefalitakis et al. (2018). However, the large variability in global models can be partly attributed to the different size ranges considered in the models. Indeed, since most of the Fe mass is associated with coarse dust aerosols (~91 % in this work), models that additionally account for super-coarse mineral dust emission sources (i.e., > 10 μm in diameter), eventually calculate a higher TFe source and thus increased TFe global deposition rates. In our EC-Earth simulations, roughly $0.88 \pm 0.01$ Tg yr$^{-1}$ of DFe are calculated to be deposited
globally (Table 2), in the range of estimates presented in Myriokefalitakis et al. (2018) ($0.8 \pm 0.2$ Tg yr$^{-1}$). Although we do not



consider a super-coarse mode of dust in our simulations, the DFe deposition rates over the remote ocean are not expected to be severely impacted (Myriokefalitakis et al., 2018), but reaching a firm conclusion in this respect will need further work.

Focusing on the marine environment, about 40 % (0.376 ± 0.005 Tg yr⁻¹) of the simulated DFe is deposited into the global ocean, indicating that a large fraction of Fe atmospheric inputs to the global ocean results from the dissolution of atmospheric aerosols. Our calculations are close to the high-end of other global estimates (0.173 - 0.419 Tg yr⁻¹) as presented in the model intercomparison study of Myriokefalitakis et al. (2018), and slightly higher than the respective DFe deposition fluxes in Ito et al. (2021) (~0.271 Tg yr⁻¹). The Fe-containing dust aerosols dominate (~70%) the total deposition fluxes over the ocean in the model, although combustion sources are calculated to have a significant impact over remote oceanic regions, such as the Pacific and the Southern Oceans. The average Fe solubility at the deposition of combustion aerosols is ~19% (Fig. 11a), much higher compared to the solubility of mineral dust aerosols (~2%; Fig. 11b), clearly indicating the importance of atmospheric processing on the potential bioavailable inputs to the global ocean. Indeed, our simulations show high Fe solubilities far from continental regions, such as the tropical Pacific and Atlantic Oceans (Fig. 11c), due to aerosol aging and lower Fe concentrations. Overall, the maximum DFe deposition fluxes occur in EC-Earth downwind of the main desert source regions, with high deposition rates being also simulated in the outflow of tropical biomass burning regions (such as South America, Africa, and Indonesia), as well as over highly populated regions due to the Fe released from anthropogenic combustion processes in the presence of polluted air masses (such as in India and China).

## 5 Summary and conclusions

This work documents the implementation of a detailed multiphase chemistry scheme in the EC-Earth3 Earth system model, aiming to provide consistent estimates of the atmospheric concentrations of the Fe-containing aerosols, along with the species that modulate its atmospheric processing, i.e., OXL and $SO_4^{2-}$. For this, a comprehensive description of the atmospheric Fe cycle is included in the model, accounting for 1) an explicit soil mineralogy, 2) the contribution of combustion emissions, and 3) an atmospheric dissolution scheme that accounts for atmospheric acidity, ambient levels of OXL, and photoinduced processes. The multiphase chemistry scheme simulates the aqueous-phase processes of the troposphere for inorganic and organic compounds, along with Fenton reaction. The KPP software is used in the model to integrate the aqueous-phase and the dissolution equations, which adds flexibility to the code. Overall, simulations of tropospheric chemistry and aerosols for present-day conditions (2000-2014) have been realized, and budget calculations for OXL, $SO_4^{2-}$, and the DFe-containing aerosols have been presented.

Model simulations have been performed both as a coupled system with IFS as well as driven by offline meteorological fields from the ERA-Interim reanalysis. Budget analysis has shown that glyoxal is the main precursor of OXL in the atmosphere (~74 %), and that the potential primary sources in the model (0.373 ± 0.005 Tg yr⁻¹) have a negligible impact on OXL concentrations. Moreover, laboratory-derived aqueous-phase production of OXL due to glyoxal oxidation pathways is also



accounted in our simulations. We have shown that when these pathways are omitted in the simulations, the calculated global OXL atmospheric concentrations can be substantially lowered (~43 %). For the ERA-Interim set-up, the OXL net chemical production is calculated as $12.61 \pm 0.06$ Tg yr$^{-1}$, resulting in an atmospheric burden of ~0.33 Tg. For the online-coupled system,

however, the OXL net chemical production is ~30% lower, mainly attributed to a lower atmospheric OH abundance in EC-Earth due to biases that can be generally found in climate-chemistry models, such as in temperature and humidity especially in the higher altitudes. Overall, the simulated oxidizing capacity along with the contribution of secondary sources other than the GLX oxidation are shown to have a significant impact on the OXL atmospheric abundance. However, we acknowledge that other formation pathways of OXL, primary or secondary, may exist in the gas- and the aqueous phases of the atmosphere

that could further contribute to OXL levels.

The dissolution of dust and combustion aerosols dominates on the dissolved Fe fraction in the model, calculated in EC-Earth at $0.806 \pm 0.014$ Tg Fe yr$^{-1}$, in good agreement with the ERA-Interim simulation and well in the range of other model estimates. Furthermore, a broad evaluation of the EC-Earth proves the models' ability to represent AOD, particularly over regions that are dominated by anthropogenic aerosol, but also over selected dusty sites, in line with previous EC-Earth evaluations.

However, dust is underestimated over remote areas. The model generally underestimates the in-situ OXL measurements, especially during winter, indicating that additional sources (primary and/or secondary) are needed to realistically simulate the atmospheric OXL concentrations. Model comparisons with cruise measurements demonstrate a strong link between atmospheric DFe concentrations and atmospheric composition. The model seems to better capture the observations in the accumulation mode than in the coarse mode. This is attributed to differences either in the atmospheric processing between the

accumulation and coarse particles (i.e., aerosol water content and acidity levels), the misrepresentation in the aerosol sizes (e.g., surface area/volume ratios), or in systematic errors in the mineralogical composition of the emitted Fe-containing soil particles. For this, several developments are planned, aiming to improve the representation of the dust aerosols' size distribution, the description of the mineralogical composition, and the Fe-content in combustion sources, which are expected to reduce the existing uncertainties in the model for a more accurate simulation of the atmospheric Fe cycle.

Emphasizing on the biogeochemistry-related implications of this study, EC-Earth calculates a global annual present-day DFe deposition flux of $0.878 \pm 0.015$ Tg Fe yr$^{-1}$, well within the range of estimates of other global modeling studies. About 40 % of the DFe deposition fluxes are calculated to occur over the global ocean with a strong spatial and temporal variability. The highest annual mean DFe inputs to the global ocean are associated with aerosols of soil origin, especially downwind of the major dust source regions. In addition, Fe-containing combustion aerosols are calculated to have a significant contribution

downwind of biomass burning source regions and highly populated areas in the Northern Hemisphere. It is further demonstrated that over the open ocean the Fe solubility in deposition for aerosols of combustion origin is about an order of magnitude higher than that of mineral dust origin, indicating that the relative contributions of the primary sources can





significantly affect bioavailable aerosol fraction and may thus play an important role in oceanic areas where the phytoplankton growth is limited by Fe supply, such as the Southern Ocean.

It is widely recognized that a combined approach considering both Fe atmospheric processing and deposition over oceans should ideally be used in Earth System Models for the assessment of the impact of nutrient-containing aerosol deposition on marine productivity. A deeper understanding of the atmospheric Fe cycle is thus needed for a better description of the biogeochemistry implications in the presence of a changing climate. Such type of knowledge, however, should be obtained by extensive model evaluation with observations, especially over the remote regions of the world like the Southern Ocean, where

currently the largest discrepancies between models and measurements exist. Therefore, a comprehensive calculation of the Fe physicochemical transformations is necessary to predict the strength of DFe inputs to the ocean, despite the complexity of the related atmospheric multiphase processes. The present study complements, thus, the marine biogeochemistry component of EC-Earth by a fully coupled calculation scheme for atmospheric dissolved Fe fluxes into the global ocean. This EC-Earth model version is expected to eventually allow for a better representation of the marine biogeochemistry perturbations in past

and future climates and air-quality.





## Appendix A

**Table A1. Summary of statistics for all points and per region (as depicted in Fig. 1) for the evaluation of a) the modeled annual mean AOD at 550 nm against AERONET version 3 level 2.0 retrievals for all available stations covering the 2000-2014 period, b) same but for selected dust-dominated AERONET sites (characterized as described in Sect. 2.5), c) the modeled annual mean dust surface concentration for 2000-2014 compared to climatological mean values from RSMA sites and AMMA campaign, and d) the modeled annual dust deposition flux averaged for the period 2000-2014 against observations compiled in Albani et al. (2014) from several sources. The number of stations (n), the Pearsons' correlation coefficients (R) between the simulated and measured monthly mean concentrations, the normalized mean bias (nMB), and the normalized root mean square errors (nRMSE) are indicated for the EC-Earth simulation.**

| | n | R | nMB (%) | nRSME (%) |
|---|---|---|---|---|
| **a) AOD(550 nm)** | | | | |
| N. America | 208 | | -8.8 | 40.9 |
| C. America | 41 | | -2.8 | 33.1 |
| S. America | 41 | | -13.3 | 54.2 |
| Europe | 159 | | -9.9 | 26.9 |
| N. Africa | 27 | | 21.6 | 44.6 |
| S. Africa | 15 | | -35.0 | 61.8 |
| W. Asia | 49 | | -21.5 | 36.6 |
| E. Asia | 162 | | -8.9 | 38.4 |
| Australian Ocean | 26 | | -21.7 | 110.8 |
| Remote oceans | 10 | | 22.9 | 54.2 |
| All points | 738 | 0.8 | -9.1 | 45.7 |
| **b) AOD(550 nm) - dust-dominated sites** | | | | |
| C. America | 2 | | -26.5 | 49.7 |
| N. Africa | 18 | | 21.5 | 43.2 |
| S. Africa | 1 | | -15.4 | 15.4 |
| W. Asia | 12 | | -22.5 | 29.5 |
| E. Asia | 5 | | -8.7 | 14.5 |
| All sites | 38 | 0.76 | 2.7 | 37.2 |
| **c) Dust surface concentrations** | | | | |
| N. America | 1 | | -74.2 | 74.2 |
| C. America | 2 | | -37.5 | 38.7 |
| Europe | 2 | | -90.8 | 116.6 |
| N. Africa | 4 | | 21.9 | 37.7 |
| E. Asia | 2 | | 71.8 | 82.2 |
| Australian Ocean | 3 | | -42.0 | 81.5 |
| S. Pacific Ocean | 3 | | -60.3 | 68.5 |
| N. Pacific Ocean | 4 | | 89.9 | 132.7 |
| Southern Ocean | 2 | | -98.5 | 112.7 |
| All points | 23 | 0.99 | 19.3 | 81.2 |
| **d) Dust deposition rates** | | | | |
| N. America | 7 | | -96.8 | 130.6 |
| C. America | 3 | | -59.6 | 97.6 |
| S. America | 1 | | -99.6 | 99.6 |
| Europe | 14 | | -94.5 | 137.1 |
| N. Africa | 23 | | -56.4 | 142.2 |
| S. Africa | 4 | | -91.2 | 109.3 |
| W. Asia | 5 | | -94.7 | 164.4 |
| E. Asia | 14 | | -25.2 | 94 |
| Australian Ocean | 9 | | -98.3 | 201.1 |
| S. Pacific Ocean | 2 | | -92.0 | 93.2 |
| N. Pacific Ocean | 13 | | 90.5 | 148.9 |
| Southern Ocean | 15 | | -91.9 | 205.0 |
| All points | 110 | 0.75 | -61.2 | 204.0 |





**Table A2. Summary of statistics for the evaluation of the simulated concentrations of a) oxalate, b) sulfate, as well as for c,d,e) the dissolved and f,g,h) the total Fe-containing aerosols for c,f) the accumulation mode, d,g) the coarse mode and e,h) the total suspended particles (tsp), respectively. The number of stations (n), the linear correlation coefficients (R) between the simulated and the measured concentrations, the normalized mean bias (nMB) and the normalized root mean square errors (nRMSE) are indicated for the EC-Earth and ERA-Interim simulation. The results for the sensitivity simulation EC-Earth(sens) for oxalate evaluation and the EC-Earth(AerChem-AMIP) for sulfate evaluation are also shown.**

|  | n | R | nMB (%) | nRSME (%) |
|---|---|---|---|---|
| **a) Oxalate** | | | | |
| EC-Earth | | 0.48 | -64.18 | 116.97 |
| ERA-Interim | 143 | 0.45 | -45.96 | 110.31 |
| ERA-Interim(*sens*) | | 0.44 | -73.66 | 124.89 |
| **b) Sulfate** | | | | |
| EC-Earth | | 0.76 | 15.89 | 55.20 |
| ERA-Interim | 3828 | 0.75 | 22.83 | 57.19 |
| ERA-Interim(*AerChem-AMIP*) | | 0.70 | -4.69 | 57.05 |
| **c) Dissolved Fe (accumulation mode)** | | | | |
| EC-Earth | 438 | 0.51 | 57.56 | 169.91 |
| ERA-Interim | | 0.59 | 56.43 | 169.45 |
| **d) Dissolved Fe (coarse mode)** | | | | |
| EC-Earth | 439 | 0.49 | 22.91 | 193.64 |
| ERA-Interim | | 0.67 | 12.87 | 174.47 |
| **e) Dissolved Fe (tsp)** | | | | |
| EC-Earth | 955 | 0.29 | 59.87 | 297.30 |
| ERA-Interim | | 0.35 | 51.21 | 292.41 |
| **f) Total Fe (accumulation mode)** | | | | |
| EC-Earth | 92 | 0.58 | 21.03 | 170.45 |
| ERA-Interim | | 0.76 | -31.26 | 152.89 |
| **g) Total Fe (coarse mode)** | | | | |
| EC-Earth | 83 | 0.51 | 60.83 | 206.87 |
| ERA-Interim | | 0.58 | 25.35 | 178.39 |
| **h) Total Fe (tsp)** | | | | |
| EC-Earth | 796 | 0.45 | 191.69 | 623.14 |
| ERA-Interim | | 0.47 | 115.75 | 805.95 |

**Code availability**

Access to the model code is restricted to institutes that have signed a memorandum of understanding with the EC-Earth community and a software license agreement with the ECMWF. Confidential access to the code can be granted to editors and reviewers.

**Data availability**

Specific fields can be made available by the authors upon request.

**Author contribution**

SM developed the aqueous-phase chemistry and the iron-dissolution schemes, designed the experiments, and performed the ERA-Interim simulations. EBM, MGA, and CPGP developed the mineralogy applied to dust emissions, co-designed the 1105 experiments, and performed the EC-Earth simulations. AI provided the Fe-containing combustion aerosol emissions. AN provided the ISORROPIA II code. Other co-authors contributed to the development of specific aspects of the model or parts of the code that are shared with EC-Earth3. SM wrote the paper with input from all co-authors.

**Competing interests**

The authors declare that they have no conflict of interest.

**Acknowledgments**

Stelios Myriokefalitakis, Evangelos Gerasopoulos, and Maria Kanakidou acknowledge support by the project "PANhellenic infrastructure for Atmospheric Composition and climatE change" (MIS 5021516) implemented under the Action "Reinforcement of the Research and Innovation Infrastructure", funded by the Operational Programme "Competitiveness, Entrepreneurship and Innovation" (NSRF 2014–2020) and co-financed by Greece and the European Union (European Regional 1115 Development Fund). This work was supported by computational time granted from the National Infrastructures for Research and Technology S.A. (GRNET S.A.) in the National HPC facility - ARIS - under project ID 010003 (ANION). Elisa Bergas-Massó, María Gonçalves-Ageitos, and Carlos Pérez García-Pando thankfully acknowledge the computer resources at Marenostrum4, granted through the PRACE project eFRAGMENT3 and the RES project AECT-2020-3-0020; as well as the technical support provided by the Barcelona Supercomputing Center (BSC) and the CES team of the Earth Sciences 1120 Department. Their work was supported by the ERC Consolidator Grant FRAGMENT (grant agreement No. 773051), and the AXA Chair on Sand and Dust Storms at BSC funded by the AXA Research Fund both led by Carlos Pérez García-Pando, who



also acknowledges the Ramon y Cajal program (grant RYC-2015-18690) of the Spanish Ministry of Science, Innovation and Universities, and the ICREA program. The research leading to these results has also received funding from the Spanish Ministerio de Economía y Competitividad as part of the NUTRIENT project (CGL2017-88911-R) and the H2020 GA 821205

project FORCeS. Support for this research was provided to A.I. by JSPS KAKENHI Grant Number 20H04329, Integrated Research Program for Advancing Climate Models (TOUGOU) Grant Number JPMXD0717935715 from the Ministry of Education, Culture, Sports, Science and Technology (MEXT), Japan. Twan van Noije, Philippe Le Sager, and Maria Kanakidou acknowledge funding from the European Union's 2020 research and innovation programme under grant agreement No. 821205 (FORCeS)". Maria Kanakidou acknowledges support by the Deutsche Forschungsgemeinschaft (DFG, German

Research Foundation) under Germany's Excellence Strategy (University Allowance, EXC 2077, University of Bremen). Maarten C. Krol is supported by the European Research Council (ERC) under the European Union's Horizon 2020 research and innovation program under grant agreement no. 742798 – COS-OCS. Model development was carried out on the GRNET HPC ARIS high-performance computer facility and model simulations were performed at the GRNET HPC ARIS and the BSC Marenostrum4 supercomputer.




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





## Table and Figures

**Table 1. Overview of the simulations performed for this study.**

| Simulation | Description |
| --- | --- |
| EC-Earth | Aqueous-phase chemistry scheme for simulating OXL production and Fe dissolution, coupled to the MOGUNTIA gas-phase chemistry scheme. Meteorology calculated online by IFS and observed sea surface temperature and sea ice concentration boundary conditions (AMIP-CMIP6) are applied. |
| ERA-Interim | As for EC-Earth simulation but driven by meteorological data from the ECMWF reanalysis ERA-Interim. |
| ERA-Interim(*sens*) | As for ERA-Interim simulation but neglecting the contribution of glyoxal high molecular weight species on GLX and OXL formation. |

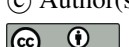



**Table 2. Global budgets, atmospheric burdens, and lifetimes, averaged for the period 2000-2014, of a) oxalate (OXL), b) sulfate as well as, for the dissolved Fe-containing aerosols from c) combustion processes (FeC) and d) mineral dust (FeD) for the EC-Earth, ERA-Interim, and ERA-Interim(*sens*) simulations.**

| | EC-Earth | Era-Interim | Era-Interim(*sens*) |
|---|---|---|---|
| **a) Oxalate** | | | |
| Emissions (Tg yr$^{-1}$) | | 0.373 | |
| Chemistry Production (Tg yr$^{-1}$) | | | |
| ▪  *GLYOLI + OH* | *10.597* | *14.764* | *-* |
| ▪  *MGLY + OH* | *1.079* | *1.415* | *1.552* |
| ▪  *GLX + OH/NO$_3$* | *3.369* | *4.618* | *9.962* |
| Chemistry Loss (Tg yr$^{-1}$) | | | |
| ▪  *OXL + OH/NO$_3$* | *1.019* | *1.189* | *0.822* |
| ▪  *[Fe(OXL)$_2$]$^-$ + hv* | *1.412* | *1.475* | *0.680* |
| Deposition (Tg yr$^{-1}$) | | | |
| ▪  *Dry Deposition* | *0.134* | *0.176* | *0.097* |
| ▪  *Wet Scavenging* | *12.850* | *18.313* | *10.286* |
| Atmospheric Burden (Tg) | 0.219 | 0.330 | 0.189 |
| Lifetime (days) | 5.175 | 5.691 | 5.810 |
| **b) Sulfate** | | | |
| Emissions (Tg S yr$^{-1}$) | | 1.593 | |
| H$_2$SO$_4$ Chemistry Production (Tg S yr$^{-1}$) | | | |
| ▪  *SO$_2$ + OH* | *11.976* | *11.088* | |
| S(VI) Chemistry Production (Tg S yr$^{-1}$) | | | |
| ▪  *S(IV) + H$_2$O$_2$* | *32.902* | *35.812* | |
| ▪  *S(IV) + O$_3$* | *5.927* | *4.760* | |
| ▪  *S(IV) + HO$_2$* | *0.004* | *0.004* | |
| ▪  *S(IV) + CH$_3$O$_2$H* | *0.051* | *0.049* | |
| Deposition (Tg S yr$^{-1}$) | | | |
| ▪  *Dry Deposition* | *3.079* | *2.912* | |
| ▪  *Wet Scavenging* | *49.368* | *50.394* | |
| Atmospheric Burden (Tg S) | 0.692 | 0.961 | |
| Lifetime (days) | 4.816 | 6.579 | |
| **c) Dissolved FeC** | | | |
| Emissions (Tg yr$^{-1}$) | | 0.012 | |
| Dissolution (Tg yr$^{-1}$) | | | |
| ▪  *FeC + H+* | *0.047* | *0.049* | *0.049* |
| ▪  *FeC + OXL* | *0.182* | *0.188* | *0.183* |
| ▪  *FeC + hv* | *0.045* | *0.047* | *0.046* |
| Deposition (Tg yr$^{-1}$) | | | |
| ▪  *Dry Deposition* | *0.081* | *0.080* | *0.077* |
| ▪  *Wet Scavenging* | *0.206* | *0.217* | *0.212* |
| Atmospheric Burden (Tg) | 0.002 | 0.003 | 0.003 |
| Lifetime (days) | 2.970 | 4.163 | 4.203 |
| **d) Dissolved FeD** | | | |
| Emissions (Tg yr$^{-1}$) | 0.059 | 0.049 | 0.049 |
| Dissolution (Tg yr$^{-1}$) | | | |
| ▪  *FeD + H$^+$* | *0.315* | *0.311* | *0.311* |
| ▪  *FeD + OXL* | *0.170* | *0.168* | *0.164* |
| ▪  *FeD + hv* | *0.047* | *0.049* | *0.048* |
| Deposition (Tg yr$^{-1}$) | | | |
| ▪  *Dry Deposition* | *0.140* | *0.132* | *0.130* |
| ▪  *Wet Scavenging* | *0.452* | *0.446* | *0.441* |
| Atmospheric Burden (Tg) | 0.006 | 0.010 | 0.010 |
| Lifetime (days) | 3.837 | 6.236 | 6.264 |







**Figure 1: Site location map of observations a) for AOD (AERONET; red dots, AERONET-DUST; blue squares), dust surface concentration (RSMA; purple squares, and AMMA; orange diamonds) and dust deposition rates (several sources compiled in Albani et al. (2014); green triangles) and b) for surface oxalate (OXL; blue triangles), cruise aerosol Fe concentrations (red circles), and sulfate (green diamonds).**



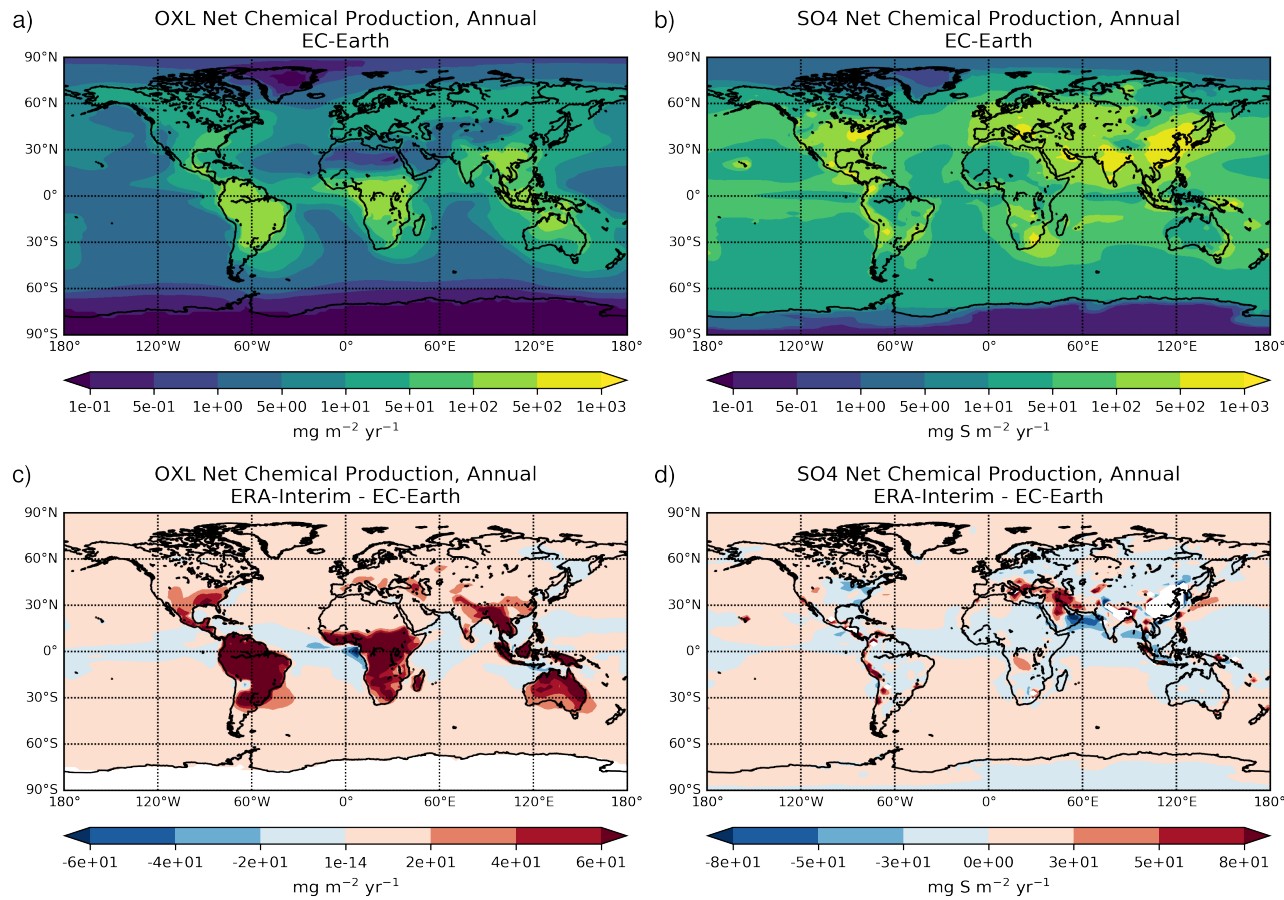

**Figure 2: Annual mean net chemical production rates for a) oxalate (mg m⁻² yr⁻¹) and d) sulfate (mg S m⁻² yr⁻¹) as calculated for the EC-Earth simulation averaged for the period 2000-2014, and the respective absolute differences to the ERA-Interim simulation (c,d).**







l670

**Figure 3: Annual mean dissolution rates (mg m⁻² yr⁻¹) of combustion (a) and mineral dust (b) aerosols as calculated for the EC-Earth simulation averaged for the period 2000-2014, and the respective absolute differences to the ERA-Interim simulation (c,d).**





**Figure 4: Comparison of a) the modeled annual mean AOD at 550 nm against AERONET retrievals for all available stations covering the 2000-2014 period, b) same but for selected dusty AERONET sites, c) the modeled annual mean dust surface concentration for 2000-2014 compared to climatological mean values from RSMA sites and AMMA campaign, and d) modeled annual dust deposition flux averaged for the period 2000-2014 against observations compiled in Albani et al. (2014) from several sources.**





**Figure 5: Oxalate (OXL) surface concentrations (µg m$^{-3}$) for the boreal winter (DJF; a) and boreal summer (JJA; b) as simulated for the EC-Earth simulation averaged for the period 2000-2014, and the respective absolute differences to the ERA-Interim simulation for surface (c,d) and zonal mean (e,f).**







**Figure 6: Comparison of daily mean observations (black line) of OXL (ng m⁻³; left) and nss-SO₄²⁻ (µg m⁻³; right) with the EC-Earth (red line) and the ERA-Interim (green line) simulations for a,b) Finokalia (Greece) for the period July 2004 - July 2006 (Koulouri et al., 2008), for c,d) Puy de Dome (France) and e,f) Azores (Portugal), for the period September 2002 - September 2004 (Legrand et al., 2007), along with d,f) scatterplot comparisons for observations around the globe; the solid line represents the 1 : 1 correspondence and the dashed lines show the 10 : 1 and 1 : 10 relationships, respectively. For completeness, the comparisons for the sensitivity simulation ERA-Interim(sens) (blue line/squares) for OXL and the EC-Earth(AerChem-AMIP) (blue line/squares) for sulfate are also presented. Gray shaded areas represent the standard deviation of the observations and for the EC-Earth simulations the error bars represent the standard error of the multi-annual mean for the individual observational period.**





**Figure 7: Sulfate (SO4) surface concentrations (µg m⁻³) for the boreal winter (DJF; a) and boreal summer (JJA; b) as simulated for the EC-Earth simulation averaged for the period 2000-2014, and the respective absolute differences to the EC-Earth(AerChem-AMIP) simulation for surface (c,d) and zonal mean (e,f).**

695



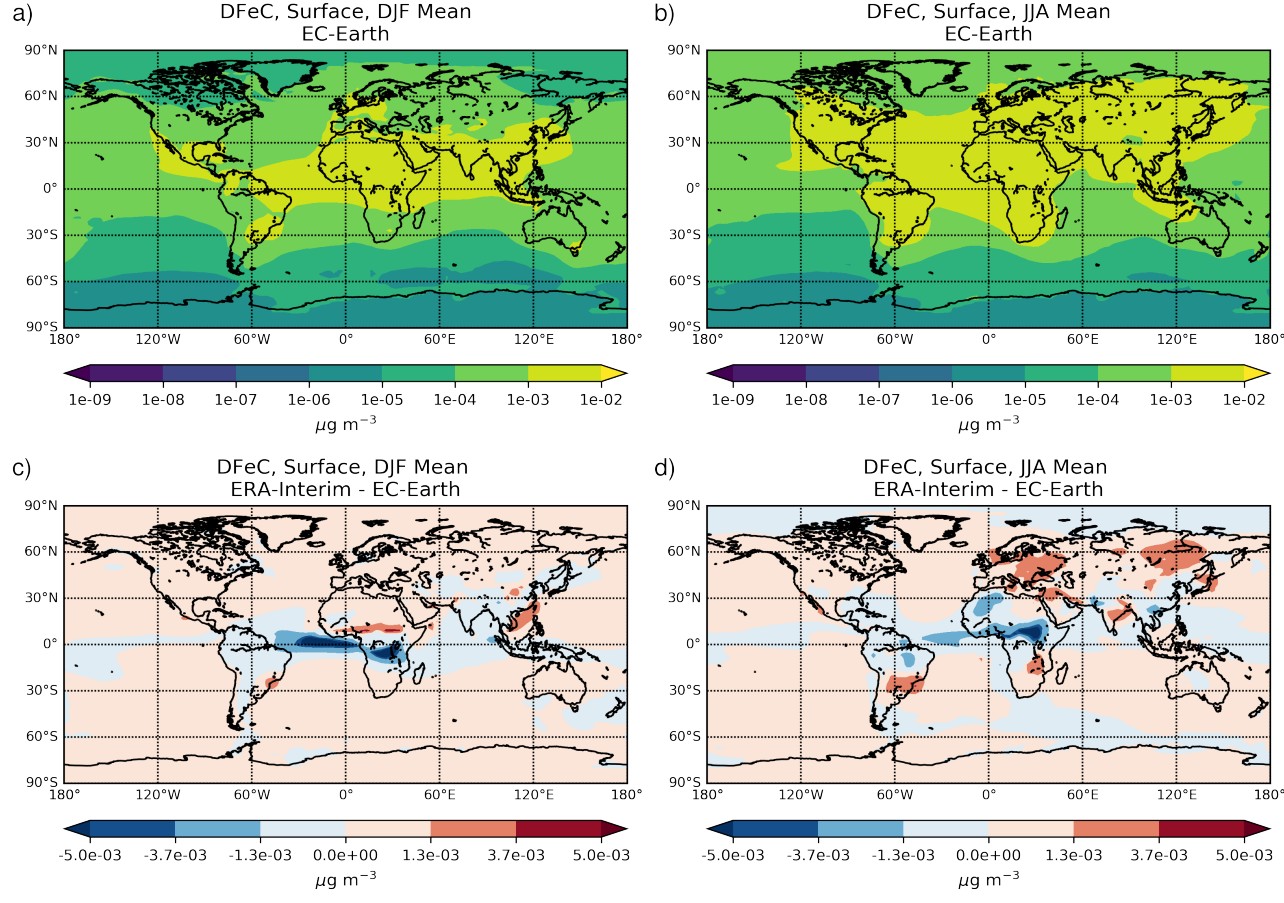

**Figure 8: Dissolved iron surface concentrations (μg m⁻³) from combustion aerosols (DFeC) for the boreal winter (DJF; a) and summer (JJA; b) seasons for the EC-Earth simulation averaged for the period 2000-2014, and the respective absolute differences to the ERA-Interim simulation (c,d).**

700





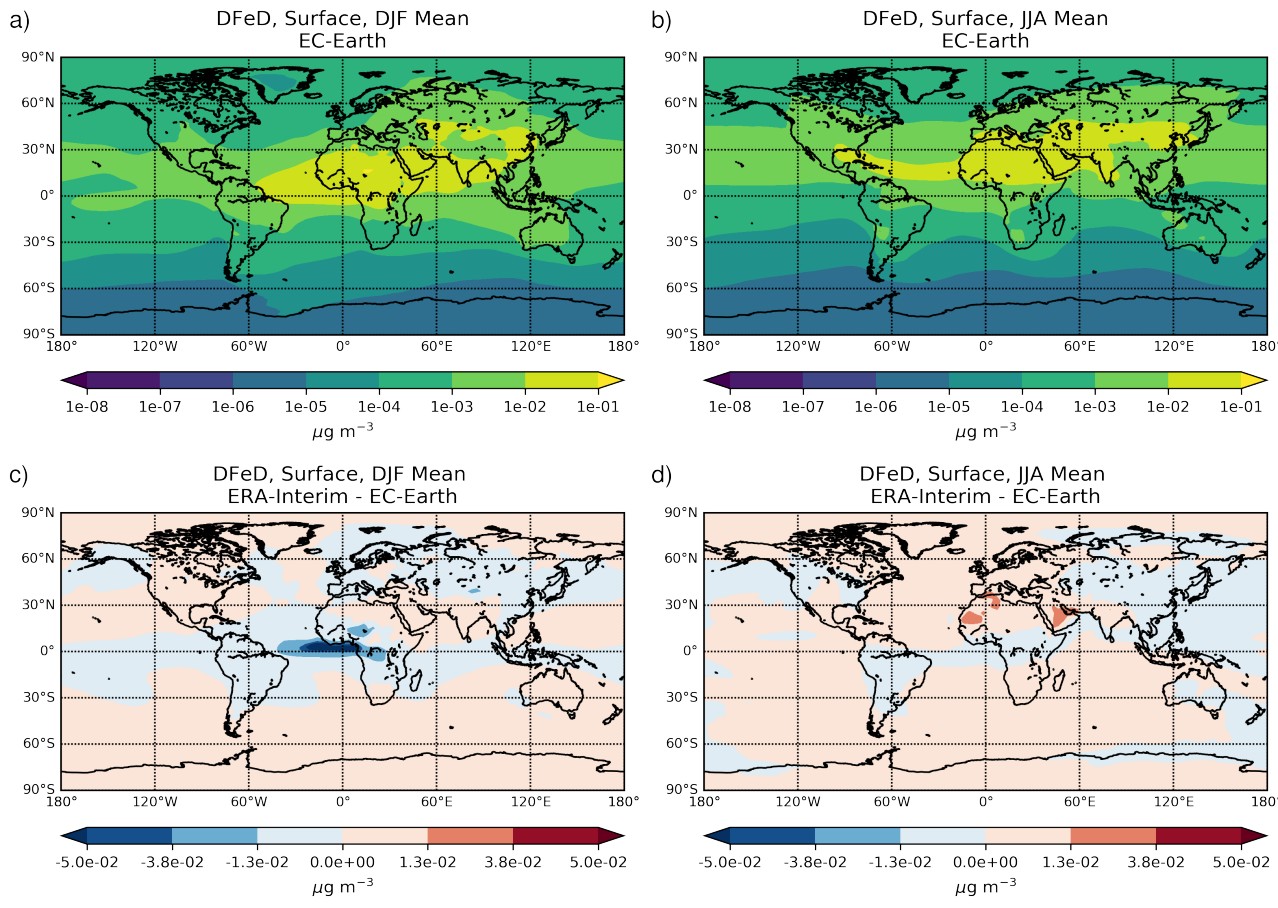

**Figure 9: Dissolved iron surface concentrations (µg m⁻³) from mineral dust (DFeD) for the boreal winter (DJF; a) and summer (JJA; b) seasons for the EC-Earth simulation averaged for the period 2000-2014, and the respective absolute differences to the ERA-Interim simulation (c,d).**

l705





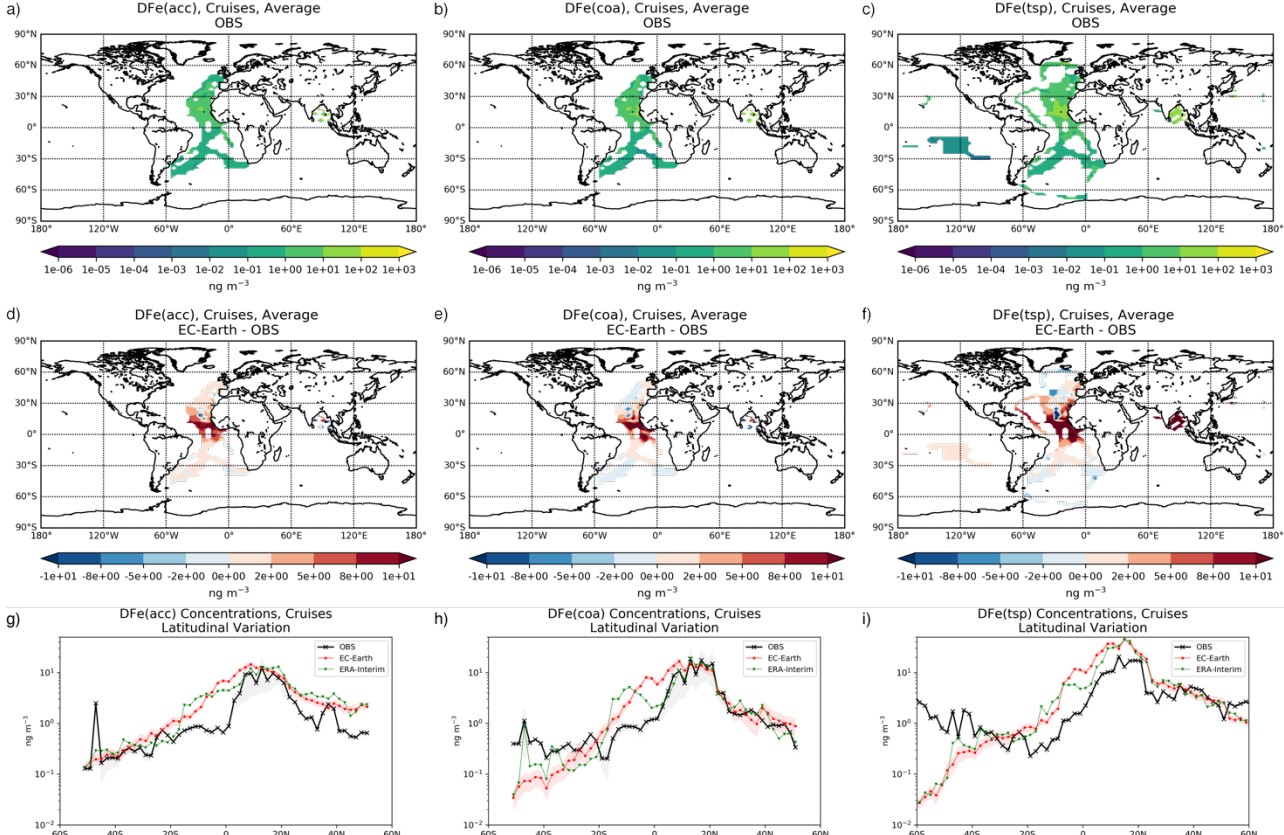

**Figure 10: Observed dissolved iron (DFe) concentrations (ng m⁻³) of a) accumulation aerosols, b) coarse aerosols, and c) total suspended particles (tsp), the respective absolute differences to the ERA-Interim simulation (d, e, f), and the comparison to observations (black x-line) in latitudinal order (g,e,f) with the EC-Earth (red circle-line) and ERA-Interim(green triangle-line) simulations; the grey shaded areas correspond to the standard deviation of the observations and the red shaded areas correspond to the standard error of the multi-annual mean for the individual observational period for the EC-Earth simulations.**

1710



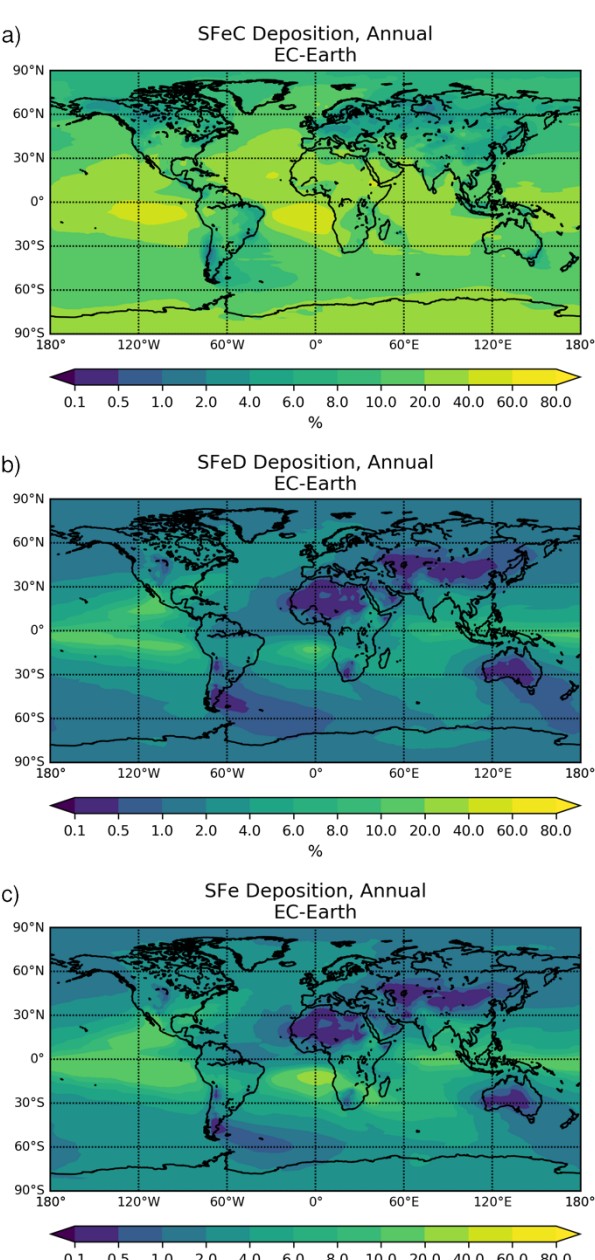

**Figure 11: Annual mean Fe-containing aerosol solubility at deposition fluxes (%) as simulated for the EC-Earth simulation averaged for the period 2000-2014, for a) mineral dust aerosols, b) sum of solid fuel combustion, liquid fuel combustion and open biomass burning aerosols, and c) sum of all aerosol sources.**