# Peer review of "Multiphase processes in the EC-Earth Earth System model and their relevance to the atmospheric oxalate, sulfate, and iron cycles"

_Geoscientific Model Development, 2021_

## Referee Comment (RC1)

**Review of "Multiphase processes in the EC-Earth Earth System model and their relevance to the atmospheric oxalate, sulfate, and iron cycles" by *Myriokefalitakis et al.* (2021)**

The manuscript by *Myriokefalitakis et al.* (2021) describes the results of EC-Earth3, an Earth System Model, after the implementation of a detailed multiphase chemistry scheme. The focus of this study is on the cycling of oxalate (OXL), sulfate, and bioavailable/soluble iron (SFe) in the aerosol phase. The model is also tested with two sets of meteorological fields. Model results of the variables of interested are compared to available observations. The article is very dense; however, in general, is well written and describes complex issues clearly. The complex chemistry scheme implemented is well referenced and all assumptions were justified to the best of the author's ability. While I feel the paper could benefit from some shortening and simplifying, the manuscript is scientifically sound and presents interesting results. I only have minor issues with this manuscript. After addressing the minor comments below I feel this paper is sufficient for publication in *Geoscientific Model Development* (GMD).

**Scientific Comments**

1. The authors mention that the metastable assumption in ISORROPIA II can lead to overprediction of aerosol acidity (i.e., lower pH values) compared to the stable aerosol assumption. Can the authors provide some estimate of how this might impact the results of the model simulations? For instance, is the atmospheric processing production of SFe noticeably larger due to the metastable assumption? It would be good to know how this uncertainty in aerosol acidity calculations might impact the multiphase chemistry incorporated in the model.

2. How many passive tracers had to be added to the model in order to simulate all the gas phase and aqueous species, and the multiphase chemical reactions, represented in Table S2? Did this significantly increase the computational expense of the model?

3. The standard deviation around the mean of the multi-year OXL (and some other species) is very small. Does this mean there is very little interannual variability (IAV) in the primary emissions (I see from Table 2 that the primary emissions are constant) and precursor species emission, and production/destruction processes? One would think that there would be IAV in emission source strength of precursor species, transport/deposition, and other meteorological conditions impacting OXL production/distribution.

4. The paper is very dense. The amount of quantitative values for species emission, production/destruction, deposition, and evaluation statistics of each species, and comparison to other recent studies, presented in the text of the article is a bit overwhelming. It makes reading and understanding the manuscript difficult. After much effort I feel that all the values seem reasonable; however, this took significant effort. This might impact the effectiveness of presenting the important results of this paper. I don't think this is 100% necessary, but I would suggest that the authors think of ways to simplify the text of the paper.

5. How important is the comparison of the model results, in all sections of the manuscript, with both sets of meteorological fields? I almost think this part of the manuscript could be a supplemental section. This would reduce the density of the article's information in the main body of the paper.

**Technical Comments**

1. Line 563. considered in that latter study.

2. The use of "~". The authors use the approximation symbol for nearly every value presented in the manuscript. It seems that the values are often pretty exact (e.g., line 576 for the atmospheric lifetime of 5.7 days) and likely do not need this symbol.

3. Line 583. "calculates that is ~ 3 % lower" needs rewording.

4. Line 603.  is produced.

5. Line 661. downwind of land areas.

---

## Referee Comment (RC2)

**Review of Myriokefalitakis et al. (2021): Multiphase processes in the EC-Earth Earth System model and their relevance to the atmospheric oxalate, sulfate, and iron cycles.**

This manuscript describes the development of a detailed multiphase chemistry scheme within EC-Earth with the worthy aim to realistically represent the atmospheric iron cycle. This can improve understanding of marine biogeochemistry perturbations, and thus carbon and nitrogen cycles, under past and future climate scenarios. The model contains iron from both natural and anthropogenic sources and contains schemes for the dissolution of insoluble iron to a soluble form which account kinetically for the solution's acidity, oxalic acid, and irradiation. The chemistry required is discussed in great detail and results from two sets of simulation are compared to observations of oxalate, sulphate, and iron aerosol. The analysis is very good, and results are interesting given that representing the full atmospheric aerosol iron cycle within a global Earth System Model is still a new development. In my opinion this manuscript is very well written and in particular the description of the chemistry was very well made. I feel this manuscript is thus suitable for publication in *Geoscientific Model Development* after addressing a series of, mainly minor, comments.

1. Please double check the level of significance throughout the manuscript. There are some instances of where a higher precision is given than likely should be. Also please be consistent with the level of precision within a given paragraph, best to not let it vary without giving the justification.
2. When reporting the average for shipborne observations (and comparing to the model) it is recommend using the median and not the mean (or give both). This is how the observational papers report these values (e.g., any Baker et al. paper) and a full reasoning is given in (Hamilton et al., 2019) outlining why for such sparse (spatially and temporally) datasets the mean can often be misleading. Using the median may also result in a better model: obs correlation. Please add medians and alter discussion where needed.
3. Maybe useful to point to Table S2 earlier within the introduction to help guide the reader through the many reaction mechanisms discussed.
4. Methods: Please describe the aerosol model further. Some points to include for example would be mixing assumptions (internal vs. external), which modes the iron aerosol goes into (and if ageing of aerosol information is needed please add), how the aerosol number concentrations are calculated and the associated new constants used for iron aerosol, the vertical distribution of biomass burning emissions (if any), and how dry and wet deposition are handled (briefly). There may be more beyond this list too.
5. If Fe fractions are used to generate combustion iron emissions, how are the coarse sized emissions of carbonaceous aerosol estimated? CMIP6 inventories have assumed all BC and OC is in the accumulation mode.
6. Please add the required additional model simulation time (e.g., core hours) for the new tracers and chemistry. It may be easiest to give both the base model EC-Earth3 run time and the new EC-Earth3-Iron runtimes.
7. Using a hematite fraction of 66% iron (Table S1) is higher than other studies (e.g., (Journet et al., 2008) gives 57.5%). What is the reasoning for using this value?
8. Have any of the modelled dust optical properties been modified by accounting for dust mineralogy? is the new iron aerosol from combustion sources interacting with the radiation scheme? Please briefly what are the potential impacts/feedbacks on relevant variables for

the chemistry (temperature, humidity, etc.) when comparing online (EC-Earth) vs. offline (ERA-Interim) simulations of adding (or not adding) these couplings.

9.  What is the dust emission flux (Tg/a)? How does that compare to recent estimates (e.g., (Kok et al., 2021; Wu et al., 2020))? And were dust emissions tuned in any way (e.g., (Ridley et al., 2016) recommendations to attain a global mean dust AOD of 0.03)?
10. Does dust minerology alter the AOD estimates in the model?
11. Is dust aerosol internally mixed with sea spray aerosol? How does this impact dust lifetimes?
12. Can the Authors describe why is the lifetime of dust iron much larger than combustion iron? I would have guessed the other way as dust is larger and thus more prone to being lost from the atmosphere by dry deposition. This is therefore maybe interesting.
13. The discussion section compares results with Myriokefalitakis et al.,2018 and Ito et al. 2021. However, neither of these studies contain results from an Earth System Model using a modal aerosol scheme containing iron aerosol. It therefore would be insightful to include a comparison to the more similar model aerosol framework from (Hamilton et al., 2020, GRL) or (Hamilton et al., 2019). Furthermore, (Hamilton et al., 2020) covers the same study years as presented here, which the other studies do not.

L70-75: What of other anthropogenic fuel sources beyond oil? How do these compare?

L140: Can the authors elaborate more on why a continental source is important?

L185: Is DMS in EC-Earth truly prognostic as it uses a climatological monthly mean ocean surface conc. from Lana et al. (2011)? The gas transfer velocity is then parameterized following Wanninkhof (2014).

L194: Please define anthropogenic (sources are those from Table S1?) and biomass burning (anthropogenic or also wildfires?). Was a metal smelting source accounted for? At L219 it states, "Fe is also emitted in the model from anthropogenic combustion and biomass burning sources following Ito et al. (2018) and Hajima et al. (2019)", Ito et al. do consider metal smelting, but I can find no reference here to this particular source.

L213: What is the mean PSD (accumulation: coarse dust) ratio?

L215: Is the 0.1% placed in "fast" at point of emission? And can the Authors describe a little more their reasoning for using 0.1% solubility for all iron bearing minerals regardless of mineralogy.

L219: I cannot quite link up how this methodology follows Ito et al. (2018) and Hajima et al. (2019). Maybe it is best to describe what was done here to estimate combustion iron emissions in more detail. For example, Ito et al. uses supermicron and submicron PM values while Hajima et al. states that MIROC uses a $0.4gFe\ gBC^{-1}$ ratio. But I cannot find how either of these link to the different values given in Table S1, and if values in Table S1 are iron fraction w.r.t. to total aerosol or to carbonaceous aerosol only. Some more description of how these were derived in the main text and the Table header would thus be beneficial. Also, within Table S1 where are values for biomass burning 0.63 (accumulation) and 2.30 (coarse) from – also Ito et al. 2019? (maybe add refs to Table S1?). For the biomass burning how do these values compare to the multi-biome ratio estimates given in (Hamilton et al., 2019, 2022)?

L225: For the study period this is the GFED4s fire emission dataset, maybe also reference as such?

L226: While an iron solubility of 79% for oil has been recorded, it is a value at the high end of the literature (8-85%; e.g., (Rathod et al., 2020)). Furthermore, MIROC uses 79% for the final solubility of oil sourced iron at the point of deposition. As 79% is used at point of emission here this iron will only increase after atmospheric processing. I think it would be useful therefore to describe what possible implications this has for the current study, future model development, and comparison to observations shown.

L228: I found it a little confusing to read biomass burning alongside anthropogenic in the same sentence here. This could be read to imply that only fires from human activity are accounted for in the model. Maybe best to sperate these sentences and explain why this assumption holds for biomass burning (e.g., assumed no change in vegetation type over the study period?).

L598: Maybe duplicate this information about no ocean sources to the methods, alongside where it is discussed there are no gasoline engines sources (L237), to be complete in the methods.

Figure 4: Would be nice to also have the statistics on the figure

Figure 6 (g+h) Would be nice to also have the statistics on the figure

Figure 10: The use of contouring in the shading of the map creates a somewhat false impression of the values given it interpolates between cells and there are some high gradients and lone cells. I feel that it would be better to colour each grid cell individually. Also how were the observations averaged; all observations in a given cell collected and the average taken I am assuming, but maybe his can be more explicitly described in the caption. Can the standard error also be included for the ERA line in the bottom panels?

Figures (general): Please alter line colours where red and green are used to make figures more accessible for colour blind readers.

Table S1: Please add refs for values.

Figure S5: Is it possible to add a scatter plot for iron solubility.

**Technical comments:**

L59: uptake of *atmospheric* $CO_2$

L132: Seems quite a precise approximate value

L211: PSD is only used twice and in the same paragraph; likely does not warrant an acronym.

L765: 'outstandingly" is hyperbole, please give the values and allow the reader to come to that conclusion if merited.

**References:**

Hamilton, D. S., Scanza, R. A., Feng, Y., Guinness, J., Kok, J. F., Li, L., Liu, X., Rathod, S. D., Wan, J. S., Wu, M. and Mahowald, N. M.: Improved methodologies for Earth system modelling

of atmospheric soluble iron and observation comparisons using the Mechanism of Intermediate complexity for Modelling Iron (MIMI v1.0), Geosci. Model Dev., 12(9), 3835–3862, doi:10.5194/gmd-12-3835-2019, 2019.

Hamilton, D. S., Scanza, R. A., Rathod, S. D., Bond, T. C., Kok, J. F., Li, L., Matsui, H. and Mahowald, N. M.: Recent (1980 to 2015) Trends and Variability in Daily-to-Interannual Soluble Iron Deposition from Dust, Fire, and Anthropogenic Sources, Geophys. Res. Lett., 47(17), e2020GL089688, doi:10.1029/2020GL089688, 2020.

Hamilton, D. S., Perron, M. M. G., Bond, T. C., Bowie, A. R., Buchholz, R. R., Guieu, C., Ito, A., Maenhaut, W., Myriokefalitakis, S., Olgun, N., Rathod, S. D., Schepanski, K., Tagliabue, A., Wagner, R. and Mahowald, N. M.: Earth, Wind, Fire, and Pollution: Aerosol Nutrient Sources and Impacts on Ocean Biogeochemistry, Ann. Rev. Mar. Sci., 14(1), 1–28, doi:10.1146/annurev-marine-031921-013612, 2022.

Journet, E., Desboeufs, K. V., Caquineau, S. and Colin, J. L.: Mineralogy as a critical factor of dust iron solubility, Geophys. Res. Lett., 35(7), 3–7, doi:10.1029/2007GL031589, 2008.

Kok, J., Adebiyi, A., Albani, S., Balkanski, Y., Checa-Garcia, R., Chin, M., Colarco, P., Hamilton, D. S., Huang, Y., Ito, A., Klose, M., Leung, D., Li, L., Mahowald, N., Miller, R., Obiso, V., Pérez García-Pando, C., Rocha-Lima, A., Wan, J. and Whicker, C.: Improved representation of the global dust cycle using observational constraints on dust properties and abundance, Atmos. Chem. Phys., 21, 8127–8167, doi:10.5194/acp-2020-1131, 2021.

Lana, A., Bell, T. G., Simó, R., Vallina, S. M., Ballabrera-Poy, J., Kettle,  a. J., Dachs, J., Bopp, L., Saltzman, E. S., Stefels, J., Johnson, J. E. and Liss, P. S.: An updated climatology of surface dimethlysulfide concentrations and emission fluxes in the global ocean, Global Biogeochem. Cycles, 25(1), 1–17, doi:10.1029/2010GB003850, 2011.

Rathod, S. D., Hamilton, D. S., Mahowald, N. M., Klimont, Z., Corbett, J. J. and Bond, T. C.: A mineralogy-based anthropogenic combustion-iron emission inventory, J. Geophys. Res. Atmos., 125(17), e2019JD032114, doi:10.1029/2019jd032114, 2020.

Ridley, D. A., Heald, C. L., Kok, J. F. and Zhao, C.: An observationally constrained estimate of global dust aerosol optical depth, Atmos. Chem. Phys., 16(23), 15097–15117, doi:10.5194/acp-16-15097-2016, 2016.

Wu, C., Lin, Z. and Liu, X.: Global dust cycle and uncertainty in CMIP5 models, Atmos. Chem. Phys., 5, 1–52, doi:10.5194/acp-2020-179, 2020.

---

## Author Comment (AC2)

We thank the reviewer for the careful reading of the manuscript and the insightful comments. Please find below our point-by-point replies:

**Scientific Comments**

**SC1.** **The authors mention that the metastable assumption in ISORROPIA II can lead to overprediction of aerosol acidity (i.e., lower pH values) compared to the stable aerosol assumption. Can the authors provide some estimate of how this might impact the results of the model simulations? For instance, is the atmospheric processing production of SFe noticeably larger due to the metastable assumption? It would be good to know how this uncertainty in aerosol acidity calculations might impact the multiphase chemistry incorporated in the model.**

- We thank the reviewer for this comment. In retrospect, this comment should have been rephrased differently: "*The metastable assumption produces pH values that are different from the stable assumption. Work to date, such as Bougiatioti et al. (2016), Guo et al. (2018; 2015) and others identified in the review of Pye et al. (2020), has shown that the metastable solution tends to provide semi-volatile partitioning of pH-sensitive species (like $NH_3/NH_4$ and $HNO_3/NO_3$) and aerosol liquid water content that is closer to observations – at least for when the RH is above 40%. For this reason, we assume that the most plausible estimates of acidity are to be obtained with the metastable assumption – and we base our simulations on that.*"

  Nevertheless, we have further investigated the sensitivity of dissolved Fe to the phase state assumption, by running the model (ERA-Interim) for one year with ISORROPIA-II in the forward mode but following the stable assumption.

[Figure]

Our results indicate that the total Fe dissolution is lower by ~20 % on average due to the total cumulative effects of the different aerosol pH and liquid water content predicted. Moreover, the expected location of soluble Fe production is also affected by the different aerosol phase state assumptions (i.e., metastable vs. stable), which in turn affects the spatial distribution of soluble iron deposition (SFe). The stable aerosol

assumption leads generally to lower solubilities in deposition over source regions, such as for dust-dominated (~ -1%) as for combustion-dominated (~ -3%). However, the larger impact is calculated over the remote ocean (to up to ~ -4%), where high iron solubilities are combined with low iron-containing aerosol concentrations. We note, however, that dust outflow and wildfire-dominated regions still receive most of the soluble iron, in contrast to the Southern Ocean that receives much lesser soluble iron.

**SC2.** **How many passive tracers had to be added to the model in order to simulate all the gas phase and aqueous species, and the multiphase chemical reactions, represented in Table S2? Did this significantly increase the computational expense of the model?**

- The aqueous phase driver alone accounts for 68 tracers that can either be partitioned, dissolved (in the case of minerals), and/or oxidized in the aqueous phase of the atmosphere. However, upon the calculation of the aqueous phase chemistry, the model does not trace all the different forms of the species participating in the aqueous-phase chemistry scheme, meaning that the different aqueous-phase forms are not prognostic variables (tracers) in the model, as we also stated in the manuscript. Instead, we trace the final concentrations that can be transported and deposited. Thus, for this work we account only for tracers needed to properly describe the mineral-Fe solubilization processes; specifically, oxalate and glyoxylic acid, as well as the mineral dust-Fe according to three dissolution classes; namely fast, intermediate, and slow Fe pools (overall, 10 species additionally).

  The aqueous-phase chemistry calculations, nevertheless, do increase the computational time of the model. To make it more clear to the reader as the reviewer proposed, we added a new subsection (i.e., Sect. 2.6) focusing specifically on the model's performance: "*The coupling of the aqueous-phase chemistry scheme along with the description of the atmospheric iron cycle for this work increases the model runtime. EC-Earth3-Iron uses 109 transported and 33 non-transported tracers, which are significantly larger numbers than in the EC-Earth3-AerChem configuration (i.e., 69 transported and 21 non-transported tracers). Note, however, that EC-Earth3-Iron used for this work employs the MOGUNTIA gas-phase chemistry scheme configuration, in contrast to the mCB05 configuration used in EC-Earth3-AerChem, which is overall found to be computationally more expensive by ~27% alone (Myriokefalitakis et al., 2020). In the Marenostrum4 supercomputer architecture (2x Intel Xeon Platinum 8160 24C at 2.1 GHz), the EC-Earth3-AerChem configuration (van Noije et al., 2021) simulates 1.85 years per day simulation time (SYPD) with 187 CPUs, while to reach to a comparable performance (i.e., 1.41 SYPD) with the EC-Earth3-Iron configurations, 432 CPUs are required, respectively. This means overall, that the EC-Earth3-Iron corresponds to 7353 computation hours per year (CHPY), which is roughly 3 times larger than the standard EC-Earth3-AerChem.*"

**SC3.** **The standard deviation around the mean of the multi-year OXL (and some other species) is very small. Does this mean there is very little interannual variability (IAV) in the primary emissions (I see from Table 2 that the primary emissions are constant) and precursor species emission, and production/destruction processes? One would think that there would be IAV in emission source strength of precursor species, transport/deposition, and other meteorological conditions impacting OXL production/distribution.**

- The EC-Earth3-Iron version used for this study is built on EC-Earth3-AerChem (van Noije et al., 2021). Thus, we used the same emission set-up for anthropogenic, biomass burning, as well as for biogenic and other natural emissions as in van Noije et al. (2021).

However, in EC-Earth3-AerChem, the biogenic emissions are prescribed using monthly estimates from the MEGAN-MACC data set (Sindelarova et al., 2014) for the year 2000, as produced by the Model of Emissions of Gases and Aerosols from Nature (MEGAN) version 2.1 under the Monitoring Atmospheric Composition and Climate (MACC) project. Accounting that OXL is mainly produced from precursors of biogenic origin (i.e., mainly isoprene), we do not thus expect a strong IAV on the calculation of its production, i.e., in EC-Earth the OXL production is $12.61 \pm 0.06$ Tg yr$^{-1}$. On the other hand, the primary emissions of OXL, although very uncertain as we note in the manuscript, are very low based on the published estimates and are not expected to have a substantial impact on OXL atmospheric concentrations. Note that in Table 2 we provide the average of the simulated period (see also caption), but in the text, we also provide the IAV, i.e., $0.373 \pm 0.005$ Tg yr$^{-1}$; see also Sect. 3.1.1. Nevertheless, we here present simulations only for a short period (2000-2014) of present-day, thus no important IAV is expected.

**SC4.** **The paper is very dense. The amount of quantitative values for species emission, production/destruction, deposition, and evaluation statistics of each species, and comparison to other recent studies, presented in the text of the article is a bit overwhelming. It makes reading and understanding the manuscript difficult. After much effort I feel that all the values seem reasonable; however, this took significant effort. This might impact the effectiveness of presenting the important results of this paper. I don't think this is 100% necessary, but I would suggest that the authors think of ways to simplify the text of the paper.**

- This manuscript aims to include the most important information about the calculations of the precursors of oxalate and how this can impact Fe-containing minerals dissolution. Such a layout can help the reader to have in one place all the critical information concerning the chemistry calculations, along with a comparison to other state-of-the-art studies. This layout is, thus, necessary to describe the complex chemical formation pathways of OXL and its precursors, as in other studies available in the literature (e.g., Lin et al., 2012; Liu et al., 2012; Myriokefalitakis et al., 2011), as well as to provide enough references to guide the reader for further reading.

**SC5.** **How important is the comparison of the model results, in all sections of the manuscript, with both sets of meteorological fields? I almost think this part of the manuscript could be a supplemental section. This would reduce the density of the article's information in the main body of the paper.**

- Our goal here is to couple a chemistry scheme that can satisfactorily simulate the aqueous-phase production of oxalate to properly simulate the mineral Fe dissolution processes in the atmosphere. However, using only the calculated (online) meteorology of EC-Earth, we cannot come to a safe conclusion based on present-day observations concerning the actual strength of oxalate production in the aqueous phase of the atmosphere, due to biases in the meteorological conditions that unavoidably exist in an ESM. On the other hand, the ERA-Interim setup allows constraining the model with the observations-based assimilated meteorological data and is therefore used for the budget analysis and comparison with other modeling estimates from the literature. All in all, the comparison with ERA-Interim is used to explore uncertainties regarding the aqueous-phase chemistry scheme in EC-Earth and thus its impact on Fe-containing aerosol solubilization processes. For this, we keep in the discussion the results of both sets of meteorological fields.

**Technical Comments**

**TC1.   Line 563. considered in that latter study.**
  - Corrected.

**TC2.   The use of "~". The authors use the approximation symbol for nearly every value presented in the manuscript. It seems that the values are often pretty exact (e.g., line 576 for the atmospheric lifetime of 5.7 days) and likely do not need this symbol.**
  - Corrected.

**TC3.   Line 583. "calculates that is ~ 3 % lower" needs rewording.**
  - Rephrased as "*while in EC-Earth is calculated 3 % lower.*"

**TC4.   Line 603.  is produced.**
  - Corrected.

**TC5.   Line 661. downwind of land areas.**
  - Corrected.

**References**

[revised manuscript text omitted]

---

## Author Comment (AC3)

We thank the reviewer for the careful reading of the manuscript and the insightful comments. Please find below our point-by-point replies:

**Specific Comments**

**SC1.** **Please double check the level of significance throughout the manuscript. There are some instances of where a higher precision is given than likely should be. Also please be consistent with the level of precision within a given paragraph, best to not let it vary without giving the justification.**
- We have revised the manuscript and have corrected the level of precision.

**SC2.** **When reporting the average for shipborne observations (and comparing to the model) it is recommend using the median and not the mean (or give both). This is how the observational papers report these values (e.g., any Baker et al. paper) and a full reasoning is given in (Hamilton et al., 2019) outlining why for such sparse (spatially and temporally) datasets the mean can often be misleading. Using the median may also result in a better model: obs correlation. Please add medians and alter discussion where needed.**
- Medians have been added and the presentation of the model comparison with observations has been adapted accordingly in the revised manuscript.

**SC3.** **Maybe useful to point to Table S2 earlier within the introduction to help guide the reader through the many reaction mechanisms discussed.**
- In the introduction, we refer to the reactions available in the literature that can take place in the aqueous phase of the atmosphere. In Table S2, however, we do not include all the available aqueous-phase reactions, but only those included in our chemistry scheme. As we have also indicated in the manuscript, such a level of complexity (i.e., as described in the introduction) cannot be implemented in a global modeling study. Thus, we use Table S2 to only present and discuss the aqueous-phase chemistry scheme of this study (including the simplifications unavoidably adopted for this work) and not the generic chemistry equations. Nevertheless, we believe that we provide enough references to guide the reader for further reading.

**SC4.** **Methods: Please describe the aerosol model further. Some points to include for example would be mixing assumptions (internal vs. external), which modes the iron aerosol goes into (and if ageing of aerosol information is needed please add), how the aerosol number concentrations are calculated and the associated new constants used for iron aerosol, the vertical distribution of biomass burning emissions (if any), and how dry and wet deposition are handled (briefly). There may be more beyond this list too.**
- Aerosols are described with the size-resolved modal microphysics scheme M7 (Aan de Brugh et al., 2011; Vignati et al., 2004). M7 uses seven log-normal size distributions or modes with predefined geometric standard deviations. There are four water-soluble modes (nucleation, Aitken, accumulation, and coarse) and three insoluble modes (Aitken, accumulation, and coarse). The aerosol module is however described in a rather detailed manner in several previous publications of the model such as van Noije et al. (2014) and more recently in van Noije et al. (2021), which presents the EC-Earth3-AerChem version of the model that is the base of our developments, as explained in the manuscript. Since for this work, we did not change the aerosol module of the model, and the paper is already very dense and lengthy, we believe that it is unnecessary to repeat the same information, and it is better to refer to van Noije et al. (2021). Overall, the iron species are added to the model on top of the aerosols already present, they follow the same size distribution as the

original aerosols represented in the model (accumulation and coarse, soluble, and insoluble).

The following sentences are added in the revised manuscript: "*M7 uses seven log-normal size distributions with predefined geometric standard deviations, including four water-soluble modes (nucleation, Aitken, accumulation, and coarse) and three insoluble modes (Aitken, accumulation, and coarse). Note that the new developments of this work are added to the model on top of the aerosols already represented by M7, and that the new aerosol components are introduced using the existing M7 modes.*"

**SC5.   If Fe fractions are used to generate combustion iron emissions, how are the coarse sized emissions of carbonaceous aerosol estimated? CMIP6 inventories have assumed all BC and OC is in the accumulation mode.**

- This work is not focused on the calculation of the iron combustion emissions but on the atmospheric dissolution processes of Fe-containing aerosols, and for this we just used the respective emissions as calculated by Ito et al. (2018). Thus, we applied here the derived emission factors (per sector) of fine and coarse-sized combustion iron emissions on the fine carbonaceous aerosol emissions available from CMIP6. According, however, to Ito et al. (2018), the emissions of total and coarse particulate matter are calculated by those of PM1 (with the term PM1 we refer to OC, BC, and particulate inorganic matters). The sub-micron and super-micron Fe emissions are then estimated by the respective Fe contents of sub-micron and super-micron particles. For more detail, please see also our reply in SC20.

**SC6.   Please add the required additional model simulation time (e.g., core hours) for the new tracers and chemistry. It may be easiest to give both the base model EC-Earth3 run time and the new EC-Earth3-Iron runtimes.**

- A new subsection (i.e., Sect. 2.6) focused on the model's performance is added in the manuscript. Please see also our reply to Reviewer#1's SC2.

**SC7.   Using a hematite fraction of 66% iron (Table S1) is higher than other studies (e.g., (Journet et al., 2008) gives 57.5%). What is the reasoning for using this value?**

- The hematite estimate in Claquin et al. (1999) and Nickovic et al. (2012) is commonly used as a proxy for iron oxides. The value used in our model follows the same approach as in Nickovic et al. (2013), who takes an average estimate of 66% for the content of iron in the iron oxides. For clarity, we changed the term "*Hematite*" to "*Iron oxides*" in Table S1.

**SC8.   Have any of the modelled dust optical properties been modified by accounting for dust mineralogy? is the new iron aerosol from combustion sources interacting with the radiation scheme? Please briefly what are the potential impacts/feedbacks on relevant variables for the chemistry (temperature, humidity, etc.) when comparing online (EC-Earth) vs. offline (ERA-Interim) simulations of adding (or not adding) these couplings.**

- The new iron species do not interact with the radiation in the model. The main purpose of adding them is to study and analyze the effects of iron deposition upon ocean biogeochemistry. The optical properties of dust remain thus unaffected after the developments included in the model and they are compared to AERONET retrievals to evaluate the overall performance of the atmospheric composition estimates by our model. It is out of the scope of this work to assess the impact of the mineralogy on the radiative effects of dust, but it is certainly an interesting aspect we could implement and explore in the future.

**SC9. What is the dust emission flux (Tg/a)? How does that compare to recent estimates (e.g., (Kok et al., 2021; Wu et al., 2020))? And were dust emissions tuned in any way (e.g., (Ridley et al., 2016) recommendations to attain a global mean dust AOD of 0.03)?**

- The dust scheme and the tuning parameters used are identical to those of the EC-Earth3-AerChem version used in the CMIP6-AerChemMIP experiments (van Noije et al., 2021). Our focus in this work, however, was not to improve or change the dust scheme itself, but to include a detailed definition of the iron sources and its solubilization process. The mean (2000-2014) dust emission flux in EC-Earth equals $1256.7 \pm 25.8$ Tg yr$^{-1}$. That said, the model dust emission estimates fall in the low range of the AEROCOM phase III models. In more detail, the annual global mean dust emissions in EC-Earth are lower than the $3250\pm77$ Tg yr$^{-1}$ calculated by Hamilton at al. (2019) corresponding to about the same period, but in the range of other literature estimates of ~500-5000 Tg yr$^{-1}$ (Huneeus et al., 2011; Kok et al., 2017, 2021; Wu et al., 2020). We note, however, that EC-Earth accounts for emission flux of dust with a geometric diameter up to 8 µm (van Noije et al., 2021), which is in the low range of 1200-2900 Tg yr$^{-1}$ reported by Kok et al. (2021) for inverse model results of about the same dust aerosol sizes. Furthermore, the average optical depth for dust at 550 nm (annual mean over the 2000-2014) yields a value of 0.032 +/- 0.005. The following text is added in the revised manuscript: "*Overall, the average optical depth for dust at 550 nm (annual mean over the 2000-2014) yields a value of $0.032 \pm 0.005$, which falls well in the range of observationally based estimates, based on in-situ measurements, satellites and global models, i.e., $0.030 \pm 0.005$ (Ridley et al., 2016). EC-Earth3-Iron annual mean dust emission amount to $1256.7 \pm 25.8$ Tg yr$^{-1}$, which falls in the lower range of the AEROCOM phase III models (Gliß et al., 2021), and is also at the low end of the range estimated by Kok et al. (2021) from inverse modelling (~1200-2900 Tg yr$^{-1}$), valid for dust aerosol with a geometric diameter $\leq 10\mu m$ (PM10).*"

**SC10. Does dust minerology alter the AOD estimates in the model?**

- No, we do not trace minerals in this model version. We include only tracers for iron and calcium, with the focus of improving the aqueous phase chemistry and the estimate of soluble iron deposition.

**SC11. Is dust aerosol internally mixed with sea spray aerosol? How does this impact dust lifetimes?**

- The aerosol population in M7 is composed of an external mixture of insoluble and internally mixed populations (Vignati et al., 2004). Part of the dust is insoluble and only the soluble part is internally mixed with sea salt in the accumulation and coarse modes. The latter part is subject to enhanced wet scavenging by clouds and rain. Thus, since part of dust aerosols are present in the mixed phase, their lifetime can be thus impacted by the wet deposition process.

**SC12. Can the Authors describe why is the lifetime of dust iron much larger than combustion iron? I would have guessed the other way as dust is larger and thus more prone to being lost from the atmosphere by dry deposition. This is therefore maybe interesting.**

- In Table 2, the presented lifetimes over deposition correspond to the soluble Fe-containing aerosols. Focusing on the Fe-combustion aerosols, roughly 91% of combustion Fe is here emitted in the coarse mode, indicating that most of the dissolved Fe from combustion processes is also produced in the coarse aerosols. This is in line with Ito et al. (2018) where the Fe oxides emitted from combustion sources largely reside in super-micron aerosols, as indicated by observations. As the reviewer notices, the Fe associated with mineral dust is

expected to reside in larger particles and thus being lost faster from the atmosphere due to dry deposition processes. This is in particular true also in our simulations when comparing the lifetimes of total Fe aerosols over dry deposition processes: the lifetime of the Fe-containing dust particles is calculated in EC-Earth at about 7.9 days, and for the Fe-containing particles from combustion processes at about 12.5 days. On the other hand, the respective lifetimes for wet deposition processes indicate that the insoluble Fe combustion aerosols are converted more easily to soluble forms compared to the dust Fe-containing aerosols, resulting in global lifetimes due to wet deposition processes of 4.5 and 10.9 days, respectively.

**SC13. The discussion section compares results with Myriokefalitakis et al. 2018 and Ito et al. 2021. However, neither of these studies contain results from an Earth System Model using a modal aerosol scheme containing iron aerosol. It therefore would be insightful to include a comparison to the more similar model aerosol framework from (Hamilton et al., 2020, GRL) or (Hamilton et al., 2019). Furthermore, (Hamilton et al., 2020) covers the same study years as presented here, which the other studies do not.**

- We thank the reviewer for attracting our attention to the publications by (Hamilton et al., 2019, 2020) and, where relevant, we added the respective information in the discussion section of the revised manuscript: "*The amount of total Fe deposited to the global ocean is calculated to be $12.94 \pm 0.31$ Tg $yr^{-1}$ in EC-Earth, which is about 50% lower than recent estimates by Hamilton et al. (2019), owing to the significantly larger (almost double) mineral dust emission flux in the latter study. On the other hand, our result is close to the high-end of other global estimates (0.173 - 0.419 Tg $yr^{-1}$) as presented in the model intercomparison study of Myriokefalitakis et al. (2018), and slightly higher than the respective DFe deposition fluxes in Ito et al. (2021) (~0.271 Tg $yr^{-1}$). Thus, even though we do not consider a super-coarse mode of dust in our simulations, the DFe deposition rates over the remote ocean are not severely impacted (Myriokefalitakis et al., 2018) by the size of the emitted minerals, but instead by the atmospheric processing during long-range transport. Nevertheless, reaching a firm conclusion in that respect will need further work. The Fe-containing dust aerosols dominate (~70%) the total deposition fluxes over the ocean in the model, although combustion sources are calculated to have a significant impact on the Fe input to remote oceanic regions, such as the Pacific and the Southern Oceans, in agreement with other studies (e.g., Hamilton et al., 2020).*"

**SC14. L70-75: What of other anthropogenic fuel sources beyond oil? How do these compare?**

- We rephrased in the revised manuscript as: "*Significantly higher Fe solubilities are found, however, for anthropogenic combustion-related Fe-containing aerosols, especially for Fe in oil fly ash from industries and shipping, which is mainly in the form of ferric sulfates (Chen et al., 2012; Ito, 2013; Rathod et al., 2020; Schroth et al., 2009).*"

**SC15. L140: Can the authors elaborate more on why a continental source is important?**

- The continental source is important for the aqueous-phase production of the Fe-oxalato complexes because they include Fe-containing aerosols from natural (dust) and combustion processes that can have an impact on the aqueous-phase OH production, especially via iron photochemistry. We add the following sentence in the revised manuscript *"...where elevated concentrations of Fe-containing aerosols, both of lithogenic and pyrogenic sources, can exist.*"

**SC16. L185: Is DMS in EC-Earth truly prognostic as it uses a climatological monthly mean ocean surface conc. from Lana et al. (2011)? The gas transfer velocity is then parameterized following Wanninkhof (2014).**

- We refer to the oceanic DMS emissions, indicating that they are calculated online in the model, i.e., meaning that they depend on parameters calculated by the model that can affect the strength of its source. We mentioned that to distinguish it from other emissions that are derived based on climatological maps (see L185). As mentioned in more detail in van Noije et al. (2021), the DMS flux in ice-free ocean areas is calculated as the product of the local surface ocean DMS concentration and the gas transfer velocity. Although the ocean concentrations are prescribed according to the monthly climatology from Lana et al. (2011), the gas transfer velocity is parameterized following Wanninkhof (2014), as a function of the wind speed (i.e., $U_{10}^2$) and the SST through the Schmidt number, both calculated online in the model. Thus, even though sea-water DMS concentrations are prescribed, DMS emissions to the atmospheric are calculated online, and atmospheric DMS is calculated prognostically by the chemistry scheme.

**SC17. L194: Please define anthropogenic (sources are those from Table S1?) and biomass burning (anthropogenic or also wildfires?). Was a metal smelting source accounted for? At L219 it states, "Fe is also emitted in the model from anthropogenic combustion and biomass burning sources following Ito et al. (2018) and Hajima et al. (2019)", Ito et al. do consider metal smelting, but I can find no reference here to this particular source.**

- The definition of anthropogenic (including fossil and biomass fuels) and biomass burning (excluding biofuel combustion) is following CMIP6. Only a part of metal smelting processes (i.e., pig-iron production) is included as the industrial emissions of carbonaceous aerosol emissions in CMIP6 (Hoesly et al., 2018), as was pointed out by Rathod et al. (2020). We note that estimates of other processes in the smelting industry remain highly uncertain (Rathod et al., 2020). We now included this in the supplement (caption of Table S2), and we rephrased in the revised manuscript as: "*Fe is also emitted in the model from anthropogenic activities (including fossil and biomass fuels) and biomass burning (excluding biofuel combustion) following Ito et al. (2018). The estimate of Fe emission from metal smelting remain highly uncertain and further works are needed (Rathod et al., 2020).*"

**SC18. L213: What is the mean PSD (accumulation: coarse dust) ratio?**

- The mean PSD (2000-2014) is equal to 0.072, as derived from the emitted dust accumulation and coarse dust aerosols in the model

**SC19. L215: Is the 0.1% placed in "fast" at point of emission? And can the Authors describe a little more their reasoning for using 0.1% solubility for all iron bearing minerals regardless of mineralogy.**

- For this work, 0.1% of all Fe-containing mineral soil emissions are assumed directly soluble (Ito and Shi, 2016). To make it more clear to the reader, we rephrased that part in the revised manuscript as: "*No relationship of Fe dissolution with other elements is observed, however, for clays and feldspars where the total Fe content of the minerals is very low (< 0.54%) and the Fe is in the form of impurities (Journet et al., 2008). For this, 0.1 % Fe content in total Fe-containing minerals is here assumed directly soluble as amorphous free iron impurities regardless of mineralogy (Ito and Shi, 2016).*"

**SC20. L219: I cannot quite link up how this methodology follows Ito et al. (2018) and Hajima et al. (2019). Maybe it is best to describe what was done here to estimate combustion iron**

**emissions in more detail. For example, Ito et al. uses supermicron and submicron PM values while Hajima et al. states that MIROC uses a 0.4gFe gBC[-1] ratio. But I cannot find how either of these link to the different values given in Table S1, and if values in Table S1 are iron fraction w.r.t. to total aerosol or to carbonaceous aerosol only. Some more description of how these were derived in the main text and the Table header would thus be beneficial. Also, within Table S1 where are values for biomass burning 0.63 (accumulation) and 2.30 (coarse) from – also Ito et al. 2019? (maybe add refs to Table S1?). For the biomass burning how do these values compare to the multi-biome ratio estimates given in (Hamilton et al., 2019, 2022)?**

For this work, we did not recalculate the iron emissions from biomass burning but, instead, we used the Fe/PM factors as derived from Ito (2011) and Ito et al. (2018) estimates. MIROC uses the global mean ratio of 0.04 gFe gBC[-1] ratio for biomass burning in fine particles from Ito et al. (2018), which is consistent with the global mean ratio of 0.032 in Hamilton et al. (2022). We note that there was a typo in the caption of Figure 5 in Hamilton et al. (2022): "Range in observed (a) iron:black carbon, (b) phosphorus:black carbon, and (c) iron:aluminum ratios in fire aerosol as reported in the studies listed in Supplemental Table 3", instead of "Supplemental Table 2". To make it more clear to the reader, we rephrase this part in the manuscript as: "*The Fe-containing fossil fuel and biofuel combustion emissions are here estimated by applying specific factors (i.e., per emission sector and per particle size) to the total particulate emissions (i.e., the sum of organic carbon, black carbon, and inorganic matter), as derived for this work based on estimates from Ito et al. (2018) for the Fe content in the sub-micron and super-micron combustion aerosols. As for the biomass burning, the iron fractions in the fine particles are related to the combustion stages of flaming (0.46 ± 0.51 %) and smoldering (0.06 ± 0.03 %) fires, while the averaged iron fraction is used for coarse particles (3.4 %) (Ito, 2011). The global mean ratio of 0.04 gFe gBC-1 for biomass burning in fine particles is consistent with that of 0.032 in the review paper by Hamilton et al. (2022).*"

We also rephrased the caption of Table S1 as: "*Averaged factors for the years 2000-2014, used to represent the Fe-containing aerosols in the emitted fine and coarse aerosols of the model, as applied to a) the calculated dust mineral emissions as derived from the updated mineralogy maps originally created by Claquin et al. (1999), b) the CMIP6 anthropogenic sectors (Hoesly et al., 2018) as retrieved based on estimates from Ito et al. (2018) for the emitted submicron carbonaceous particulate matter (i.e., sum of OC and BC, and inorganic matters), and c) the CMIP6 biomass burning (van Marle et al., 2017) also based on Ito et al. (2018). In parentheses, the standard deviation (where available) is also provided.*"

We further note that Ito et al. (2018) estimated Fe emission for the present day. The description of historical emission from shipping sources, from coal to oil fuel, were described in Hajima et al., 2019). For clarity, we removed the respective Hajima et al. (2019) reference since we are here only focus on the years 2000-2014.

**SC21. L225: For the study period this is the GFED4s fire emission dataset, maybe also reference as such?**

- We here refer to the historical fire emissions from van Marle et al. (2017) as also mentioned in van Noije et al. (2021). For clarity, we prefer to keep it as it is.

**SC22. L226: While an iron solubility of 79% for oil has been recorded, it is a value at the high end of the literature (8-85%; e.g., (Rathod et al., 2020)). Furthermore, MIROC uses 79% for the final solubility of oil sourced iron at the point of deposition. As 79% is used at point of emission here this iron will only increase after atmospheric processing. I think**

**it would be useful therefore to describe what possible implications this has for the current study, future model development, and comparison to observations shown.**

- An iron solubility of ~79% for oil corresponds to the average solubility (2000-2014) applied in ship emissions for our study. The value of 79% represents the high solubility of iron emissions in oil fly ash (Ito et al., 2021). Rathod et al. (2020) proposed a lower solubility in emissions (i.e., 47.5 % for iron-sulfates) with an upper value, however, up to ~90%.

  To further investigate this in our model, a sensitivity simulation for one year using the central iron sulfate solubility for ship oil emissions of 47.5% is performed, leading, however, only to a slight decrease up to ~2% in iron solubility in deposition, mainly in the northern Atlantic Ocean, whereas globally the solubility flux increases by only ~0.7% due to the relatively higher availability of insoluble iron in ship oil emissions. On more open oceanic regions, however, the differences in iron solubility are smaller, meaning that an initially solubility at the high end of reported values (Rathod et al., 2020) is not expected to significantly impact our results.

[Figure]

  Ship oil combustion can nevertheless contribute significantly to the high Fe solubility found in low Fe loadings over the remote oceans. Nevertheless, the range of the solubility in emissions of oil combustion processes remains still highly uncertain (Rathod et al., 2020) and current global model capabilities might be unable to resolve such subscale (combustion) processes. The focus of this work, however, is the impact of atmospheric processing on the Fe aerosols, and not the Fe-related emissions that are successfully created in other studies, such as Ito et al. (2018) and more recently by Rathod et al. (2020).

  We added the following parts in the revised manuscript: "*We note that the value of 79 % represents the high solubility of iron emissions in oil fly ash (Ito et al., 2021).* (Ito et al., 2018) *Rathod et al. (2020) proposed a lower solubility in emissions (i.e., 47.5 % for iron-sulfates) with an upper value at ~90%. We further note that although here a relatively high Fe solubility is applied for ship oil combustion emission, a sensitivity simulation (not shown) using an iron solubility for ship oil emissions of 47.5%, as proposed by Rathod et al. (2020), leads overall to only slight decreases (up to ~2 %) in iron solubility in deposition, mainly in the northern Atlantic Ocean, and does not substantially affect our results.*"

**SC23. L228: I found it a little confusing to read biomass burning alongside anthropogenic in the same sentence here. This could be read to imply that only fires from human activity are accounted for in the model. Maybe best to sperate these sentences and explain why this assumption holds for biomass burning (e.g., assumed no change in vegetation type over the study period?).**

- We here referred to the fractions used to incorporate Fe emissions in our model, as derived here by Ito et al. (2018). Following the reviewer's comment, however, we split the sentence

as: "*The year-to-year variation in anthropogenic combustion Fe emission factors follows Ito et al. (2018). On the contrary, no such variation for the factors on wildfires Fe emissions is provided.*"

**SC24.  L598: Maybe duplicate this information about no ocean sources to the methods, alongside where it is discussed there are no gasoline engines sources (L237), to be complete in the methods.**

- In L237, we only discuss for the potential primary OXL sources, not the secondary ones. Although GLY is the main precursor of OXL, we prefer to keep all information about the known OXL precursors in one place not to confuse the reader. For this, we keep the discussion for GLY sources in L598.

**SC25.  Figure 4: Would be nice to also have the statistics on the figure**

- We removed statistics from all figures upon the editor's request for visibility reasons. However, also following the editor's comment, we now include all statistics in the Appendix.

**SC26.  Figure 6 (g+h) Would be nice to also have the statistics on the figure**

- Please see reply SC25.

**SC27.  Figure 10: The use of contouring in the shading of the map creates a somewhat false impression of the values given it interpolates between cells and there are some high gradients and lone cells. I feel that it would be better to colour each grid cell individually. Also how were the observations averaged; all observations in a given cell collected and the average taken I am assuming, but maybe this can be more explicitly described in the caption. Can the standard error also be included for the ERA line in the bottom panels?**

- All observations in each cell are averaged (spatially and temporally) to produce Fig. 10. We now include it in the figure's caption along with the standard error for the ERA-Interim simulation.

**SC28.  Figures (general): Please alter line colours where red and green are used to make figures more accessible for colour blind readers.**

- Done

**SC29.  Table S1: Please add refs for values.**

- We added references in the caption of Table S1 (please see our reply in SC22).

**SC30.  Figure S5: Is it possible to add a scatter plot for iron solubility?**

- The respective iron solubility plots for accumulation aerosols, coarse aerosols, and total suspended matter have been added.

**Technical comments**

**TC1.   L59: uptake of atmospheric CO$_2$**
- Corrected.

**TC2.   L132: Seems quite a precise approximate value**
- Corrected.

**TC3.   L211: PSD is only used twice and in the same paragraph; likely does not warrant an acronym.**
- Corrected.

**TC4.   L765: 'outstandingly" is hyperbole, please give the values and allow the reader to come to that conclusion if merited.**
- Rephrased as "*to reproduce satisfactorily…*".

**References**

Aan de Brugh, J. M. J., Schaap, M., Vignati, E., Dentener, F., Kahnert, M., Sofiev, M., Huijnen, V. and Krol, M. C.: The European aerosol budget in 2006, Atmos. Chem. Phys., 11(3), 1117–1139, doi:10.5194/acp-11-1117-2011, 2011.

Claquin, T., Schulz, M. and Balkanski, Y. J.: Modeling the mineralogy of atmospheric dust sources, J. Geophys. Res. Atmos., 104(D18), 22243–22256, doi:10.1029/1999JD900416, 1999.

Hajima, T., Watanabe, M., Yamamoto, A., Tatebe, H., Noguchi, A., Abe, M., Ohgaito, R., Ito, A., Yamazaki, D., Okajima, H., Ito, A., Takata, K., Ogochi, K., Watanabe, S. and Kawamiya, M.: Description of the MIROC-ES2L Earth system model and evaluation of its climate-biogeochemical processes and feedbacks, Geosci. Model Dev. Discuss, doi:10.5194/gmd-2019-275, 2019.

Hamilton, D. S., Scanza, R. A., Feng, Y., Guinness, J., Kok, J. F., Li, L., Liu, X., Rathod, S. D., Wan, J. S., Wu, M. and Mahowald, N. M.: Improved methodologies for Earth system modelling of atmospheric soluble iron and observation comparisons using the Mechanism of Intermediate complexity for Modelling Iron (MIMI v1.0), Geosci. Model Dev., 12(9), 3835–3862, doi:10.5194/gmd-12-3835-2019, 2019.

Hamilton, D. S., Moore, J. K., Arneth, A., Bond, T. C., Carslaw, K. S., Hantson, S., Ito, A., Kaplan, J. O., Lindsay, K., Nieradzik, L., Rathod, S. D., Scanza, R. A. and Mahowald, N. M.: Impact of Changes to the Atmospheric Soluble Iron Deposition Flux on Ocean Biogeochemical Cycles in the Anthropocene, Global Biogeochem. Cycles, 34(3), doi:10.1029/2019GB006448, 2020.

Hoesly, R. M., Smith, S. J., Feng, L., Klimont, Z., Janssens-Maenhout, G., Pitkanen, T., Seibert, J. J., Vu, L., Andres, R. J., Bolt, R. M., Bond, T. C., Dawidowski, L., Kholod, N., Kurokawa, J., Li, M., Liu, L., Lu, Z., Moura, M. C. P., O Rourke, P. R. and Zhang, Q.: Historical (1750–2014) anthropogenic emissions of reactive gases and aerosols from the Community Emissions Data System (CEDS), Geosci. Model Dev., 11(1), 369–408, doi:10.5194/gmd-11-369-2018, 2018.

Huneeus, N., Schulz, M., Balkanski, Y., Griesfeller, J., Prospero, J., Kinne, S., Bauer, S., Boucher, O., Chin, M., Dentener, F., Diehl, T., Easter, R., Fillmore, D., Ghan, S., Ginoux, P., Grini, A., Horowitz, L., Koch, D., Krol, M. C., Landing, W., Liu, X., Mahowald, N., Miller, R., Morcrette, J.-J., Myhre, G., Penner, J., Perlwitz, J., Stier, P., Takemura, T. and Zender, C. S.: Global dust model intercomparison in AeroCom phase I, Atmos. Chem. Phys., 11(15), 7781–7816, doi:10.5194/acp-11-7781-2011, 2011.

Ito, A.: Mega fire emissions in Siberia: potential supply of bioavailable iron from forests to the ocean, Biogeosciences, 8(6), 1679–1697, doi:10.5194/bg-8-1679-2011, 2011.

Ito, A., Lin, G. and Penner, J. E.: Radiative forcing by light-absorbing aerosols of pyrogenetic iron oxides, Sci. Rep., 8(1), 7347, doi:10.1038/s41598-018-25756-3, 2018.

Ito, A., Ye, Y., Baldo, C. and Shi, Z.: Ocean fertilization by pyrogenic aerosol iron, npj Clim. Atmos. Sci., 4(1), 30, doi:10.1038/s41612-021-00185-8, 2021.

Kok, J. F., Ridley, D. A., Zhou, Q., Miller, R. L., Zhao, C., Heald, C. L., Ward, D. S., Albani, S. and Haustein, K.: Smaller desert dust cooling effect estimated from analysis of dust size and abundance, Nat. Geosci., 10(4), 274–278, doi:10.1038/ngeo2912, 2017.

Kok, J. F., Adebiyi, A. A., Albani, S., Balkanski, Y., Checa-Garcia, R., Chin, M., Colarco, P. R., Hamilton, D. S., Huang, Y., Ito, A., Klose, M., Leung, D. M., Li, L., Mahowald, N. M., Miller, R. L., Obiso, V., Pérez García-Pando, C., Rocha-Lima, A., Wan, J. S. and Whicker, C. A.: Improved representation of the global dust cycle using observational constraints on dust properties and abundance, Atmos. Chem. Phys., 21(10), 8127–8167, doi:10.5194/acp-21-8127-2021, 2021.

Lana, A., Bell, T. G., Simó, R., Vallina, S. M., Ballabrera-Poy, J., Kettle, A. J., Dachs, J., Bopp, L., Saltzman, E. S., Stefels, J., Johnson, J. E. and Liss, P. S.: An updated climatology of surface dimethlysulfide concentrations and emission fluxes in the global ocean, Global Biogeochem. Cycles, 25(1), GB1004, doi:10.1029/2010GB003850, 2011.

van Marle, M. J. E., Kloster, S., Magi, B. I., Marlon, J. R., Daniau, A.-L., Field, R. D., Arneth, A., Forrest, M., Hantson, S., Kehrwald, N. M., Knorr, W., Lasslop, G., Li, F., Mangeon, S., Yue, C.,

Kaiser, J. W. and van der Werf, G. R.: Historic global biomass burning emissions for CMIP6 (BB4CMIP) based on merging satellite observations with proxies and fire models (1750–2015), Geosci. Model Dev., 10(9), 3329–3357, doi:10.5194/gmd-10-3329-2017, 2017.

Nickovic, S., Vukovic, A., Vujadinovic, M., Djurdjevic, V. and Pejanovic, G.: Technical Note: High-resolution mineralogical database of dust-productive soils for atmospheric dust modeling, Atmos. Chem. Phys., 12(2), 845–855, doi:10.5194/acp-12-845-2012, 2012.

Nickovic, S., Vukovic, A. and Vujadinovic, M.: Atmospheric processing of iron carried by mineral dust, Atmos. Chem. Phys., 13(18), 9169–9181, doi:10.5194/acp-13-9169-2013, 2013.

van Noije, T. P. C., Le Sager, P., Segers, A. J., van Velthoven, P. F. J., Krol, M. C., Hazeleger, W., Williams, A. G. and Chambers, S. D.: Simulation of tropospheric chemistry and aerosols with the climate model EC-Earth, Geosci. Model Dev., 7(5), 2435–2475, doi:10.5194/gmd-7-2435-2014, 2014.

van Noije, T., Bergman, T., Le Sager, P., O'Donnell, D., Makkonen, R., Gonçalves-Ageitos, M., Döscher, R., Fladrich, U., von Hardenberg, J., Keskinen, J.-P., Korhonen, H., Laakso, A., Myriokefalitakis, S., Ollinaho, P., Pérez García-Pando, C., Reerink, T., Schrödner, R., Wyser, K. and Yang, S.: EC-Earth3-AerChem: a global climate model with interactive aerosols and atmospheric chemistry participating in CMIP6, Geosci. Model Dev., 14(9), 5637–5668, doi:10.5194/gmd-14-5637-2021, 2021.

Rathod, S. D., Hamilton, D. S., Mahowald, N. M., Klimont, Z., Corbett, J. J. and Bond, T. C.: A Mineralogy-Based Anthropogenic Combustion-Iron Emission Inventory, J. Geophys. Res. Atmos., 125(17), e2019JD032114, doi:10.1029/2019JD032114, 2020.

Vignati, E., Wilson, J. and Stier, P.: M7: An efficient size-resolved aerosol microphysics module for large-scale aerosol transport models, J. Geophys. Res. Atmos., 109(D22), D22202, doi:10.1029/2003JD004485, 2004.

Wanninkhof, R.: Relationship between wind speed and gas exchange over the ocean revisited, Limnol. Oceanogr. Methods, 12(6), 351–362, doi:10.4319/lom.2014.12.351, 2014.

Wu, C., Lin, Z. and Liu, X.: The global dust cycle and uncertainty in CMIP5 (Coupled Model Intercomparison Project phase 5) models, Atmos. Chem. Phys., 20(17), 10401–10425, doi:10.5194/acp-20-10401-2020, 2020.

---

## Author Response (AR2)

Dear Editor,

Please find here the final version of our manuscript, including all relevant changes made upon revision. In the final version, we removed some extra "~" characters, as referee#1 asked. Moreover, we now include the correct Fig. 4d, after correction of a small bug in the relevant plotting script. Some small differences in the values of Table A2, and the respective discussion in Sect. 3.2.1, are exist in this final version, without however changes at all our manuscript.

In more detail, in the submitted revised text in Sect. 3.2.1 it was:
"This issue is also reflected in the evaluation of the dust deposition field (Fig. 4d), with positive and negative biases over source regions but a clear underestimation of the deposited mass on transport and remote regions (e.g., the Southern Ocean). These discrepancies point towards an overestimation of dust deposition. For instance, EC-Earth3-Iron may share the difficulties of many global models in representing the long-range transport of dust, in particular coarse particles downwind of dust source regions (e.g., Adebiyi and Kok, 2020)."

and now it reads as:

"The evaluation of the dust deposition field (Fig. 4d) shows both positive and negative biases over source and transport regions (see Table A1), with the deposited mass being generally underestimated, except for the Southern Ocean where the model tends to overestimate the observations. EC-Earth3-Iron may, thus, share the difficulties of many global models in representing the long-range transport of dust, in particular, coarse particles downwind of dust sources (e.g., Adebiyi and Kok, 2020)"

Sincerely,

S. Myriokefalitakis